# Standard operating procedure combined with comprehensive quality control system for multiple LC-MS platforms urinary proteomics

Xiang Liu[1,20], Haidan Sun[2,20], Xinhang Hou[1,20], Jiameng Sun[2], Min Tang[1], Yong-Biao Zhang [1], Yongqian Zhang [3], Wei Sun[2] ✉, Chao Liu [1] ✉ & Urine Test Sample Working Group

Urinary proteomics is emerging as a potent tool for detecting sensitive and non-invasive biomarkers. At present, the comparability of urinary proteomics data across diverse liquid chromatography–mass spectrometry (LC-MS) platforms remains an area that requires investigation. In this study, we conduct a comprehensive evaluation of urinary proteome across multiple LC-MS platforms. To systematically analyze and assess the quality of large-scale urinary proteomics data, we develop a comprehensive quality control (QC) system named MSCohort, which extracted 81 metrics for individual experiment and the whole cohort quality evaluation. Additionally, we present a standard operating procedure (SOP) for high-throughput urinary proteome analysis based on MSCohort QC system. Our study involves 20 LC-MS platforms and reveals that, when combined with a comprehensive QC system and a unified SOP, the data generated by data-independent acquisition (DIA) workflow in urine QC samples exhibit high robustness, sensitivity, and reproducibility across multiple LC-MS platforms. Furthermore, we apply this SOP to hybrid benchmarking samples and clinical colorectal cancer (CRC) urinary proteome including 527 experiments. Across three different LC-MS platforms, the analyses report high quantitative reproducibility and consistent disease patterns. This work lays the groundwork for large-scale clinical urinary proteomics studies spanning multiple platforms, paving the way for precision medicine research.

Molecular biosignatures of altered proteomes in urine are promising in the diagnostic, prognostic, long-term disease progression, and treatment response monitoring of diseases for precision medicine studies, owing to the advantage of costs, time, and the noninvasive nature of urine[1–4]. Increasing numbers of urinary proteomic studies have been used to discover potential disease biomarkers across various diseases, such as urological cancers[5–7], colorectal cancer[2], virus infection[8,9], neurodegenerative disorders[3], and many others[10]. Recent advances in mass spectrometry techniques and informatic pipelines have greatly extended the research to large-scale clinical sample cohorts, to

[1]School of Biological Science and Medical Engineering & School of Engineering Medicine, Beihang University, Beijing, China. [2]Proteomics Center, Core Facility of Instrument, Institute of Basic Medical Sciences Chinese Academy of Medical Sciences, School of Basic Medicine Peking Union Medical College, Beijing, China. [3]School of Medical Technology, Beijing Institute of Technology, Beijing, China. [20]These authors contributed equally: Xiang Liu, Haidan Sun, Xinhang Hou. ✉e-mail: sunwei@ibms.pumc.edu.cn; liuchaobuaa@buaa.edu.cn

improve the statistical significance of the discovered biomarkers and yield more valuable clinical insights[3,11,12].

For large-scale cohort study, the assessment of reproducibility is crucial owing to the data usually generated across instruments, platforms, and laboratories[13,14]. Recent studies have demonstrated that data-independent acquisition (DIA) proteomic data generated across 11 laboratories revealed high consistency and reproducibility of identification and quantitation results for cell and tissue samples[15–17]. This was achieved by employing harmonized mass spectrometry (MS) instrument platforms and standard operating procedure (SOP)[15–17]. However, the unified SOP for bodily fluids, with higher complexity or wider protein concentration dynamic range samples as represented by urine have not been provided[11]. Currently, each platform applied its own experiment and data acquisition procedures, which results in a large variation across instruments, platforms, and laboratories[2,3,8,9,18].

For establishing and applying an SOP, there is still a lack of comprehensive quality control (QC) system to ensure reproducibility and robustness during data generation and maximize the accessibility of downstream data[19]. Over the years dozens of QC metrics have been proposed[20,21], generated by a range of bioinformatics tools, which contain NIST MSQC[20], QuaMeter[22,23], RawBeans[24], DO-MS[25,26], PTXQC[27], QCloud[28,29], QC-ART[30], MSstatsQC 2.0[31], QuiC[32], et al. However, these QC tools and metrics mainly focus on data-independent acquisition (DDA) experiments, and few of them were developed specifically for DIA experiments. In addition, these tools only extract a limited number of metrics and display them in the form of charts, without illustrating the relationship between the metrics and identification results. Users need to rely on expert experience to optimize parameters and locate experimental problems step by step, which is time-consuming and laborious[33]. Especially, for large-scale DIA experiments, a fully automated and comprehensive QC system with systematic evaluation and optimization for the entire LC-MS workflow is in great demand, in the case of evaluating the system performance, locating potential problems, detecting low-quality experiments, etc. These hindered the application of large-scale urinary proteomics into clinical research.

In this work, we first summarize and integrate the metrics extracted by existing QC software and introduce new metrics, striving to provide the most comprehensive set of proteomics QC metrics. We develop the MSCohort QC system to perform individual experiments and the whole cohort data quality evaluation. Especially, we propose a scoring system for individual DIA experiment evaluation and optimization. Based on this comprehensive QC system, we develop a SOP for high-throughput urinary proteome. We systematically investigate the consistency and reproducibility of urinary proteome across 20 LC-MS platforms with and without unified SOP, respectively. Then, benchmarking samples, consisting of tryptic digests of human urine, yeast, and *E. coli* proteins in defined proportions, are generated to mimic differential expressed biological samples and investigate the performance in the quantitative accuracy, precision, and robustness of the different platforms, as well as the capability to detect differentially expressed proteins (DEPs) deemed as biomarkers. Furthermore, the above SOP is applied to colorectal cancer (CRC) urinary samples across 3 different LC-MS platforms to further demonstrate the combination of a comprehensive QC system and unified SOP could guarantee high consistency and reproducibility in cohort clinical proteome analyses (Fig. 1). Taken together, our study provides a comprehensive QC system and reference SOP for large-scale urine proteomic analysis spanning different platforms, which could benefit the applications of urinary proteomics to clinical disease researches. The MSCohort software tool is accessible for download from Github (https://github.com/BUAA-LiuLab/MSCohort).

## Results

### Design of MSCohort for comprehensive data quality control
We developed the MSCohort QC system, which provides more comprehensive quality control metrics (Supplementary Note 1) and

enhances more extensive quality control functionalities (Supplementary Note 2), compared to existing quality control software. It assists users in assessing and optimizing individual experimental procedures, evaluating system stability across multiple experiments, and identifying outlier experiments (Fig. 2).

The primary phase of quality control involves the extraction of QC Metrics. The MSCohort QC system summarizes and integrates metrics extracted by existing quality control software and introduces 26 new metrics, totaling 81 QC metrics (Supplementary Note 1): (1) For individual experiment data quality evaluation, we investigated the QC metrics proposed by NIST MSQC[20], QuaMeter[22,23], DO-MS[26], RawBeans[24], Spectronaut[34], MSRefine[35], and added 10 metrics for DIA individual experiment. This comprehensive set, amounting to a total of 58 metrics, termed intra-experiment metrics in this study (Supplementary Data 1A); (2) For evaluating data performance across experiments, we investigated the QC metrics proposed by PTXQC[27], QCloud[28,29], QC-ART[30], MSstatsQC 2.0[31], QuiC[32], and added 16 new metrics. This yields a set of 23 metrics encompassing in precursor, peptide, and protein group levels, termed inter-experiment metrics in this study (Supplementary Data 1B). We strive to provide the most comprehensive set of proteomics QC metrics. The quality control software tool extracts relevant metrics tailored to specific data types (e.g., DDA or DIA, individual experiments or cohort experiments), then conducts the following quality evaluation, to generate detailed analysis reports.

For individual experiments, MSCohort extracts comprehensive metrics that map to the whole LC-MS workflow, illustrates the relationship between extracted metrics and identification results, scores the metrics, and reports visual results, assisting users in evaluating the workflow, and locating problems. We have previously proposed a quality evaluation system for individual DDA experiments[35], and here, we developed a quality evaluation system for individual DIA experiments. We designed a scoring formula to characterize different kinds of DIA experiments:

$$N_{identified\_precursors} = N_{acquired\_MS2} \times Q_{MS2} \times (N_{precursor\_per\_MS2} / R_{precursor})$$
(1)

where $N_{identified\_precursors}$ is the number of identified peptide precursors, $N_{acquired\_MS2}$ is the number of acquired MS2 scans, $Q_{MS2}$ is the identification rate of the MS2 scans, $N_{precursor\_per\_MS2}$ is the spectra complexity of MS2 scans, $R_{precursor}$ is the precursors duplicate identification rate, and $N_{precursor\_per\_MS2} / R_{precursor}$ is the utilization rate of the MS2 scans. MSCohort scores the relative metrics, reports metric–score diagram, and flags the metrics with low scores to assist experimenters in assessing the quality of data directly, enabling systematic evaluation and optimization of individual DIA experiments (See Methods and Supplementary Fig. 1 for details).

For the cohort proteomics data, it is imperative not only to conduct meticulous quality control analysis on individual experiments but also to perform longitudinal tracking to evaluate the performance over time for the cohort experiments. MSCohort reports corresponding scores to each of the inter-experiment metrics, and provides a heatmap overview, which yields an assessment of the quality at a glance and facilitates pinpointing the low-quality experiments (Supplementary Fig. 2). MSCohort also incorporates unsupervised machine learning algorithm (isolation forest) to detect potential outlier experiments. Furthermore, to guarantee the reliability of the subsequent statistical analyses, it incorporates various normalization methods to remove systematic bias in peptide/protein abundances that could mask true biological discoveries or give rise to false conclusions[36] (See Methods for details).

Notably, MSCohort serves as a potent tool that offers comprehensive support for data originating from diverse vendor platforms, including the Thermo Scientific Orbitrap, Bruker timsTOF, and SCIEX ZenoTOF systems. This tool facilitates rigorous quality assessment and

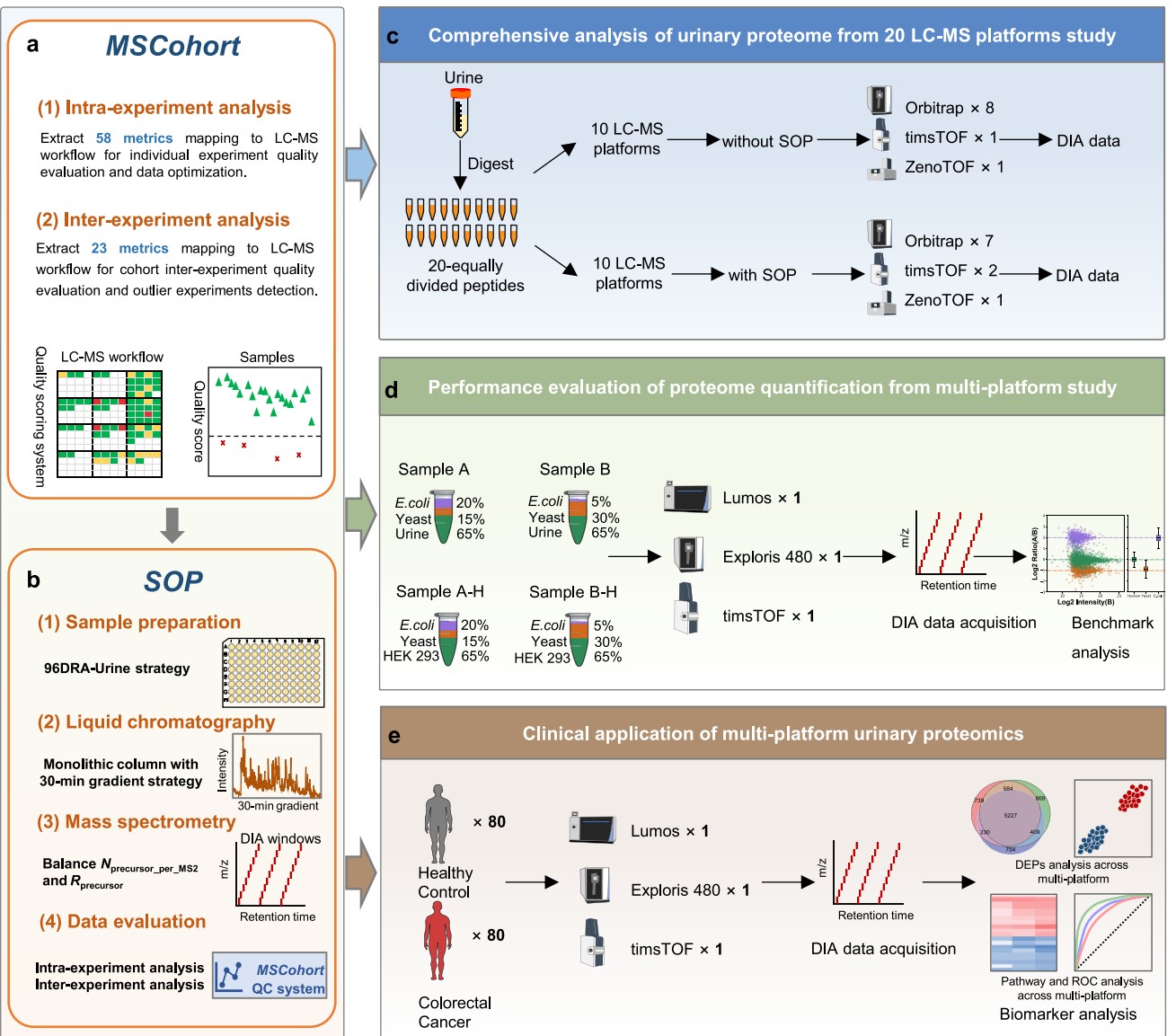

**Fig. 1 | Overall study design and implementation. a** The MSCohort quality control (QC) system extracts 81 metrics (58 intra-experiment metrics and 23 inter-experiment metrics) and supports both intra-experiment analysis and inter-experiment analysis, to facilitate the comprehensive quality evaluation of individual experiments and cohort DIA datasets, and assists users in monitoring the entire workflow performance, detecting potential problems, providing optimizing direction, flagging low-quality experiments, and improving experimental outcomes for subsequent analyses. **b** The standard operating procedure (SOP) for urinary proteomics, which integrates the optimized strategies at each step, including the 96DRA-Urine sample preparation strategy, a monolithic column with 30-min gradient strategy, balance $N_{percursor\_per\_MS2}$ and $R_{precursor}$ DIA-based MS method, and comprehensive MSCohort QC system. $N_{percursor\_per\_MS2}$ is the spectra complexity of MS2 scans, $R_{precursor}$ is the precursor duplicate identification rate. **c** A 20 LC-MS platforms analysis under a comprehensive quality control system of urinary proteomics was performed to analyze the variation and the consistency among different LC-MS platforms. **d** Benchmarking samples were prepared containing known ratios of peptide digestions from human, yeast and *E. coli* organisms, to mimic differential expressed biological samples and provide proof of the quantitative robustness and reproducibility of the different LC-MS platforms under unified SOP. **e** Clinical colorectal cancer (CRC) urinary proteome datasets derived from 3 different LC-MS platforms were performed to analyze the performance of urinary proteomics from multi-platform in biomarker discovery.

optimization of both DDA and DIA experiments, as well as inter-experimental quality control analysis across DDA, DIA, and PRM studies. MSCohort applies not only to urinary proteomics analysis but also to the analyses of other samples (such as cell, tissue, blood, etc.). The MSCohort software tool and the user manual are available from Github: https://github.com/BUAA-LiuLab/MSCohort.

**Optimization and establishment of SOP for urinary proteomics with MSCohort QC system**

To meet the demands of extensive clinical urinary proteomics analysis, we have developed an SOP for urinary proteomics that demonstrates high throughput, sensitivity, and reproducibility (See Fig.3a and Supplementary Note 3 for details).

Herein, we underscore the importance of incorporating a data quality control step in the development and implementation of the SOP. Previous studies have reported the SOPs for other proteomics applications[16,23,37], yet they primarily rely on manual inspection of identification results and limited metrics to assess experimental quality. Consequently, it becomes difficult to promptly identify and address issues or propose specific strategies for further experimental refinement when identification results are not satisfactory. Supplementary Note 4 showed how to use the comprehensive MSCohort QC

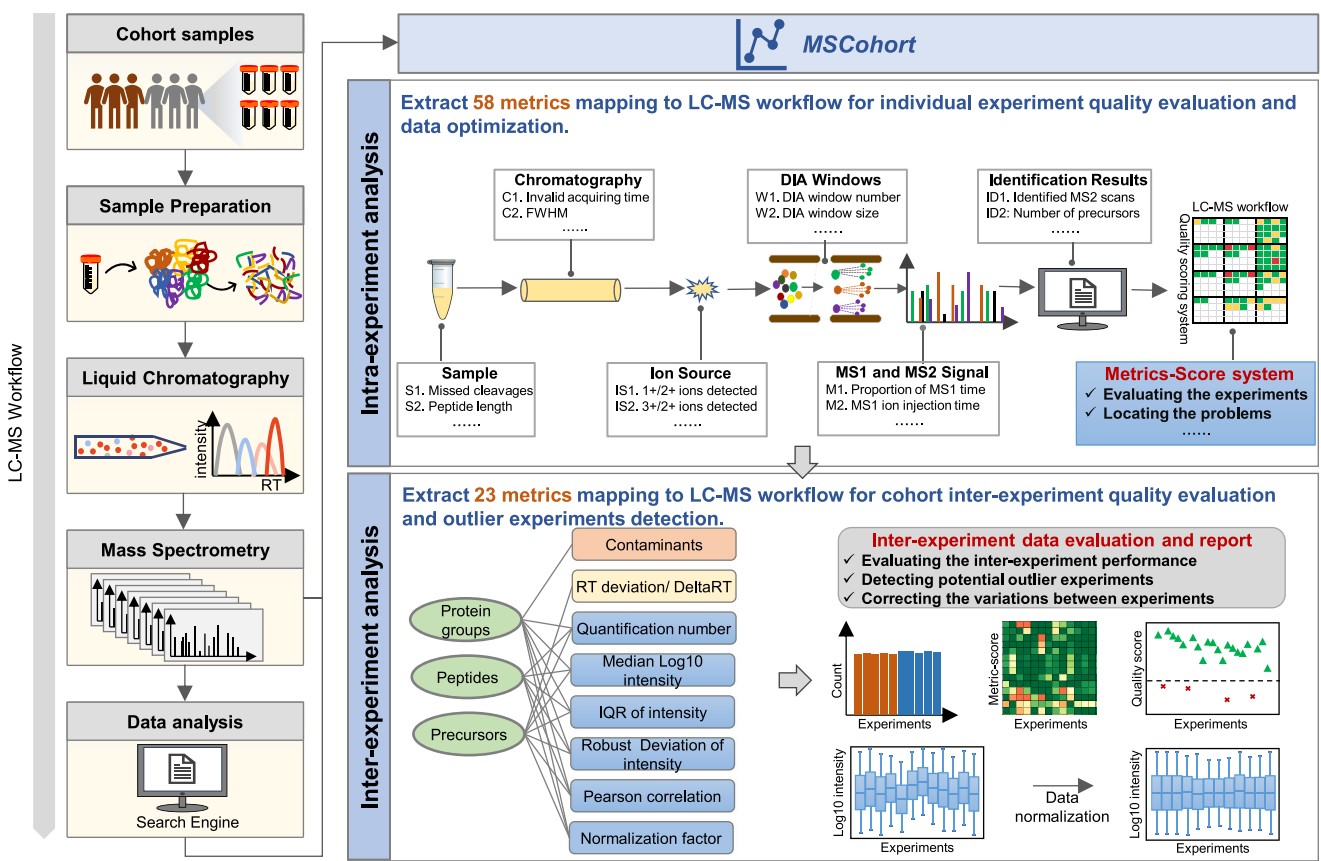

**Fig. 2 | The workflow of MSCohort.** The LC−MS based proteomics workflow includes multiple procedures: sample preparation, liquid chromatography, mass spectrometry, and data analysis. MSCohort consists of two modules: Intra-experiment analysis and inter-experiment analysis. Intra-experiment module extracted 58 metrics over the whole LC−MS workflow, and divided them into sample, chromatography, DIA windows, ion source, MS1 and MS2 signal, and identification result categories. We established a metric-score system for individual experiment quality evaluation and data optimization. Inter-experiment module extracted 23 metrics in precursor, peptide, and protein groups level mapping to LC−MS workflow, to facilitate the comprehensive quality evaluation for large cohort data, flag potential outlier data, and correct data heterogeneity for subsequent analyses.

system to optimize the LC-MS methods to establish the SOP for urinary proteomics. Through the DIA scoring formula (1) and comprehensive metrics analysis reports, MSCohort provides direct explanations for the underlying causes of data results. By conducting a limited number of experiments instead of iterating through all parameter combinations, we could achieve the whole procedure optimization of urinary proteomics, including sample (Figure S.Note 4.1), chromatography (Figure S.Note 4.2), and mass spectrometry (Figure S.Note 4.3), and established the SOP for urinary proteomics.

This SOP integrates the optimal strategies at each step, including the 96DRA-Urine[38] (Direct reduction/alkylation in urine) high-throughput sample preparation method, stable and efficient chromatography system, highly sensitive and high throughput DIA-based MS method[16], and comprehensive MSCohort QC system. As illustrated in Supplementary Note 5, the adoption of this SOP yields the following benefits: (i) Pretreating of nearly 200 samples in a single day, meeting the demands of large-scale analysis; (ii) Identifying over 3000 protein groups in a single sample within 30-min gradient. Figure 3b demonstrates that the SOP increases protein identification numbers per unit time by 3 to 90 times compared to representative methods; (iii) Achieving excellent inter-experimental stability with a retention time deviation of less than 0.2 minutes for the same peptides over 7 days (Figure S.Note 4.2). To the best of our knowledge, this SOP presents the deepest urinary proteome coverage for single-run analysis in short gradient, a promising basis for the discovery of urinary biomarkers in large-scale sample cohorts (Fig. 3b).

## Comprehensive and comparative analysis of urinary proteome data from multi-platform study

To validate the performance of this SOP across different LC-MS platforms, we performed urinary proteomics experiments collected across multiple LC-MS platforms including different types of mass spectrometers. We prepared a urine peptide QC sample and distributed aliquots to 20 LC-MS platforms, which were classified into two groups: 1) Ten platforms employed the unified SOP developed above (numbered U01-U10), termed with LC-SOP in this study (Supplementary Note 3). 2) Another ten platforms without LC-SOP. These LC-MS platforms were encouraged to use their individual optimized experimental parameters (numbered M01-M10)[39]. The detailed data acquisition parameters were provided in Supplementary Data 2.

Firstly, the qualitative results showed clear difference among the 10 LC-MS platforms without LC-SOP (M01-M10), the number of identified proteins ranged from 2371 (M03) to 3695 (M07), with a relative standard deviation (RSD) of 8% (Fig. 4a and Supplementary Data 3A). Among them, four platforms (M03, M04, M06, and M09) showed obviously lower identification results than the others. We investigated the possible causes in detail based on MSCohort. As illustrated in Supplementary Fig. 3 and Supplementary Data 4, the metric scores related to the identification rate of MS2 scans were low for M03-E480[F] (Exploris 480 with a High Field Asymmetric Waveform Ion Mobility Spectrometry (FAIMS)). Its peak intensities and peak counts of MS1 and MS2 were significantly lower than other instrument platforms, resulting in a low MS2 identification rate (49%). Consequently, the final identified protein number was also reduced. M04-E480[F] also showed a

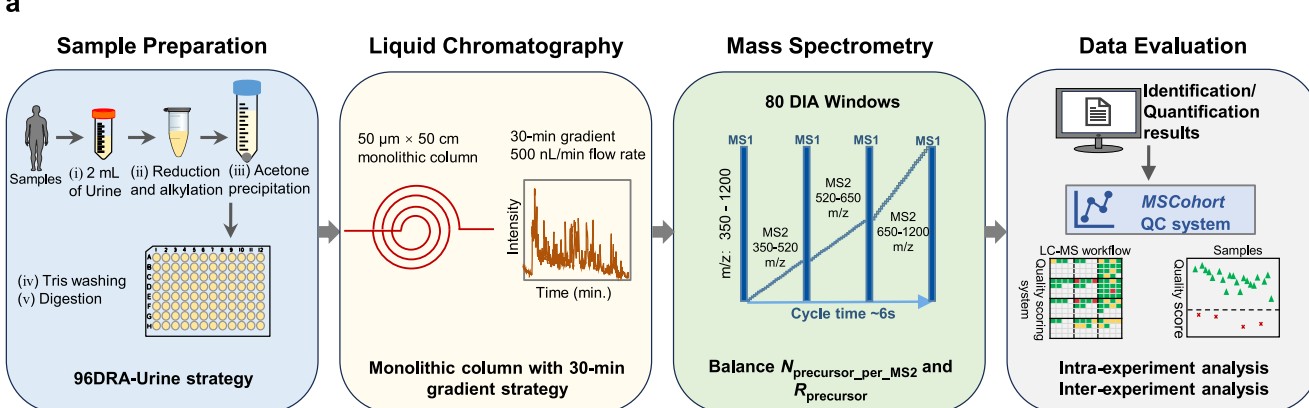

**b**

| Year | Instrument | Method | Acquisition time | Average identified proteins in one sample | Average identified proteins per minute | Samples number | Total non-redundant proteins | Reference |
|---|---|---|---|---|---|---|---|---|
| 2019 | TripleTOF 5600 | DDA 2D LC-MS/MS | 50 min ×8 faction | 1738 | 4.5 | 49 | 3008 | (Shao, et al. 2019) [4] |
| 2019 | TripleTOF 5600 | iTRAQ 2D LC-MS/MS | 60 min ×20 faction | 2005 | 1.7 | 14 | 2086 | (Zou, et al. 2019)[10] |
| 2019 | Orbitrap Fusion Lumos | DIA DDA - library | 60 min | 1763 | 29.4 | 36 | 2236 | (Zhao, et al. 2019)[18] |
| 2020 | timsTOF Pro | DIA-PASEF DDA - library | 90 min | 4000 | 44.4 | 37 | 5991 | (Tian, et al. 2020)[9] |
| 2021 | Q Exactive HF-X | DIA DDA - library | 45 min | 2162 | 48.0 | 235 | 3049 | (Winter, et al. 2021)[3] |
| 2022 | Orbitrap Fusion Lumos | TMT 2D LC-MS/MS | 45 min ×24 fraction | 2869 | 2.7 | 36 | 4229 | (Sun, et al. 2022)[2] |
| 2022 | Q Exactive HF | TMT 2D LC-MS/MS | 60 min ×30 fraction | 2662 | 1.5 | 90 | 3854 | (Bi, et al. 2022)[8] |
| 2024 | Orbitrap Exploris 480 | DIA directDIA | 30 min | 3443 | 114.8 | 160 | 7089 | This study |
| 2024 | Orbitrap Fusion Lumos | DIA directDIA | 30 min | 3013 | 100.4 | 160 | 6780 | This study |
| 2024 | timsTOF Pro 2 | DIA directDIA | 30 min | 4291 | 143.0 | 160 | 6620 | This study |
| 2024 | 3 instruments | DIA directDIA | 30 min | / | / | 480 | 8812 | This study |

**Fig. 3 | The developed standard operating procedure (SOP) for urinary proteomics. a** The SOP integrates the optimized strategies at each step, including the 96DRA-Urine sample preparation strategy, monolithic column with 30-min gradient strategy, balance $N_{precusor\_per\_MS2}$ and $R_{precursor}$ DIA-based MS method, and comprehensive MSCohort QC system. $N_{percusor\_per\_MS2}$ is the spectra complexity of MS2 scans, $R_{precursor}$ is the precursors duplicate identification rate. **b** Summary of key experimental parameters and results comparing urinary proteome analysis data published in the last five years and data obtained using the unified SOP in this study. The average identified proteins per minute increased by 3 (143/48) to 90 (143/1.5).

similar low identification rate of MS2 scan (64%). Additionally, the chromatographic peak width and full width at half maximum (FWHM) of M04 were significantly wider than other instrument platforms, which also affects the LC separation efficiency and identification rate. For M06-Eclipse, the precursors duplicate identification rate was higher than the complexity of the MS2 scans, which led to the low utilization rate of the MS2 scans ($N_{precursor\_per\_MS2}/R_{precursor} = 0.74$). M06 also showed a short MS2 ion injection time and low signal-to-noise ratio, which was correlated with its relatively low setting threshold (50000) of AGC target. The chromatographic invalid acquiring time (LC delay time) for M09-E240 (Exploris 240) was 12 minutes, leading to over 40% of spectra being wasted without identifying precursors. Consequently, the overall identification rate was only 50%. The above metric-score results demonstrated that

MSCohort can effectively and accurately locate potential problems and provide clear insights for DIA optimization. Furthermore, we also assessed the consistency of identification results among the 10 instrument platforms. Only 2045 proteins were found to overlap across all platforms (Fig. 4b), constituting 55%–86% of the identified proteins on individual instrument platforms, which indicates relatively low consistency of identification results. After removing 4 platforms with lower identification results, the overlap proteins increased to 2876, showing an improved qualitative consistency (Supplementary Fig. 4).

We also analyzed the precision and reproducibility of quantitative results among the 10 platforms without LC-SOP using MSCohort. As illustrated in Fig. 4c, d, the median coefficients of variation (CV) of protein intensity for each LC-MS platform were below 20% and the

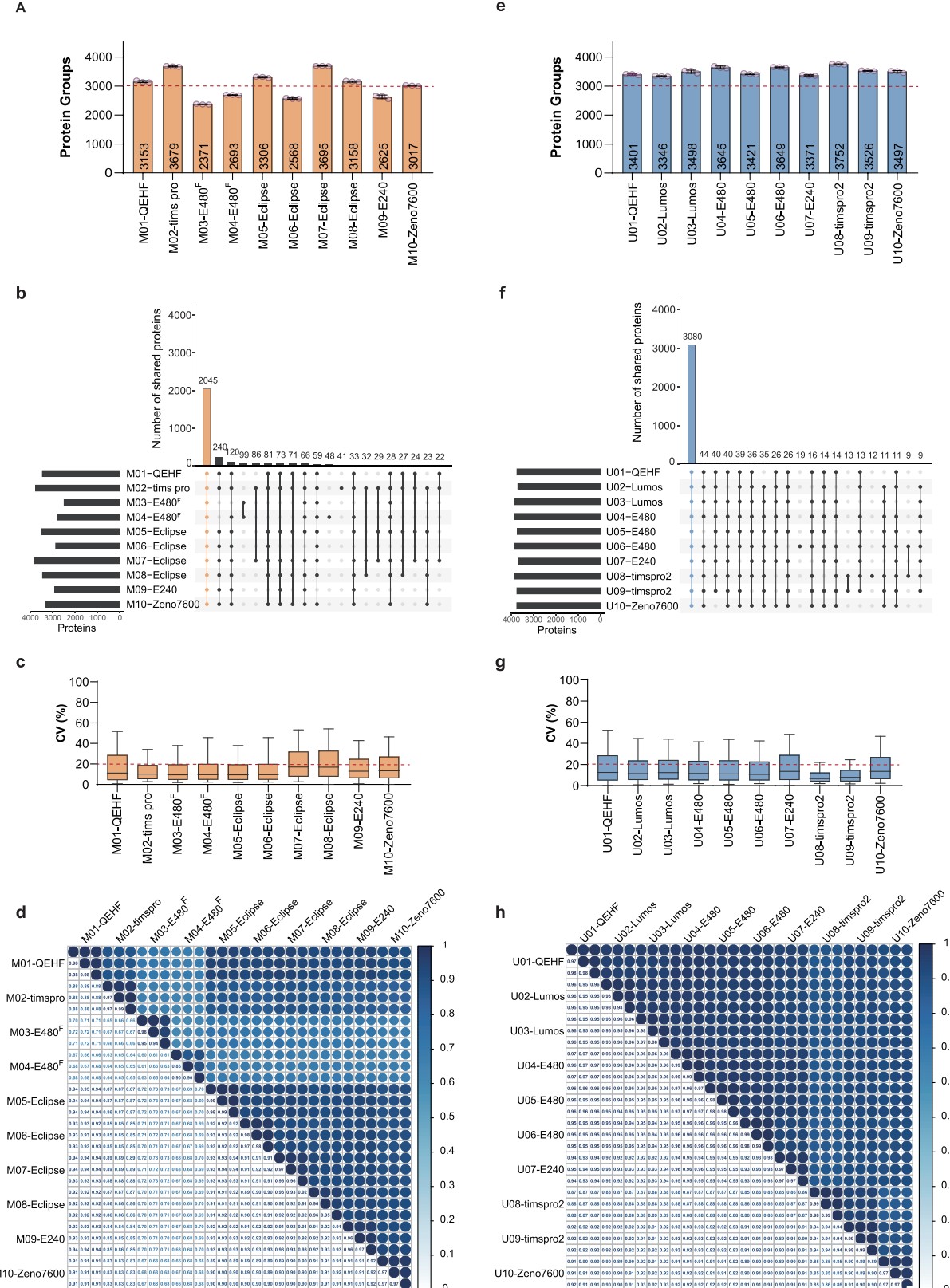

Pearson correlation coefficients of protein intensity were greater than 0.96, demonstrating that DIA approaches achieved high quantitative precision and reproducibility in the intra-instrument level. For inter-instrument analysis, Pearson correlations of M03 and M04 with other LC-MS platforms were low (0.6–0.73). The Pearson correlation between the other 6 Orbitrap instrument platforms was 0.9-0.94, and

the Pearson correlation between timsTOF and Orbitrap instrument was 0.83–0.89.

While, the qualitative and quantitative results showed higher consistency and reproducibility among the 10 LC-MS platforms with LC-SOP (U01-U10), the range of identified proteins was 3346–3752, with RSD of the identified numbers less than 4% (Fig. 4e). Within

**Fig. 4 | The qualitative and quantitative performance across 20 LC-MS platforms.** The number of proteins identified (with 1% FDR) from the 10 platforms without LC-SOP (**a**, orange) and with LC-SOP (**e**, blue). Three technical replicates were collected in each platform. The average number was shown at the bottom of the bar, and the individual data points were indicated by purple dot points (*n* = 3), with error bars representing +/− SD. Horizontal dashed lines were added at 3000, as a reference value. **b** and **f** The upset plot showed the numbers of proteins co-identified across various combinations of 10 platforms without LC-SOP (b) and with LC-SOP (f) (vertical bars). The colored vertical bar reflected the number of proteins co-identified by all 10 platforms without LC-SOP (b, orange) and with LC-SOP (f, blue). The horizontal bars

reflected the number of proteins identified in total at each platform. **c**, **g** The distribution of the coefficients of variation (CV) obtained on the quantified protein intensity across the three technical replicates was plotted for each platform. All box plots indicated the median and the first and third quartiles as the box ends. Whiskers were positioned 1.5-times in the interquartile range. Horizontal dashed lines were added at 20% CV, as a reference value. The Pearson correlation matrix based on log2 protein intensity of the common proteins from 10 platforms without LC-SOP (**d**) and with LC-SOP (**h**). The color-scale indicated the magnitude of the Pearson correlation coefficient. 20 platforms arranged in chronological order based on the MS release dates. Source data are provided as a Source Data file.

30 minutes, each LC-MS platform was able to identify more than 3300 proteins. Among them, 3080 proteins overlap across all platforms, showing high qualitative consistency (Fig. 4f). In addition, MScohort analysis report among 10 platforms with LC-SOP also showed good consistency (Fig. 4g and Supplementary Data 4). In particular, the Pearson correlation of protein intensity between the 7 Orbitrap instruments was 0.93–0.97, and the Pearson correlation between timsTOF and Orbitrap instrument was 0.86–0.92 (Fig. 4h). Besides, the average sequence coverage and the dynamic range of the proteins that were identified among the 10 LC-MS platforms with LC-SOP also improved than that in 10 LC-MS platforms without LC-SOP.

Furthermore, we sought to utilize data collected by multiple Orbitrap instrument platforms to investigate the effects of different LC and MS conditions on the reproducibility of results. We classified different platforms into 3 groups based on the conditions, A (same MS condition, same LC condition), B (different MS condition, same LC condition), C (different MS condition, different LC condition) (Supplementary Fig. 6) (M03, M04, M06, M09 were not included in the group). The results displayed that the data collected with the same LC condition (group A and B) outperformed data without LC-SOP (group C), and group A with the same LC and MS conditions showed the highest reproducibility. Furthermore, group B with the same LC condition and different MS type also showed good qualitative and quantitative reproducibility, with RSD of qualitative data <2% and Pearson correlation of quantitative data > 0.9. Above group A and group B results were comparable to the previous inter-instrument reproducibility results obtained from harmonized instruments[15].

## Performance evaluation of proteome quantification and detection of differentially expressed proteins from multi-platform study

To evaluate the quantitative accuracy, precision, and sensitivity of urinary proteome from different platforms under the unified SOP, we prepared benchmarking samples, with adding yeast and *E. coli* peptides in specified distinct ratios to a complex urine peptide background. Sample A was 65% human urine peptides, 15% yeast, and 20% *E. coli*, and Sample B was 65% human urine peptides, 30% yeast, and 5% *E. coli*, similar to previous benchmarks[21]. We analyzed samples A and B in technical triplicates on three LC-MS platforms, Orbitrap Fusion Lumos, Orbitrap Exploris 480, and timsTOF Pro 2 (Lumos, E480, and TIMS for short) based on the above with LC-SOP data acquisition methods.

Firstly, we conducted quality assessment for the data generated from the three platforms using MSCohort. Three technical repetitions within each platform demonstrated high qualitative and quantitative repeatability, with RSD of the identified numbers less than 1% and median CV of protein intensity below 15% (Supplementary Fig. 7a,b). For inter-instrument analysis, within 30 minutes, the total identified protein numbers were 4953, 5224, 6006 in Lumos, E480, and TIMS, respectively. Among them, 4667 proteins overlap across 3 platforms, showing high qualitative consistency (Supplementary Fig. 7c). The Pearson correlation of protein intensity among the 3 LC-MS platforms was 0.93–0.97, which was comparable with that in the above 10 platforms with LC-SOP (Supplementary Fig. 7d).

Next, more precise assessment of quantitative performance was performed based on benchmarking samples. The number of quantified proteins from different species among 3 LC-MS platforms was shown in Fig. 5a–c (Supplementary Data 5). Different species proteins correspond to different theoretical quantitative ratios (1:1, 1:2, 4:1 and the log2 ratio is 0, 1, 2 for human, yeast, and *E. coli*, respectively). As illustrated in Fig. 5d–f, the median values of the log2 ratios were 0.02 – 0.07, (-0.86) – (-0.74), 1.65 – 1.91, and the medians of relative deviation from theoretical ratio were 0.09 (0.05 – 0.11), 0.15 (0.11 – 0.2), 0.18 (0.15 – 0.18) for human, yeast and *E. coli* proteins, respectively. The median CV values were typically below 15% for Lumos and E480 data and below 10% for TIMS data (Fig. 5g–i). These results demonstrated that excellent label-free quantitative accuracy and precision were achieved in different LC-MS platforms.

As the ultimate goal of most proteomic analysis is to detect differentially expressed proteins (DEPs) from different conditions, we assessed the sensitivity and specificity of different LC-MS platforms in DEP detection using our benchmarking data sets. DEPs were extracted using the widely applied criteria of a fold change >1.5 and an adjusted *p*-value < 0.05. In this condition, yeast and *E. coli* in urine could both be considered as significantly different proteins, as their expected ratios are 2-fold and 4-fold, respectively. Three LC-MS platforms showed high sensitivity of DEP detection from Lumos data (67.5% for yeast and 94.5% for *E. coli* quantified proteins as DEPs), E480 data (74.7% for yeast and 94.0% for *E. coli* quantified proteins as DEPs) and TIMS data (80.4% for yeast and 96.0% for *E. coli* quantified proteins as DEPs) (Fig. 5j). Moreover, in the pairwise comparison of human proteins with an expected ratio of 1:1, three LC-MS platforms resulted in detection of false DEPs at comparable rates (0.2%, 0.2%, 1.1% false DEPs rates for Lumos, E480 and TIMS data, respectively) (Fig. 5j). Furthermore, we assessed the robustness of DEP detection based on the receiver operating characteristic (ROC) curve analysis, which led to the similar conclusion (Fig. 5k). In summary, all three LC-MS platforms could detect DEPs from the benchmarking dataset with high sensitivity and specificity. In addition, we provided further proof by applying this workflow to benchmark samples mixed with human HEK 293 cell, yeast, and *E. coli*, labeled Sample A-H and Sample B-H (Supplementary Fig. 8).

Collectively, these results demonstrated that under the unified SOP, most of the DEPs could be recalled accurately in complex samples of mixed species with high quantitative accuracy and precision even in different type of LC-MS platforms, which would broadly increase the confidence in DIA-based urinary proteomics as a reproducible method for large cohort protein quantification.

## Analyses of cohort clinical proteomics from multi-platform study

We further illustrated the generalization by applying the above urinary proteomics SOP to clinical biomarker discovery research in different platforms. Herein, we collected a clinical cohort comprising 80 urine samples from colorectal cancer (CRC) patients and 80 samples from matched healthy controls (HC) (The detailed clinical information is shown in Supplementary Data 6). Based on the above SOP, LC-MS data collection for these 160 samples was conducted on Lumos, E480, and

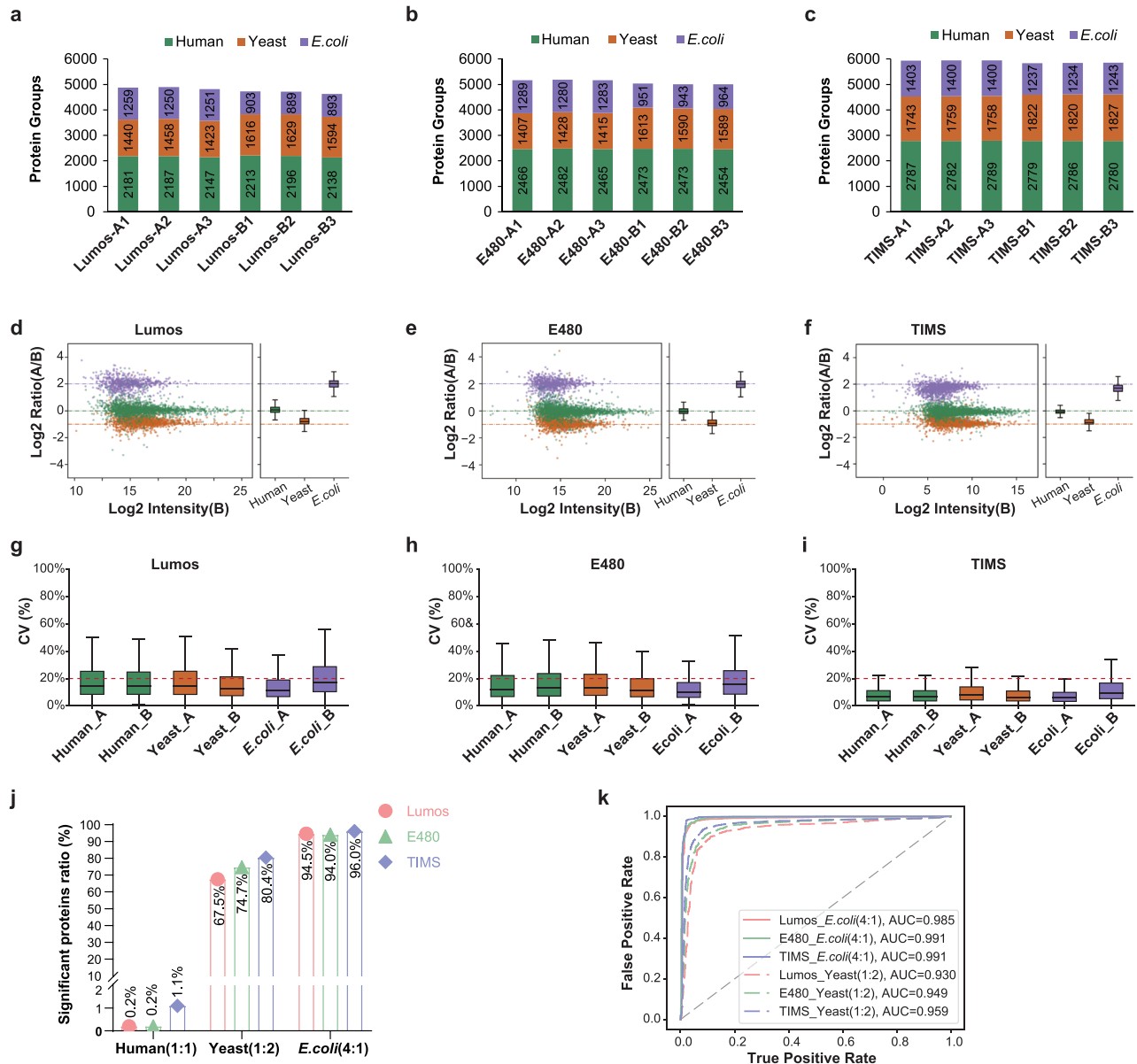

**Fig. 5 | Overall quantitative performance of benchmarking sample-A and sample-B from 3 LC-MS platforms.** Benchmarking samples A and B were prepared containing known ratios of peptide digestions from human urine, yeast and *E. coli*, resulting in expected peptide and protein ratios of 1:1 (A/B) for human, 1:2 for yeast and 4:1 for *E. coli* proteins. The samples A and B were analyzed in three technical replicates on three LC-MS platforms Orbitrap Fusion Lumos, Orbitrap Exploris 480, and timsTOF Pro 2 (Lumos, E480, and TIMS for short), respectively. **a**–**c** The number of proteins identified for each individual organism from sample-A and sample-B in Lumos, E480, and TIMS, respectively. **d**, **e** Log-transformed ratios (log2(A/B)) of proteins plotted over the log-transformed intensity of sample B in Lumos, E480, and TIMS (proteins). Colored dashed lines represented the expected log2(A/B) values for human (green), yeast (orange), and *E. coli* (purple) proteins. All box plots indicated the median and the first and third quartiles as the box ends. Whiskers were positioned 1.5-times the interquartile range. **g**–**i** The distribution of the coefficients of variation (CV) obtained on the quantified protein intensity across the three technical replicates was plotted for each individual organism. All box plots indicated the median and the first and third quartiles as the box ends. Whiskers were positioned 1.5-times the interquartile range. **j** Differentially expressed protein (DEP) detection for three organisms from Lumos, E480, and TIMS, respectively. Percentages of significantly changed proteins as DEPs over the total number of quantified proteins in 1:2 and 4:1 condition were used to estimate the sensitivity, while those in 1:1 condition were used to estimate the specificity. **k** Sensitivity and specificity of the DEP analysis based on receiver operating characteristic (ROC) curves. Source data are provided as a Source Data file.

TIMS, respectively (see Methods). QC samples were randomly analyzed during the collection process for systematic evaluation of reproducibility. In total, the three platforms generated 527 DIA experiments including 47 QC experiments.

In the process of cohort experiments collection, an intra-experiment analysis based on MSCohort QC system was performed for each newly collected experiment to evaluate the individual data quality. After the number of experiments collected exceeded 2 runs, an inter-experiment analysis based on MSCohort QC system was performed to evaluate the stability and reproducibility of the instrument system. Finally, after all the samples have been collected on one instrument platform, inter-experiment analysis based on MSCohort QC system was performed to evaluate cohort data quality and detect low-quality experiments.

This cohort experiment is a time course study with at least five consecutive days of acquisition on each instrument platform. We analyzed the overall cohort data quality based on the MSCohort QC system. First, the QC samples demonstrated good technical repeatability, with median Pearson correlation of protein intensity > 0.94 for each of the 3 LC-MS platforms (Supplementary Fig. 9a), indicating good LC-MS system stability. The results showed that the overall chromatographic retention time was stable (the average retention time deviation <0.25 min) for 7 consecutive days (Figure S.Note 4.2 e). In addition, MSCohort detected and reported low-performance experiments based on isolation forest algorithm, as shown in Supplementary Fig. 9b, 8 low-quality samples were reported in TIMS. Among them, 7 and 6 samples were also reported in Lumos and E480, respectively. The corresponding heatmap in MSCohort report indicated that there were significant differences between these 8 samples and other samples in multiple inter-experiment metrics (at least 7 of 23 metrics showed a variation more than two standard deviations (SD) from its median). Among them, 5 samples (D943, D1116, D1412, H771, D994) showed higher ratios of contaminants (erythrocytes, cellular debris, or serum high abundance proteins) than the other samples, which resulted in the sample-to-sample variability compared to regular urinary proteins (Methods). Another 3 samples (H349, D1036, D1069) showed lower Pearson correlation, and higher robust standard deviation at precursor, peptide, and protein groups intensity with the other samples, indicating these samples were heterogeneous compared with other samples (Supplementary Fig. 9c–e). Thus, these 8 experiments were excluded from further analysis (Supplementary Fig. 9b).

A total of 6780, 7089, and 6620 protein groups were identified in Lumos, E480, and TIMS, respectively. And 5227 proteins overlap across the three LC-MS platforms (Fig. 6a, Supplementary Fig. 10a). Subsequently, we analyzed the quantitative consistency among the three LC-MS platforms. The unsupervised learning t-Distributed Stochastic Neighbor Embedding (t-SNE) results showed that the results from three different LC-MS platforms for the same sample cluster together, and no platform effect was observed (Fig. 6b, c, Supplementary Fig. 10b). In the Orthogonal Projections to Latent Structures Discriminant Analysis (OPLS-DA) model, the CRC and Control groups can be clearly separated, and 100-fold cross-validation experiments indicate that the model did not overfit (Supplementary Fig. 10c–f). These results demonstrated the good consistency and reproducibility of the 3 LC-MS platforms.

Different expressed proteins analysis showed that a total of 455, 539, and 679 proteins were reported as DEPs (Benjamini-Hochberg adjusted $p$-value < 0.05) in Lumos, E480, and TIMS, respectively (Fig. 6d and Supplementary Data 8). And 215 DEPs were overlapped across 3 platforms. CRC/HC fold changes of these 215 proteins were highly correlated with Pearson correlation coefficients at $r = 0.99$, $r = 0.95$, and $r = 0.95$ for the comparisons of E480 and Lumos, E480 and TIMS, and Lumos and TIMS, respectively (Fig. 6e–g). Above results showed the high quantitative reproducibility of three platforms.

According to function annotation, the top enriched pathways for DEPs of 3 LC-MS platforms showed a high degree of consistency, with upregulation of cell proliferation, inflammatory response, and metabolism pathways (actin cytoskeleton signaling, acute phase response signaling, complement system, etc.) and downregulation of cell death and apoptosis-related pathways (FAK Signaling, FAT10 Cancer Signaling Pathway, etc.) (Fig. 6h, Supplementary Data 9). These results were consistent with the previous reports that tumor proliferation, migration, and metabolism modules were activated and cell death and apoptosis were inhibited in CRC patients[2]. We also compared our results with previous CRC tissue proteomics analysis study[40]. GO enrichment analysis (Supplementary Fig. 11) showed that extracellular matrix proteins were both enriched in urine and tissue and ribosome proteins were only enriched in tissue. The proteins involved in complement activation, immune response, and cell growth/development

process were highly enriched in urine, and the proteins involved in glucose metabolic, amino acid metabolic, and ribonucleotide metabolic processes were highly enriched in tissue. In addition, the proteins involved in cell adhesion, coagulation, and regulation of peptidase activity were both enriched in urine and tissue. These results suggested that urinary proteome changes could reflect not only the changes of tissue[5], but also the changes of body immune systems, etc.

Among them, the protein related to complement activation was significantly upregulated and showed a high degree of consistency (Supplementary Fig. 12). Complement is a key player in the innate immune defense against pathogens and the maintenance of host homeostasis[41]. In the tumor-immune interaction, complement-associated proteins play a vital role whether directly or indirectly by regulating tumorigenesis, development, and metastasis[42]. In CRC, tumor cells were found to produce Complement C3 (C3) component thus leading to modulation of the response of macrophages and its anti-tumor immunity, via the C3a-C3aR axis and PI3K signaling pathways. The complement C5a/C5aR pathway was found to induce cell proliferation, motility, and invasiveness[43]. Complement components C5b, C6, C7, C8, and C9 form the membrane attack complex (MAC), MAC accumulation on the cell membrane promotes cell proliferation and differentiation, inhibits apoptosis, and protects cells against complement-mediated lysis in a sublytic density[42,44–46]. Complement factor H (CFH) and Complement factor I (CFI) modulated the fundamental processes of the tumor cell, promoting proliferation and tumor progression when tested in animal models[47,48]. Collectively, these highlighted the value of the complement system in tumor progression, especially that of CRC.

Next, we evaluated these 67 upregulated proteins of 215 proteins as input variables and investigated the classification performance in the CRC/HC stratification. The top 15 proteins with the highest area under the curve (AUC) of each protein in 3 LC-MS platforms were chosen as candidates (Supplementary Fig. 13a) and 8 common highly ranked DEPs in 3 platforms were submitted to further machine learning model building (Supplementary Data 10). We compared six machine learning classifiers and performed cross-validation by training the model on one platform and testing it on the other two platforms. Finally, a 5-protein panel (C9, CFI, CFH, RELT, GDF15) showed the highest and reproducible AUC values in 3 different LC-MS platforms (Supplementary Fig. 13). On average, the Support Vector Machines (SVM) model showed the highest AUC values of 0.88, 0.93, and 0.89 for the classification of CRC and HC in Lumos, E480, and TIMS, respectively (Fig. 6i–l). A previous study reported that C9 was significantly upregulated in colorectal cancer plasma[49]. Growth/differentiation factor 15 (GDF15) is a divergent member of the transforming growth factor-b (TGF-b) superfamily. Experimental evidence shows that GDF15 enhances tumor growth, stimulates cell proliferation, and promotes distant metastases[50]. Previous blood and colorectal tumor samples from 2 large studies also found high plasma levels of GDF15 before diagnosis of CRC are associated with greater CRC specific mortality[51].

Taken together, three different LC-MS platform data indicated consistent and excellent performance for biomarker discovery and patient stratification in CRC. These results demonstrated the generalization of urinary proteomics to support clinical discovery proteomics research under the condition of a unified SOP and MSCohort QC system.

## Discussion

Large-scale cohort studies usually involve multi-center and long-term experiments for which comprehensive QC system is needed to ensure reproducibility and robustness during data generation and integration[16,30]. In this study, we developed MSCohort QC system to perform urinary individual DIA experiment and the whole cohort data quality evaluation. MSCohort extracted 70 metrics covering the intra-

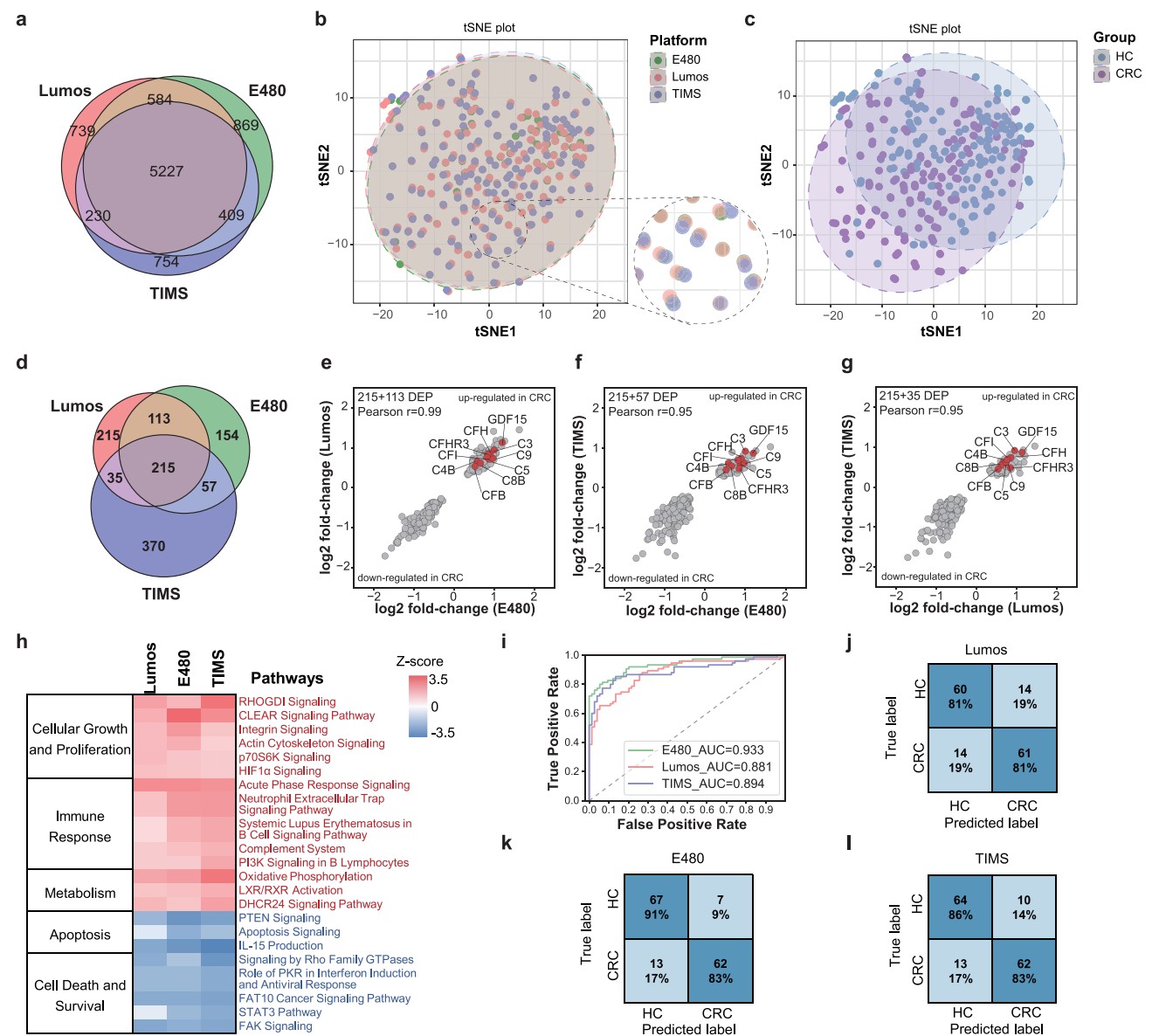

**Fig. 6 | Clinical colorectal cancer urinary proteomics analysis among three LC-MS platforms. a** Overlap of identified protein groups from Lumos, E480, and TIMS. **b**, **c** The unsupervised learning t-Distributed Stochastic Neighbor Embedding (t-SNE) plot overview of urinary proteomics among Lumos, E480, and TIMS (b) and colorectal cancer (CRC) and healthy control (HC) group. **d** The number of proteins that significantly different (*q*-value < 0.05) in abundance by CRC and HC within each platform. **e-g** Correlation of protein CRC/HC fold changes in pairwise combinations of three platforms. Combinations are E480 vs. Lumos (**e**), E480 vs. TIMS

(**f**), Lumos vs. TIMS (**g**). **h** Heatmap of the dysregulated canonical pathways between CRC and HC in the three platforms depicted by IPA ingenuity pathway analysis. Red: Z_score > 0, activated; Blue: Z score <0, inhibited. **i** Receiver operating characteristic (ROC) curve for the Support Vector Machines (SVM)-based model to classify CRC vs. HC individuals when trained on E480 data. **j**–**l** Confusion matrix showed the model performance for classifying CRC vs. HC individuals. Source data are provided as a Source Data file.

and inter-experiment, as well as established a DIA scoring system to provide the relationship between metrics and identification/quantification results, assisting users in monitoring the LC-MS workflow performance, detecting potential problems, providing optimizing direction, detecting low-quality experiments, and facilitating the data quality control and experimental standardization with large cohort studies. This system could be applied not only to urinary proteomics analysis but also to large-scale data analysis of other samples (such as cell, tissue, blood, etc.).

The unified urinary proteome SOP was developed based on the MSCohort QC system and applied in multiple laboratories. Analysis results from 20 LC-MS platforms demonstrated the necessity of establishing the SOP. Meanwhile, results from 10 platforms without

SOP indicated that these metrics showed lower scores, including the identification rate of MS2 scans, the utilization rate of MS2 scans, peak counts of MS2, peak intensities of MS2, and chromatographic invalid acquiring time, etc. in experiments with fewer identification results (Supplementary Fig. 3), which also indicated that we should pay attention to above these metrics when conducting urinary proteomics experiments. In particular, the identification rate of MS2 scans and the utilization rate of MS2 scans in the DIA scoring formula play an important role in the evaluation and optimization of individual experiments.

We further applied the comprehensive QC system and unified SOP to the analysis of the complex mixture of digests from human urine, yeast, and *E. coli*, to investigate the ability to detect DEPs with the

quantitative accuracy, precision, and robustness of the different platforms. These results demonstrated that most of the DEPs could be recalled accurately in complex urine backgrounds with high quantitative accuracy and precision even in different types of LC-MS platforms, which would broadly increase the confidence in DIA-based urinary proteomics as a reproducible method for large cohort biomarker discovery research.

Moreover, the above workflow was applied to clinical large-cohort colorectal cancer (CRC) urinary proteome with more than 500 proteome experiments from three LC-MS platforms. More than 8000 proteins were reported from the three platforms. To the best of our knowledge, this study presented the deepest urinary proteome coverage, representing a promising basis for the discovery of biomarkers. Three different LC-MS platform analyses reported consistent quantitative precision and disease patterns. Interestingly, our data revealed complement systems were significantly activated in CRC patients. When combined with machine learning, the urinary proteome data achieved an AUC > 0.9 to classify CRC and HC. These results validated urinary proteomics as a valuable strategy for biomarker discovery and patient classification in CRC.

The demand for precision medicine is driving the need to increase throughput, improve consistency and accuracy, facilitate longitudinal research, and make data obtained across laboratories more comparable. Previous studies have demonstrated the reproducibility and quantitative performance of DIA proteomics with harmonized mass spectrometry instrument platforms and standardized data acquisition procedures in benchmark cell and tissue samples[15,16]. Our study expanded the technology to different types of mass spectrometers from different vendors, and higher complexity urine samples. The results showed that the highest reproducibility was achieved with the same LC and MS condition, which was consistent with the previous study[15] (Supplementary Fig. 4). Our study also found that different LC-MS platforms (Lumos, E480, and TIMS) also achieved high consistency under the same LC conditions and comprehensive QC system. Consistent quantitative accuracy and the ability to discover biomarkers were also validated in complex benchmarking samples and large-scale cohort clinical samples. These results highlighted the robustness of urinary proteomics under the combination of comprehensive QC system and unified SOP to support both basic discovery proteomics research and population-scale clinical sample analyses in a high-throughput manner. This work also increased the confidence that distributed urinary proteomics studies with hundreds to thousands of samples and data integration between labs are becoming feasible.

Recent advances in mass spectrometry hardware have provided a boost in the depth of standard analyses and enabled near-complete model proteome quantification in minimal measuring time[52,53]. Coupled with the development of data processing software, and the establishment of comprehensive quality control systems, urinary proteomics based on DIA technology is poised to mature further, showing potential for routine analysis of large clinical cohorts with the necessary depth and sample size to support clinical decision-making based on biomarker signatures.

## Methods

### Preparation and distribution of the quality control urine samples

First-morning urine (midstream) samples were collected from ten healthy individuals at Peking Union Medical College. Ten urine samples were combined and centrifuged at 3000 × g for 30 minutes at 4 °C to remove cell debris. The supernatant was transferred into the 2 mL EP tube (Corning, USA) and stored at −80 °C for further analysis.

The quality control (QC) urine samples were prepared at the Institute of Basic Medical Sciences Chinese Academy of Medical Sciences, School of Basic Medicine Peking Union Medical College, and then the urine peptide QC samples were distributed to 20 LC-MS platforms.

Urine peptides were prepared by the 96 DRA-Urine method by following the same steps as in the previous work[38]. Briefly, a total of 400 mL of pooled urine mixture was processed. Urine (2 mL each tube) was reduced with 20 mM dithiothreitol (DTT) for 5 min at 95 °C, and then alkylated with 50 mM iodoacetamide (IAM) at room temperature (RT) in the dark for 45 min, then urine proteins were pelleted using 6-fold volume precooled acetone, and centrifuged at 10,000 × g for 10 min at 4 °C. The protein precipitate was re-dissolved in 200 μL of 20 mM Tris(pH 8.0) and then combined. The concentration of pooled urine proteins was quantified using Pierce™ BCA protein assay kit (Thermo Fisher Scientific, USA) following the manufacturer's protocol. During protein precipitation and quantification, each well of the 96-well PVDF plate (MSIPS4510, Millipore, Billerica, MA) was prewetted with 150 μL of 70% ethanol and equilibrated with 300 μL of 20 mM Tris. For each well, one hundred micrograms of proteins were transferred to the 96-well PVDF plate. The samples were then washed three times with 200 μL of 20 mM Tris buffer (pH 8.0) and centrifugated. Proteins were digested by adding 30 μL of 20 mM Tris buffer (pH 8.0) with trypsin at a ratio of 50:1 (w:w) on the membrane. The samples were subjected to microwave-assisted protein enzymatic digestion twice in a water bath for 1 min under microwave irradiation[54] and then at 37 °C water bath for 2 h. Subsequently, the resulting peptides were collected by centrifugation at 3000 × g for 5 min. The eluted peptides were combined together, and purified with Sep-Pak C18 Vac Cartridge (Waters). The concentration of pooled peptides was determined by using Pierce™ Quantitative Colorimetric Peptide Assay kit (Thermo Scientific) following the manufacturer's protocol. Then peptides were aliquoted and lyophilized. Twenty micrograms of urinary peptides were resolved in 0.1 % formic acid (FA) to 1 μg/μL. Subsequently, eleven non-naturally occurring synthetic peptides from the iRT kit (Biognosys) were spiked into the sample at a ratio of 1:30 (v/v) to correct relative retention times between acquisitions. Finally, samples were shipped to the 20 LC-MS platforms.

### Preparation of Cell, *E. coli*, and *S. cerevisiae* samples

HEK 293 cells were grown in Dulbecco's modified eagle medium (DMEM) supplemented with 10% fetal bovine serum (FBS) and penicillin/streptomycin (1:1000) in 37 °C chamber supplied with 5% CO$_2$. Cells were lysed in buffer (50 mM Tris-HCl pH 8.0, 2 % sodium dodecyl sulfate (SDS), Protease Inhibitor) and sonicated for 5 minutes with power on 4 seconds and off 3 seconds at 25% energy. Proteins were pelleted by cold acetone, then resolved in 25 mM Tris-HCl buffer (pH 8.0), and concentration was determined by the bicinchoninic acid assay (BCA) kit (Thermo Scientific).

*Escherichia coli DH5a* was cultured in Luria Broth (LB) medium at 37 °C to mid-log phase, shaking at 240 rpm/min, in Luria Broth (LB). *S. cerevisiae CG1945* were grown at 30 °C to mid-log phase, shaking at 300 rpm/min in the yeast-peptone-dextrose (YPD) medium. Cells of *E. coli* and *S. cerevisiae* were harvested by centrifugation at 4000 × g for 5 min and washed twice with ice-cold phosphate buffered saline (PBS). Cell pellets were resuspended in lysis buffer (50 mM Tris-HCl pH 8.0, 2% SDS, Protease Inhibitor). Then sonicated for 10 minutes with power on 4 seconds and off 3 seconds at 25% energy. Proteins were pelleted by cold acetone, then resolved in 25 mM Tris-HCl buffer (pH 8.0) and concentration was determined by the bicinchoninic acid assay (BCA) kit (Thermo Scientific).

The HEK 293 cells, *Escherichia coli*, and *S. cerevisiae* proteins were delivered to digestion by Filter-aided sample preparation (FASP) method. In brief, 100 μg of cell lysates were reduced with 20 mM DTT at 95 °C for 5 minutes and alkylated with 50 mM IAM for 45 min at room temperature with dark. Protein solutions were loaded into the 10 kD ultracentrifugation tube equivalented with 25 mM NH$_4$HCO$_3$ buffer. Then proteins were digested with trypsin (Promega) with a 50:1 ratio (w/w) overnight at 37 °C in an ultracentrifugation tube. Peptides were desalted by SPE column (Waters), aliquoted, and dried by SpeedVac.

Peptides were resuspended at a concentration of 1 μg/μL with HPLC-grade water containing 0.1% (v/v) FA.

The pooled sample A was prepared by mixing human urine, *S. cerevisiae* (yeast), and *Escherichia coli* (*E. coli*) peptides at 65%, 15%, and 20% w/w, respectively. The pooled sample B was prepared by mixing human urine, yeast, and *E. coli* protein digests at 65%, 30%, and 5% w/w, respectively. The pooled sample A-H was prepared by mixing human (HEK 293), yeast (*S. cerevisiae*), and *E. coli* (*Escherichia coli*) peptides at 65%, 15%, and 20% w/w, respectively. The pooled sample B-H was prepared by mixing human HEK 293, yeast, and *E. coli* protein digests at 65%, 30%, and 5% w/w, respectively. The iRT kit (Biognosys) was added to each of the pooled samples at a ratio of 1:30 (v/v). For LC-MS analysis, 1 μg of pooled sample was adopted.

### Preparation of human colorectal cancer and healthy control samples

A total of 80 CRC patients (48 males and 32 females; median age 57 years, min-max: 42–69 years) were recruited from the Cancer Hospital, Chinese Academy of Medical Sciences. All patients were pathologically diagnosed by two senior pathologists, and first-morning midstream urine samples were collected before surgical operations or chemotherapy/radiotherapy. In addition, 80 urine samples from healthy control (HC) (52 males and 28 females; median age 55 years, min-max: 40–68 years) were obtained from the Health Medical Center of the Cancer Hospital. The enrollment criteria for HC subjects were as follows: (1) the absence of benign or malignant tumors; (2) a qualified physical examination finding no dysfunction of vital organs and (3) normal renal function and without albuminuria. Supplementary Data 6 lists the demographic and clinical characteristics of the 80 CRC patients and 80 HCs.

CRC and HC samples urine protein preparation and digestion were performed in the same way as the quality control urine samples by the 96 DRA-Urine method. In addition, each 96-well plate contains 3 Quality Assurance (QA) samples (pooled urine samples of equal protein amount from each sample) to monitor the reproducibility of the sample preparation. The resulting urine peptides from each sample were equally divided into triplicates for data acquisition on three LC-MS platforms, respectively.

### Data acquisition of urinary proteome from 20 LC-MS platforms

The 20 participant platforms used 3 different types of mass spectrometers, including Orbitrap (ThermoFisher Scientific), timsTOF (Bruker Daltonik), and ZenoTOF (SCIEX). To systematically analyze the variation among different LC-MS platforms and the main influencing factors, we divided 20 platforms into 2 groups. Ten of these platforms used the procedures and parameters they routinely used (number M01-M10), and the other 10 LC-MS platforms employed a unified LC condition (the same type of column and same gradient) and consistent MS parameters for the same type of instrument (number U01-U10). The detailed LC and MS acquisition parameters are provided in Supplementary Data 2, and the data acquisition was performed according to the SOP for urinary proteomics (Supplementary Note 3). All participant LC-MS platforms collected 1 pure iRT (Biognosys) data, 3 DDA data, 3 DIA data, and 1 blank to assess carry-over. The acquisition time was 30 min. The performance of DDA data acquired across multiplatform was investigated and provided in Supplementary Fig. 15.

### Data processing of urinary proteome from 20 LC-MS platforms

Data-independent acquisition data from the 20 LC-MS platforms dataset were processed with Spectronaut v.18.0 performing the directDIA analysis. All searches were performed against the human SwissProt database (Homo sapiens, 20386 reviewed entries, 2022_06 version), concatenated with iRT peptide.fasta file (downloaded from the Biognosys webpage). Briefly, the specific enzyme used Trypsin/P, peptide length from 7 to 52, max missed cleavages was set 2, toggle

N-terminal M turned on, Carbamidomethyl on C was set as fixed modification, and Oxidation on M as variable modification. The extraction of data used dynamic MS1 and MS2 mass tolerances, a dynamic window for extracted ion current extraction window, and a non-linear iRT calibration strategy. The identification was carried out using a kernel density estimator and Qvalue cut-off of 0.01 at precursor and protein levels. The top N (min:1; max:3) precursors per peptide and peptide per protein were used for quantification. Peptide intensity was calculated by the mean precursor intensity. Cross-run normalization was turned off. Additional DDA data collected under the same conditions as DIA data were added to create a hybrid library. The data processing results were exported using customized reports for further data analysis using MSCohort. The customized reports required in MSCohort were provided in Supplementary Data 1C. According to the different DIA methods of different LC-MS platforms, PG.MS2Quantity results were chosen for conventional DIA method, and PG.MS1Quantity results were chosen for HRMS1-DIA[16] methods for subsequent quantitative analysis.

### Data acquisition of pooled samples A and B

Equivalent amounts of pooled sample A, sample B, sample A-H, and sample B-H were shipped to three LC-MS platforms (U02-Lumos, U06-E480, and U08-timsPro 2). Samples were resuspended to final concentrations of 1 μg/μL in 0.1% FA with iRT and analyzed in three technical replicates using the DIA method provided in SOP on each LC-MS platform.

### Data processing of pooled samples A and B

Data-independent acquisition spectra in the 3 participant platforms dataset were analyzed with Spectronaut v.18.0 performing the directDIA analysis. All searches were performed against the Uniprot database for human (organism ID 9606, 20386 reviewed entries, 2022_06 version), yeast (organism ID 559292, 6727 entries, 2023_02 version), *E. coli* (organism ID 83333, 4634 entries, 2023_02 version) taxonomies, concatenated with iRT peptide.fasta file (downloaded from the Biognosys webpage). Default settings were used unless otherwise noted. Cross-run normalization was turned off. Carbamidomethyl on C was set as fixed modification, and Oxidation on M as variable modification.

### Data acquisition of CRC and HC proteomes

Equivalent amounts of urine peptides were shipped to three LC-MS platforms (U02-Lumos, U06-E480, and U08-timsPro 2). Samples were resuspended to final concentrations of 1 μg/μL in 0.1% FA with iRT and analyzed using the DIA method provided in SOP. QC samples were analyzed in triplicate before CRC and HC samples analyses and a single QC sample analysis was performed midway through the overall analysis. The acquisition of samples was randomized to avoid bias.

LC-MS analysis of generating dataset for three different LC-MS platforms was collected by Orbitrap Fusion Lumos coupled with an EASY-nLC 1000 system, an Orbitrap Exploris 480 mass spectrometer coupled with Vanquish Neo UHPLC system (Thermo Fisher Scientific), and a timsTOF Pro 2 mass spectrometer (Bruker) coupled with an UltiMate 3000 UHPLC system. All three LC-MS platforms were operated in DIA mode over a 30-minute total gradient. For three LC systems, peptides ware separated at a constant flow rate of 500 nL/min by the same type of analytical column (50 cm × 50 μm monolithic silica capillary column (Beijing Uritech Biotech)). LC mobile phases A and B were 100% $H_2O$ with 0.1% FA (v/v) and 80% ACN / 20% $H_2O$ with 0.1% FA (v/v), respectively. In 30 min experiments, the gradient of mobile phase B increased from 5% to 20% over 22 min and then increased to 30% over 3 min, a further 1 min plateau phase at 90% B, and a 4 min wash phase of 1% B.

Data acquisition on Orbitrap Fusion Lumos was performed in DIA mode using 80 variable windows covering a mass range of 350–1200 m/z. The resolution was set to 120,000 for MS1 and 30,000

for MS2. The Normalized AGC Target was 300% for MS1 and 200% for MS2, with a maximum injection time of 50 ms in MS1 and 50 ms in MS2. HCD Normalized Collision Energies was set to 32%.

Data acquisition on Orbitrap Exploris 480 was performed in DIA mode using 80 variable windows covering a mass range of 350–1200 m/z. The resolution was set to 120,000 for MS1 and 30,000 for MS2. The Normalized AGC Target was 300% for MS1 and 200% for MS2, with a maximum injection time of 50 ms in MS1 and 50 ms in MS2. HCD Normalized Collision Energies was set to 30%.

Data acquisition on timsTOF Pro 2 was performed in diaPASEF mode using 50 windows. The MS spectra were acquired from 100 to 1700 m/z. The ion mobility was scanned from 0.75 to 1.3 Vs/cm². The ramp time was set to 100 ms. The collision energy was ramped linearly as a function of the mobility from 59 eV at $1/K0 = 1.6$ Vs/cm² to 20 eV at $1/K0 = 0.6$ Vs/cm². Isolation windows of a 16 m/z width were set to cover the mass range of 350 to 1200 m/z in diaPASEF.

### Data processing and analysis of CRC and HC proteomes

The DIA data from 3 LC-MS platforms were analyzed separately with Spectronaut v.18.0 performing the directDIA analysis. Default settings were used unless otherwise noted. Cross-run normalization was turned off. Carbamidomethyl on C was set as fixed modification, and Oxidation on M as variable modification. All searches were performed against the human SwissProt database (Homo sapiens, 20,386 reviewed entries, 2022_06 version), concatenated with iRT peptide.fasta file.

The data processing results were exported using customized reports for further data analysis using MSCohort. The low-quality data analyzed by MSCohort was excluded. Then log2 transform and directLFQ[55] normalization were performed for all samples. Proteins with missing values < 50% of the samples in each group were retained for further analysis. Missing values were imputed based on the sequential k-nearest neighbor (Seq-KNN) method using NAguideR[56].

### Statistical analyses

Differentially expressed proteins analysis was performed using the LIMMA[57] package (version 3.58) in R (version 4.3) with the expectation that proteins significantly altered between CRC and HC exhibited Benjamini & Hochberg-adjusted $p < 0.05$. Pathway analysis of protein alterations was performed using Ingenuity Pathway Analysis (Qiagen). The correlation, t-SNE, and heatmap plots were performed using Corrplot (version 0.92), Rtsne (version 0.16), and ComplexHeatmap (version 2.16.0) packages in R (version 4.3). Pattern recognition analysis (OPLS-DA) was performed using SIMCA 14.0 (Umetrics, Sweden) software. Six machine learning models (Logistic Regression, K-Nearest Neighbor, Gaussian Naïve Bayes, Support Vector Machines, Random Forest, Gradient Boosting Decision Tree) were performed using scikit-learn[58] modules (version 0.23) in Python (version 3.7). The UniProtKB/Swiss-Prot public database was used to map the gene names of DEPs, and enrich GO terms and KEGG pathways was performed using clusterProfiler[59] (v.4.8.3) in R (version 4.3). Protein–protein interaction (PPI) plot was performed using STRING (v.11.0) and Cytoscape (v.3.7.2).

### MSCohort system

The workflow of MSCohort consists of two modules: Intra-experiment analysis and inter-experiment analysis.

(1) **Intra-experiment analysis** enables the systematic evaluation and optimization of individual experiment. We have developed the quality control software tool MSRefine to evaluate and optimize the performance of individual DDA experiment in previous study[35]. MSCohort integrated the metrics and function of MSRefine, and established a quality control system for individual DIA experiment. Here, we mainly focus on introducing the metrics and steps for evaluation and optimization of individual DIA experiment in MSCohort. The QC metrics for DIA experiments are divided into six categories: sample,

chromatography, DIA windows, ion source, MS1 and MS2 signal, and identification result. We established a metric-score system for individual experiment quality evaluation and data optimization (Supplementary Fig. 1a). A detailed description of the metrics is provided in Supplementary Data 1A. The workflow for intra-experiment analysis consists of three steps: reading.raw/.d/.wiff files and identification/quantitation results, extracting metrics and scoring, and generating a visual report.

**Step 1: Reading the.raw/.d/.wiff files and identification/quantitation results.** The.raw/.d/.wiff files can be converted to.ms1/.ms2 files using pXtract, timsTOFExtract, and wiffExtract, respectively. For processing of timsTOF data, timsTOFExtract used TimsPy[60] to convert the proprietary format (Bruker Tims data format (TDF)) to the textual.ms1/.ms2 files. All three in-house tools were embedded in MSCohort. Besides, the identification and quantification results were extracted from Spectronaut[34], and the customized reports required in MSCohort were provided in Supplementary Data 1C.

**Step 2: Extracting metrics and scoring. This module carries out two tasks:** (1) extraction of metrics and (2) calculation of the first-level and second-level scores.

To represent the experimental conditions of DIA with a mathematical model, we designed a quality scoring system for DIA data based on our previous DDA data quality scoring system[35].

The DDA scoring formula is expressed as:

$$N_{identified\,precursors} = N_{acquired\,MS2} \times Q_{MS2} \times P_{MS2\,per\,precursor} \quad (2)$$

Where $N_{identified\_precursors}$ is the number of identified peptide precursors, $N_{acquired\_MS2}$ is the number of acquired MS2 scans, $Q_{MS2}$ is the identification rate of the MS2 scans (the number of identified MS2 scans/ the number of acquired MS2 scans), and $P_{MS2\_per\_precursor}$ is the utilization rate of the MS2 scans (the number of unique peptide precursors/ the number of identified MS2 scans).

The DIA scoring formula is expressed as:

$$N_{identified\_precursors} = N_{acquired\_MS2} \times Q_{MS2} \times (N_{precursor\_per\_MS2} / R_{precursor})$$
$$(3)$$

where $N_{identified\_precursors}$ is the number of identified peptide precursors, $N_{acquired\_MS2}$ is the number of acquired MS2 scans, $Q_{MS2}$ is the identification rate of the MS2 scans (the number of identified MS2 scans/ the number of acquired MS2 scans), $N_{precursor\_per\_MS2}$ is the spectra complexity of MS2 scans (the number of redundant identified precursors/ the number of identified MS2 scans), $R_{precursor}$ is the precursors duplicate identification rate (the number of redundant identified precursors/ the number of identified precursors), and $(N_{precursor\_per\_MS2} / R_{precursor})$ is the utilization rate of the MS2 scans (the number of unique peptide precursors/ number of identified MS2 scans).

This DIA scoring formula was designed based on the DDA scoring formula, the utilization rate of the MS2 scans was divided into the spectra complexity of MS2 scans and the precursors duplicate identification rate. Since the DIA method was to fragment all the parent ions in the isolation window to obtain a mixture MS2 spectrum, theoretically an MS2 scan can be identified to multiple precursors. Therefore, we established a spectra complexity index to represent the number of precursors that can be identified by an average MS2 spectrum/scan. The spectra complexity depends on the DIA windows number and the window size. For example, in Supplementary Note 3, when we set 80 MS2 windows per cycle, the average window size was 6 Da, the

spectra complexity (Redundant identified precursors/ Identified scan rate) was 2.27, and the precursors duplicate identification rate (Redundant identified precursors/ Identified precursors rate) was 1.58. Therefore, the utilization rate of the MS2 scans of 80 MS2 windows was 1.44 (2.27/1.58 = 1.44); When the number of MS2 windows was set to 22 windows per cycle, the average window size was 26 Da, and the spectra complexity was increases to 4.1. However, the corresponding precursors duplicate identification rate was 4.07, so the utilization rate of MS2 scans was 1.01 (4.1/4.07 = 1.01). Therefore, balancing spectra complexity and precursors duplicate identification rate was the key to improve the utilization rate of MS2 scans. Formula (3) was a naive formula but can be extended and used to characterize different kinds of DIA experiments.

**Step 2.1 Extraction of metrics**. MSCohort read.ms1/.ms2 files, extracting and calculating MS1- or MS2-related metrics, including cycle time, ion injection time, peaks intensity, peak counts, and scans number of MS1 or MS2. At the same time, MSCohort extracted peptide features[61] from MS1 scans, analyzing how many features are detectable by high-resolution MS and how many of them are identified by search engine. After data processing, MSCohort obtained more metrics according to the Spectronaut report, including the number of identified precursors, peptides, and proteins. In addition, the data processing results also included missed cleavages of peptides, peak width, FWHM, mass accuracy, precursors intensity, and protein group intensity-related metrics. MSCohort also calculated the spectra complexity of MS2 scans and the precursors duplicate identification rate to analyze whether the MS2 scans were fully utilized for peptide precursors. We provided a detailed introduction to all the metrics proposed by MSCohort in Supplementary Data 1A, including their specific meanings and extraction processes.

**Step 2.2 Calculation of the first-level and second-level scores**.

It is not convenient for users to comprehensively evaluate the data if only display the values of each metric. Therefore, we assigned scores to each metric and established a first-level and second-level scoring system (Supplementary Fig. 1a, 1b). MSCohort computes a quality score for each of the QC metrics using a score function (see below), and these QC metrics quality scores were defined as second-level scores. These five categories in Formula (3) were defined as first-level scores, including the number of identified peptide precursors, the number of acquired MS2 scans, the identification rate of MS2 scans, the spectra complexity of the MS2 scans, and the precursors duplicate identification rate.

Each metric was scored on a scale of one to five, with "5 points" being excellent and "1 point" indicating plenty of room for improvement. Taking the metric M12. Median of MS1 raw mass accuracy (The median of delta mass between the monoisotopic theoretical and the measured m/z of precursors) as an example, for Thermo Orbitrap instrument, the median of MS1 raw mass accuracy ≤1 ppm was 5 points. The median of MS1 raw mass accuracy ≥ 5 ppm was 1 point. For the median of MS1 raw mass accuracy between > 1 ppm and <5 ppm, a linear scoring algorithm is applied.

Due to the diversity of experimental methods and instruments, scoring standards are often not fixed and uniform in practice. In this study, the scoring standards for each metric were set based on the data of urinary proteomic optimization experiments collected under different parameter conditions and the data of urine QC samples collected on 20 platforms. Users can adjust scoring define standards according to the actual situation. After determining the scoring standards, each of the metrics is assigned an individual score. We provided the

detailed instructions for users to adjust the scoring standards in the user manual of MSCohort (https://github.com/BUAA-LiuLab/MSCohort). As shown in Supplementary Fig. 1a, different scores are represented in different colors. Users can directly determine which metric is low-performance based on the colors.

The QC metrics in the DIA LC – MS workflow would affect the first-level scores. For example, high missed cleavage, long chromatographic invalid acquiring time, and lower peaks intensity of MS2 would result in low MS2 identification rate. The metrics that affect first-level scoring are measured by second-level scores. MSCohort calculates the second-level scores, then averages them to obtain the corresponding first-level scores, and subsequently applies a similar process to calculate the total score. The final scoring results will be visually presented in various forms.

**Step 3: Generating a visual report**.

MSCohort generates a report with comprehensive tables and charts. We use suitable graphs or tables, such as graphs showing the accumulated number of MS2 scans or precursors, to give a global view of the performance of the LC-MS workflow. More details are complemented by various graphs, such as a histogram showing the peptide eluting width, a statistical bar graph showing the number of precursors identified by one MS2 scan, or a scatter graph showing the peak counts of each MS1 or MS2 Scan. The outputs are also exported to simple tab-delimited text files, so visualization or analysis can also be performed using external tools or code scripts.

(2)  **Inter-experiment analysis**. This module incorporates an additional 23 inter-experiment metrics for inter-experiment comparisons and low-quality experiments detection. The workflow for inter-experiment analysis consists of three steps: reading intra-experiment data and identification/quantitation results, extracting inter-experiment metrics and scoring, and generating a visual report.

**Step 1: Reading intra-experiment analysis results**.

For the submitted cohort data, MSCohort first conducted the intra-experiment analysis on each original data to generate the intra-experiment metrics value and score result for each data. The scores are collated to create an overview chart (heatmap) that displays metric scores per Raw file for a comprehensive overview of the whole cohort.

**Step 2: Extracting inter-experiment metrics and scoring**. This module carries out two tasks: extraction of metrics and calculation of quality scores.

**Step 2.1 Extraction of metrics**

MSCohort inter-experiment metrics can be assigned to five categories (Sample Preparation, Liquid Chromatography, and Precursor-level, Peptide-level, and Protein group-level Quantification Results) according to the experimental workflow and quantification results. We provided a detailed description to all the inter-experiment metrics proposed by MSCohort in Supplementary Data 1B. The key metrics are listed below.

**Metrics SP1-SP3: Customizable contaminant search**
Pre-analytical variation caused by contaminations during sample collection or inconsistent sample processing can have an impact on the results and may cause the reporting of incorrect biomarkers[62]. MSCohort offers configurable lists of custom protein contaminants to help users assess each sample for potential quality issues.

For urine sample quality control, we used three urine-specific quality marker panels to assess the degree of contamination of the samples. Firstly, we used two previously reported quality marker panels to determine the degree of contamination with erythrocytes[62,63] and cellular debris[3]. Contamination of erythrocytes occurs during urine collection due to hematuria or hemolysis caused by kidney

function issues or systemic disorders, leading to a high sample-to-sample variability compared to regularly secreted urinary proteins[62,63]. Insufficient removal of cells and cellular debris from urine will lead to increased detection of intracellular proteins with a high sample-to-sample variability compared to regularly secreted urinary proteins[64]. In addition, proteinuria occurs due to abnormalities in kidney function or systemic disorders, resulting in the leakage of serum proteins into the urine. This can lead to increased detection of serum proteins with high sample-to-sample variability[65]. We generate the third urine-specific quality marker panel to asses of contamination with serum high abundant protein[11].

MSCohort reports the proportion of the summed contaminant protein intensity/ the sum intensity of all proteins for each sample. For each metric, we initially defined potentially contaminated samples as those with a value more than two standard deviations above the median.

### Metrics LC1 and LC2: Retention time deviation
The retention time (RT) of each analyte in MS data usually has shifts for multiple reasons, including matrix effects and instrument performances, especially for large cohort studies[66]. MSCohort extracts the retention time (RT) and deltaRT of precursors from the Spectronaut report. MSCohort also calculates the mean-square error (MSE) between any two LC-MS experiments.

$$MSE = \frac{1}{n}\sum_{i=1}^{n}(Y_i - Y_j)^2 \qquad (4)$$

Where n is the number of LC-MS experiments, Y is the RT array of the retention times of the same precursors identified in both two experiments. MSE is calculated for precursors RT between all LC-MS experiments ($i = 1,...,n; j = 1,...,n$).

### Metrics MS1 - MS18: Precursor-level, Peptide-level, and Protein group-level quantification results
Six statistical metrics were chosen to describe the distribution of precursor/ peptide/ protein abundance across all experiments. These metrics include the number of identifications, the median intensity distribution, the interquartile range (IQR), robust standard deviation[67], Pearson correlation, and the normalization factor of the intensity at the precursor, peptide, and protein levels. Currently, MSCohort offers three commonly used normalization algorithms, directLFQ[55], maxLFQ[68], and quantile[69,70].

#### Step 2.2 Calculation of cohort quality scores
For each inter-experiment metric, quality control scores are calculated based on statistical assessment of the median and standard deviations for all experiments. We exploit the assumption that the majority of the proteome typically does not change between any two conditions so that the median behavior could be used as a relative standard[68]. The values of more than two standard deviations (SD) from its median indicating heterogeneity with other experiments.

We initially defined potentially low-quality data as those with a value more than two standard deviations from the median. Each score is scored on a scale of one to five, with "5 points" when the parameter is close to or above the overall median and "1 point" when the parameter is more than two standard deviations (SD) from its median indicating plenty of room for improvement. This setup enables automated non-subjective inter-experiment or instrument performance evaluation. Taking the median of protein intensity as an example, the median of protein intensity close to or above the overall median is 5 points, less than the overall median minus two standard deviations is 1 point. Due to the diversity of experimental conditions and sample type, scoring standards are often not fixed and uniform in practice. Users can adjust scoring and define standards according to the actual situation.

### Step 2.3 Identify outlier LC-MS experiment(s) using the isolation forest algorithm
Previous studies have shown that the LC-MS experimental process is complex, with numerous factors influencing LC-MS data, and these factors are not independent but may affect each other. Therefore, for high-dimensional and complex LC-MS data, supervised classifiers heavily rely on training data. Data from different instruments, laboratories, and sample types require re-labeling and retraining, leading to poor generalization. Consequently, unsupervised machine learning algorithms are commonly used for outlier data analysis[30,71].

Here, we applied an excellent unsupervised and online outlier detection algorithm, Isolation forest (iForest), to distinguish outlier experiments. iForest achieves outstanding success in most scenarios by taking advantage of the anomalous nature of "few and different"[72]. It has unique advantages in dealing with large datasets due to its low-computational complexity[73,74].

As mentioned in the original iForest paper, the unsupervised and online outlier detection algorithm is a two-stage process. The first (training) stage builds isolation trees using sub-samples of the training set. The second (testing) stage passes the test instances through isolation trees to obtain an anomaly score for each instance. This algorithm does not require a labeled dataset or pre-training of offline models, it can dynamically construct isolation trees online for any batch of data.

First, the 23 inter-experiment metrics value for each experiment in the cohort were integrated into a two-dimensional matrix. Then, the outlier experiments detection were performed using iForest algorithm with two-stage process:

(1) Training stage: iForest randomly selects subsamples from the cohort, then a feature (metric) is randomly selected, and a separation value is randomly generated within the selected feature value range to "isolate" the sample point. Then iForest recursively selects different features and values from the child subset to split the child into smaller subsamples. iTrees are constructed by recursively partitioning the given training set until instances (samples) are isolated or a specific tree height is reached of which results a partial model. Many iTrees will make up the iForest. Thus, we can get the average path length of all iTrees in the iForest.

(2) Evaluating stage: iForest passes the samples through isolation trees to obtain an anomaly score for each sample. Outliers are those samples which have short average path lengths on the iTrees and low anomaly score. The iForest was implemented using scikit-learn[58] python library (version 0.23) module sklearn.ensemble.IsolationForest with default parameters.

### Step 3: Generating a visual report
MSCohort generates a report with comprehensive tables and charts. The scores are collated to create an overview heatmap that displays the metrics scores per experiment for a compressed overview of the whole cohort. The user can subsequently follow up on detailed quality metric plots of interest in the remainder of the report. In summary, quality control metrics offer a visual guide to users to judge the data quality, whereas scores computed from the underlying data represent a mathematically more rigid way to automatically flag data sets as failed or successful. In addition, the underlying metrics values and scores are automatically exported to a text file and can be readily used for manual comparison and annotation of data sets.

### Reporting summary
Further information on research design is available in the Nature Portfolio Reporting Summary linked to this article.

## Data availability

The mass spectrometry proteomics data, search results, and detailed analysis report from MSCohort have been deposited to the ProteomeXchange Consortium via the iProX partner repository[75,76] with the dataset identifier PXD050291 and IPX0008194000. Source data are provided with this paper.

## Code availability

MSCohort is developed in Python and is freely available. The latest software version and the user manual can be downloaded from GitHub (https://github.com/BUAA-LiuLab/MSCohort).

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

## Acknowledgements

This work was supported by grants from the National Key Research and Development Program of China (2021YFA1301602,2021YFA1301603, 2024YFA1307201 to C.L.), the National Natural Science Foundation of China (32171442 and 92474115 to C.L., 82170524 and 31901039 to W.S.), the Fundamental Research Funds for Central Universities, Beijing Municipal Public Welfare Development and Reform Pilot Project for Medical Research Institutes (JYY2018-7), CAMS Innovation Fund for Medical Sciences (2021-I2M-1-016, 2022-I2M-1-020), Beijing Natural Science Foundation-Daxing Innovation Joint Fund (L246002) and Biologic Medicine Information Center of China, National Scientific Data Sharing Platform for Population and Health.

## Author contributions

X.L. contributed to the experiment design, MSCohort system development, data analysis, figures preparation, wrote the manuscript, and revised the manuscript. H.S. contributed to MS-based proteomics data generation and revised the manuscript. X.H. contributed to the MSCohort system development. J.S. contributed to the colorectal cancer proteomics experiments. M.T. contributed to the MSCohort system development. Yong-Biao Zhang and Yongqian Zhang contributed to revise the manuscript. Y.G., S.T., Z.S., K.L., L.J., Jing Wei, Jianqiang Wu, X.T., Y.L., G.W., X.S., L.Z., H.Y. Xinxin Liu, D.L., Q.Z., X.Q., G.W., M.H., Y.T., M.T., P.X., L.G., Q.Z., Y.C., J.J., W.H., W.Z., M.H., Y.Q., Xianming Liu, X.D., J.L., L.C., Y.Z. contributed to urinary proteomics data across 20 LC-MS platforms. W.S. and C.L. supervised and guided the project, interpreted MS-based urinary proteomics data and revised the manuscript. All authors reviewed and approved the manuscript.

## Competing interests

The authors declare no competing interests.

## Ethics

This study was approved by the Ethics Committee of the Institute of Basic Medical Sciences, Chinese Academy of Medical Sciences (#047-2019) with an exemption of informed consent and was performed according to the Declaration of Helsinki Principles.

## Additional information

## Urine Test Sample Working Group

Youhe Gao[4], Shuxuan Tang[4], Ziyun Shen[4], Kehui Liu[5], Lulu Jia[6], Jing Wei[6], Jianqiang Wu[7], Xiaoyue Tang[7], Yanchang Li[8], Guibin Wang[8], Xinying Sui[8], Lihua Zhang[9], Huiming Yuan[9], Xinxin Liu[9], Dong Liu[10], Qi Zhang[10], Xindan Qiu[10], Guanbo Wang[11,12], Mo Hu[11], Ye Tian[11], Minjie Tan[12], Peng Xue[13], Liman Guo[14], Qing Zhang[14], Yongsheng Chen[13,15], Jianguo Ji[16], Weiyi Hu[16], Wenyuan Zhu[16], Min Huang[17], Yingzi Qi[17], Xianming Liu[18], Xiaoxian Du[18], Ji Luo[19], Lingsheng Chen[19] & Yinghua Zhao[19]

[4]Gene Engineering Drug and Biotechnology Beijing Key Laboratory, College of Life Sciences, Beijing Normal University, Beijing, China. [5]State Key Laboratory of Membrane Biology, Institute of Zoology, Chinese Academy of Sciences, Beijing, China. [6]Department of Pharmacy, Clinical Research Center, Beijing Children's Hospital, Capital Medical University, National Center for Children's Health, Beijing, China. [7]Institute of Clinical Medicine, Center for Biomarker Discovery and Validation, National Infrastructure for Translational Medicine, Peking Union Medical College Hospital, Chinese Academy of Medical Sciences and Peking Union Medical College, Beijing, China. [8]State Key Laboratory of Medical Proteomics, Beijing Proteome Research Center, National Center for Protein Sciences (Beijing), Institute of Lifeomics, Beijing, China. [9]State Key Laboratory of Medical Proteomics, National Chromatographic R. & A. Center, CAS Key Laboratory of Separation Science for Analytical Chemistry, Dalian Institute of Chemical Physics, Chinese Academy of Sciences, Dalian, China. [10]Mass-spectrum Core at the National Center for Protein Sciences at Peking University, Beijing, China. [11]Changping Laboratory, Beijing, China. [12]Institute of Chemical Biology, Shenzhen Bay Laboratory, Shenzhen, China. [13]Guangzhou National Laboratory, Guangzhou, China. [14]Proteomics and Metabolomics Core Facility, Guangzhou National Laboratory, Guangzhou, China. [15]Graduate School of Guangzhou Medical University, Guangzhou, China. [16]State Key Laboratory of Protein and Plant Gene Research, School of Life Sciences, Department of Biochemistry and Molecular Biology, School of Life Sciences, Peking University, Beijing, China. [17]Thermo Fisher Scientific, Shanghai, China. [18]Bruker (Beijing) Scientific Technology Co. Ltd, Beijing, China. [19]SCIEX China, Shanghai, China.

