## [Transparent Peer Review File · Nature Communications]

Standard operating procedure combined with comprehensive quality control system for multiple LC-MS platforms urinary proteomics

Corresponding Author: Professor Chao Liu

Version 0:

Reviewer comments:

Reviewer #1

(Remarks to the Author)

QC is crucial for large scale proteomics experiments as the throughput increases rapidly. Urine is also very important specimens for biomarker discovery. This manuscript certainly touched the frontier of the field, However, it is unclear to the reviewer that what is the focus of this manuscript. They mentioned the following aspects: 1) a SOP for generating urine MS-based proteome data using urine protein digests; 2) a QC software system called MSCohort. 3) Consistency of proteome analysis using different MS instruments (Lumos, E480, TIMS), including an application of CRC cohort. The reviewer thinks the 2nd point (MSCohort) is probably the major novelty of this manuscript since the major advantage of the SOP compared to the existing ones is the MSCohort. Comparative proteome analyses across different MS instruments have been published too.

In this case, the reviewer would expect to see more technical details of MSCohort than that provided here. From the manuscript as it is, it is difficult to evaluate the novelty of this QC tool. Several key issues should be clarified:

- 1) explain more about the 70 metrics. Are they all existing ones used by the literature? Any new metrics developed here? What's the difference between MSCohort compared to others in the literature?
- 2) Are these metrics for DDA or DIA? DDA and DIA are different methods. Throughout the manuscript, it is frequently unclear whether the authors performed DDA or DIA QC runs, and how they analyzed the data.
- 3) It seems the 70 metrics are divided into 47 metrics over the entire LC-MS workflow for DIA + 23 inter-experiment metrics. If so, the authors need to explain this much clearer. Are they applicable to DDA or SRM/PRM?
- 4) Line 148: how did the authors justify this equation? What is the "corresponding scores" in line 153? Related to this: line 713-743: more details of this equation is explained, but it is confusing since this was used for DDA. It is unclear why the DDA equation is applicable to DIA data.
- 5) line 156: how to define "large cohort proteome data"? When should one use MSCohort?
- 6) line 168: please provide more details about "unsupervised machine learning algorithm". How did you justify this machine learning model? How was it trained, validated and tested?
- 7) As a user, can you provide detailed protocols as how to download, and use this software?

So the reviewer suggests the authors to largely expand this MSCohort section, if this is the core of this manuscript. In particular, stress the novelty of the software compared to the literature.

More major issues in the rest sections to be clarified:

- 8) In the section starting from line 174 "comprehensive and comparative....": it is unclear what is the benefit of MSCohort. LC-

SOP samples exhibited more reproducible results than that without LC-SOP, because the latter have different LC systems. Can MSCohort rescue reproducibility for samples without LC-SOP, and further enhance reproducibility for samples with LC-SOP? How can one use MSCohort? This is a demo, so the author should explicitly demonstrate how to use MSCohort, and what is the added value.

9) line 251+: it is not a surprise that DDA is less reproducible than DIA. Again, the authors should explicitly show the added value of MSCohort.

10) in the section starting from line 262: again, what really matters here is the added value of MSCohort, unless the authors aimed to only show here that MSCohort works in three types of MS instruments.

11) line 294+: It is unclear how the DEPs analysis contributes to the added value of MSCohort.

12) line 318: "large cohort" -- a total of 160 samples may not well justify a large cohort.

13) in the last section of the CRC urine, the authors are commended for the comprehensive analysis, but it is unclear to the readers how MSCohort was applied, and what are the added value of MSCohort.

Some other big picture issues:

14) is MSCohort specific to analyzing urine samples? Has it been optimized for urine samples? It is understood that most of the samples used in this study are from urine. But are there any specific QC metrics or algorithm customized for urine proteome?

Minor issues:

15) In supplementary Figure 1, "median total ion current (TIC)" is mentioned, whereas supplementary table 1 lists "median total ion chromatography (TIC)". Correct one should be used.

16) Descriptions of the 23 inter-experiment features in the appendix are confusing.

17) Tables in the appendix are inconsistently formatted with colons and periods, requiring standardization.

(Remarks on code availability)

Reviewer #2

(Remarks to the Author)

As the authors describe, there is a significant need to implement the standardization of large-scale urinary proteomic analyses, particularly those with clinical relevance. To systematically assess data quality from large-scale urinary proteomics datasets, Liu et al. developed a quality control (QC) system named MSCohort, which evaluates 70 metrics. A total of 20 LC-MS platforms were evaluated using pooled urine samples; standard mixtures of *E. coli*, yeast, and HEK293 lysate; and urine from healthy controls and patients with colorectal cancer. The evaluated MS systems include Orbitrap, TimsTOF, ZenoTOF, Orbitrap Fusion Lumos, and Exploris 480. The evaluated LC systems included EASY-nLC 1000, Vanquish Neo UHPLC, and Ultimate 3000 UHPLC.

The current study falls short of its stated goals in three main areas: 1) the lack of clarity regarding the establishment of the acceptability criteria for the 70 QC metrics included in MSCohort, 2) a less-than optimal experimental design wherein the same LC system was not evaluated in combination with each MS system, and 3) the lack of an explanation of the scores assigned to the QC metrics included in MSCohort, which are presented on a scale from 1 – 5 with "5 points being excellent and 1 point indicating plenty of room for improvement". Consequently, this reviewer is unable to recommend the manuscript for publication in its current state. Several issues should be addressed:

1. One of the paper's main premises is that the use of SOPs improves the data reproducibility and overall quality. This is as to be expected and is not a new observation originating from the current study that warrants any significant explanation.
2. Lines 82-84: "Increasing numbers of urinary proteomic studies have been used to discover potential disease biomarkers, such as urological cancers, colorectal cancer, virus infection, neurodegenerative disorders, and many others." This sentence should be re-worded to avoid the current, incorrect implication that "urological cancers, colorectal cancer", etc. are biomarkers.
3. Lines 182 – 187 and Fig. 1: It does not appear that the same 10 LC-MS platforms were used with and without an LC SOP; therefore it is not accurate to make conclusions regarding the role of the LC SOP in contributing to the data quality.
4. Lines 196 – 197: This is the only place where FAIMS is mentioned throughout the entire manuscript. It is unclear whether the instrument performance comparisons were conducted separately for the platforms with and without FAIMS. One would expect the use of this ion separation method to significantly influence several of the MS-specific metrics that are included within MSCohort.
5. Lines 228 – 237 and other locations throughout the manuscript where protein identification data is mentioned: The authors do not provide any critical insight or interpretation of the properties of the proteins that were identified across all LC-MS platforms. For example, it would be informative to know the average sequence coverage of these proteins, in addition to their relative abundance. What was the dynamic range of these proteins?
6. For the instances when any of the QC metrics evaluated by MSCohort are not within the stated acceptability limits/scores, the authors do not provide any guidance regarding whether these scores are indicative of random or systematic

errors/variation.

7. Line 437: "...we should pay attention to these metrics when conducting urine proteomics experiments". Such language is overly vague and does not provide any constructive guidance for readers.

8. Line 162, 336, 810 & 814: The description of "contaminants" as a components of the "whole LC-MS workflow evaluation" in MSCohort should be clarified. Erythrocyte contaminants are mentioned in Line 814 in the context of erythrocyte lysis that occurs during the preparation of plasma; however, it is unclear how these type of contaminants would be relevant to the evaluation of other sample types.

9. Considering the potential broader/"real world" applicability of MSCohort, what is the minimum number of LC-MS data files that can be analyzed by MSCohort given that intra- and inter-experiment parameters are evaluated? For the determination of intra-experiment metrics, it would seem that technical replicates are required.

10. Figure 2: The "Hours" label on the far left side of the figure does not provide any useful information.

11. Figure 3: E240 is listed as one of the LC-MS platforms. Presumably, this refers to the Exploris 240; however, this instrument is only mentioned in one other location in the manuscript – line 208: "M09-E240" – again, presuming that E240 refers to the Exploris 240.

12. Figure 4 title: "sample-A and sample-B". These names are too vague and should be explained in more detail.

13. Figure 4 legend: Instead of describing the lengths of the whiskers on the box plots as "1.5 box lengths away from the box ends", it would be more appropriate to describe the length of the whiskers as 1.5-times the interquartile range.

(Remarks on code availability)

Reviewer #3

(Remarks to the Author)

(Remarks on code availability)

Reviewer #4

(Remarks to the Author)

The manuscript introduces a comprehensive quality control (QC) metric system for evaluating the performance and validity of large-scale MS-based proteomics analysis of human urine samples. The QC metrics were applied to human urine proteomics data from different platforms to illustrate the impact of implementing the QC metric system. The experimental design and data provided for three MS platforms do not align with the objective of the manuscript to provide a QC metric and reference standard operating procedure for a "large-scale cohort" for urine proteomics. The manuscript also proposes "a comprehensive QC system for systematic and measurable quality assessment in large cohort LC-MS analysis over time and across instruments" (pg 6 lines 171-173); however, the experimental design or data do not include information about a time course study (short- or long-term). The manuscript suggests that the QC metrics provide "optimizing strategies" (pg 4, lines 107-108, pg 5, lines 134-135, 141-142); however, the author does not demonstrate how the optimization strategies for experimental workflows are provided in the QC metric output without relying on "expert expertise to optimize parameters and locate experimental problems" (pg 4 lines 102-104). The discrepancy between the study objective and the presented data/results affects the significance of the manuscript. I do not recommend this manuscript for publication in its current state unless major revisions are made to the study objective, data presentation, format, and manuscript content. The manuscript and supplemental materials contain a substantial amount of data to demonstrate the performance of the QC metrics. Below are additional concerns with the manuscript:

1. Pg. 4 lines 88-89 – sentence structure issue
2. Reference not included for DIA data studies - Pg. 4 lines 89-92
3. Pg. 4 line 94 – change "body" to "bodily"
4. Pg. 4 line 99 – add "a" before SOP
5. Pg. 4 lines 102-104: The experimental design and rationale are comparable to the work published by Rudick PA et al. *Mol Cell Proteomics* 9, 225-41 (2010), as referenced in the manuscript. The application of LC-MS/MS QC metrics for the optimization of a proteomics workflow proposed in the manuscript is similar to work published by Beasley-Green A, et al. *Proteomics* 12, 923-31 (2012); however, this publication is not referenced in the manuscript.
6. Pg. 7 lines 182-184: How was the "unified LC and MS" or reference SOP optimized for the study?
7. Figures 3a and 3e should be combined, and the uncertainty (error bars) of the protein identification results for each platform should be included.
8. The color scheme in Figure 3 should be maintained for each study condition (with LC-SOP-blue and without LC-SOP-orange) for clarity.
9. The y- and x-axis labels for the figures should provide clear and concise information about the data presented. For example, the y-axis labels for Fig. 3b and 3f ("intersection size" for vertical bars and "set size" for horizontal bars) do not align with the figure text. The text states that the vertical bars represent the proteins identified across the platforms (or inter-/between-platform data) and the horizontal bars represent the proteins identified in each platform (or intra-/within-platform data).

10. The discussion section states that "more than 500 proteome experiments" (pg 14 lines 447-449) were performed; however, this statement does not align with the experimental design in the methods section or the supplemental sections.
11. In the Methods section (pg 16 lines 494-510), the author states that the peptide concentration is determined via colorimetric assay. How is the protein concentration determined for the accurate enzyme-to-protein ratio for the digestion protocol? What is the concentration of peptide material aliquoted prior to lyophilization (line 505)?

(Remarks on code availability)

Version 1:

Reviewer comments:

Reviewer #1

(Remarks to the Author)

Thanks for the revisions. A few more comments emerged from the revision.

1. In the clinical cohort application, the authors mentioned that the software detected several issues. However, in Supplementary Figures 9b and 9c, I noticed that some files (e.g., D1259, H71) show very low average scores (light yellow) in Figure 9c, but these files are not marked as "low-quality". How should this discrepancy be explained?

2. The authors mention many metrics in the metrics section, but when using the software, they refer to vague terms such as "quantitative intensity" for analysis. What intensity? Need to specify. This issue arises multiple times throughout the manuscript. It is essential to clearly define which metrics were used in the analysis.

3. In Supplementary Table 1A, there is a spelling error ("ionzation") and formatting issues, such as extra spaces. Similar issues should be checked throughout the manuscript.

4. Some figure captions and titles in the main text are unclear. For instance, "median peaks number of MS1" should be revised to "median peak count in MS1 spectra" for clarity.

(Remarks on code availability)

Reviewer #2

(Remarks to the Author)

The effort the authors put into addressing the reviewers' comments is acknowledged and appreciated. The authors conducted additional experiments and revised their manuscript accordingly by adding new data and text to address the concerns raised by the reviewers. The manuscript has also been restructured to effectively highlight the main points the authors would like to convey regarding the novelty and applicability of their MSCohort QC system. The overall quality of the manuscript has been improved considerably. It would be helpful for the authors to add their response to my general comment #2 regarding the evaluation of different LC systems to their manuscript. Otherwise, I do not have any additional concerns regarding the soundness of the research described in the manuscript.

(Remarks on code availability)

Reviewer #3

(Remarks to the Author)

(Remarks on code availability)

Reviewer #5

(Remarks to the Author)

In their revised manuscript, the authors have taken strides forward by addressing comments from the earlier revision. While some sections of text remain somewhat nebulous, the revision helped clarify the metric formulations and their applications.

Supplementary Note 1 in the "sm3njz" Supplementary Information provides a table detailing the 58 intra-experiment metrics employed by MSCohort, and the authors indicate novel metrics by blue text. They observe that DIA has been less well-supported by quality metrics software, and they're correct in that assessment. An interesting preprint describing iDIA-QC appeared in June of 2024, but I believe it has not appeared "in press" yet. The choices they have made for DIA metric inclusion seem reasonable. I do not believe that I could reproduce all their metrics based on the five- or six-word descriptions, but their captions are reasonably clear.

The description of applying the metrics is improved. The authors describe an interesting mapping process for grading each metric to one of five values; I believe the intent here is to contextualize a given metric in light of the values produced by all other experiments in a set. The authors describe the use of unsupervised learning to judge whether a RAW file represents an outlier (inconsistent performance); this has been handled in other cases by PCA, but unsupervised learning can certainly be applied here.

On the other hand, I believe this statement represents an overclaim: "Through the DIA scoring formula (1) and comprehensive metrics analysis reports, MSCohort provides direct explanations for the underlying causes of data results." The DIA scoring formula relates identifications to precursors and MS/MS scans, so it surely contributes little to identify specific causes of DIA failure. If the electrospray needle "coughs" in the middle of an experiment, it may impact identification relatively little even though it disrupts quantification for a range of retention times; no aspect of the scoring formula (1) would capture that event.

The authors delay citations for "96 DRA-Urine" and "HRMS1" to relatively late in the manuscript even though both make their first appearances earlier. I was surprised to see that the SOP DIA methods (Supplementary Note 3) rarely include overlaps between successive windows. This can reduce sensitivity when peptide ions have m/z values very close to the window boundary. I believe that the inclusion of 80 windows in the Orbitrap DIA method will mean that fragment ion chromatograms are updated with new intensities not more than once every nine or ten seconds, destroying the quantitative value of the fragment ion chromatograms. The frequency of MS scans will be higher due to multiple MS events per cycle, so the precursor chromatograms should be reasonably good. The shortcomings in fragment ion chromatograms would likely be increased problems with "interference," when different precursors have similar m/z and retention time (we have battled this problem with in-house use of the "High-Resolution MS1" protocol for DIA).

The authors praise their SOP for its technical achievement in individual samples, and yet the goal of an SOP is comparability among instruments rather than excellence in individual experiments. I would have expected to see more commentary about improved CV values for invariant peptides or proteins to establish its quantitative stability.

The authors have used human SwissProt proteins from the reference proteome rather than the full reference proteome. If they wanted their sequence database to contain a sequence for every protein-coding gene in the human genome, they cannot restrict the TrEMBL components of the reference proteome from inclusion. The database they've chosen here omits many human genes. This is a common phenomenon for human proteomics papers, but I believe most authors are unaware that they are omitting real protein-coding genes by the practice.

I would have hoped to see mzML rather than .ms1, .ms2 for metric computation, but I appreciate that simple text formats are easier to manage.

The manuscript frequently contains missing articles ("a," "an") or other language problems. I would point to "outstanding success in most sceneries" (meaning scenarios) or "anomalies nature" (meaning anomalous) since they appear in close proximity.

(Remarks on code availability)

Point-by-point Response to Reviewers' Comments

Standard operating procedure combined with comprehensive quality control system to enable large-scale urinary proteomics across multiple LC-MS platforms for precision medicine

To all reviewers,

We thank all the reviewers for their useful and detailed suggestions and comments. We have revised the manuscript accordingly, making substantial scientific improvements for both readers and users.

To fully address the comments, we have performed additional experiments, included new results, and added new supplementary notes to better present our findings.

Compared to the previous manuscript submitted, this revised manuscript added two additional notes related to the MSCohort QC system in the **Supplementary Information**: “Supplementary Note 1: Comparison of the Metrics in MSCohort and the Existing Metrics in other Quality Control Tools” and “Supplementary Note 2: Comparison of MSCohort and other Quality Control Tools”. These notes primarily explain the sources of metrics in MSCohort, the newly added metrics, as well as the novelty and application of the MSCohort software.

We also added a new section in the **Results** titled “Optimization and establishment of SOP for urinary proteomics with MSCohort QC system”, which illustrated the optimization and establishment process of our SOP, along with its advantages. Additionally, we provided and revised the detailed SOP in **Supplementary Information**: “Supplementary Note 3: The Standard Operating Procedure (SOP) for Urinary Proteomics”. We also added two additional notes related to the optimization and establishment of SOP in the **Supplementary Information**: “Supplementary Note 4: Optimization of LC-MS Method for Establishment of SOP for Urinary Proteomics” and “Supplementary Note 5: The Novelty of Our Developed Standard Operating Procedure (SOP) for Urinary Proteomics”. These notes primarily explain the process of optimizing SOP parameters using the MSCohort QC system and the novelty of our SOP for urinary proteomics.

The complete list of changes and new data is detailed in For-review Table 1. Our point-by-point replies are provided below.

For-review Table 1. The revised and newly added items in the revised manuscript.

Type	Current item	Revised/New	Contents
Main Figure	Figure 1	Revised	Overall study design and implementation
Main Figure	Figure 2	Revised	The workflow of MSCohort
Main Figure	Figure 3	New	The developed standard operating procedure (SOP) for urinary proteomics
Main Figure	Figure 4	Revised	The qualitative and quantitative performance across 20 LC-MS platforms

Supplementary Information	Supplementary Note 1 Table S.Note1	New	Comparison of the metrics in MSCohort and the existing metrics in other quality control tools
Supplementary Information	Supplementary Note 2 Table S.Note2	New	Comparison of MSCohort and other quality control tools
Supplementary Information	Supplementary Note 3	Revised	The standard operating procedure (SOP) for urinary proteomics
Supplementary Information	Supplementary Note 4 Figure S.Note 4.1 Figure S.Note 4.2 Figure S.Note 4.3	New	Optimization of LC-MS method for establishment of SOP for urinary proteomics
Supplementary Information	Supplementary Note 5	New	The novelty of our developed standard operating procedure (SOP) for urinary proteomics
Supplementary Information	Supplementary Figure 1	Revised	The overview of MSCohort intra-experiment analysis for DIA experiment
Supplementary Information	Supplementary Figure 5	New	The analysis of the protein sequence coverage, the relative abundance, and the dynamic range across 20 LC-MS platforms.
Supplementary Tables	Supplementary Table 1A, 1B	Revised	The detailed description of 58 intra-experiment metrics and 23 inter-experiment metrics.
Supplementary Tables	Supplementary Table 2A	New	The overview of the LC-MS conditions for 20 LC-MS platforms
For-review only	For-review Figure 1	New	The screenshots of the MSCohort interface
	For-review Note 1	New	The user manual for MSCohort
	For-review Figure 2	New	The scoring formula for DDA experiment in our previously published quality control software MSRefine
	For-review Table 2	New	Publications in MS-based large cohort proteome research
	For-review Figure 3	New	An overview of publications dealing with large cohort proteome studies
	For-review Figure 4	New	The screenshots of the MSCohort software download link in GitHub
	For-review Figure 5	New	The screenshots of explanation for users to adjust the scoring standards in MSCohort software
	For-review Figure 6	New	The instrument performance comparisons for the platforms with and without FAIMS
	For-review Figure 7	New	The screenshots of explanation for users to set the time-line report in MSCohort software

Reviewer #1 (Remarks to the Author):

QC is crucial for large scale proteomics experiments as the throughput increases rapidly. Urine is also very important specimens for biomarker discovery. This manuscript certainly touched the frontier of the field, However, it is unclear to the reviewer that what is the focus of this this manuscript. They mentioned the following aspects: 1) a SOP for generating urine MS-based proteome data using urine protein digests; 2) a QC software system called MSCohort. 3) Consistency of proteome analysis using different MS instruments (Lumos, E480, TIMS), including an application of CRC cohort. The reviewer thinks the 2nd point (MSCohort) is probably the major novelty of this manuscript since the major advantage of the SOP compared to the existing ones is the MSCohort. Comparative proteome analyses across different MS instruments have been published too.

Reply to general concerns:

We thank the reviewer for the valuable comments, pointing out the problems, and putting forward suggestions, which help us improve our manuscript.

As described in our title “**Standard operating procedure combined with comprehensive quality control system to enable large-scale urinary proteomics across multiple LC-MS platforms for precision medicine**” and “Fig.1 Overall study design and implementation”. In the technical perspective, there are two novelties in this manuscript: (1) the MSCohort QC system, and (2) the standard operating procedure (SOP) for urinary proteomics. Then, the above MSCohort QC system and SOP were applied to urinary proteome research.

Compared to the previous manuscript submitted, this revised manuscript added two additional notes related to the MSCohort QC system in the **Supplementary Information**: “Supplementary Note 1: Comparison of the Metrics in MSCohort and the Existing Metrics in other Quality Control Tools” and “Supplementary Note 2: Comparison of MSCohort and other Quality Control Tools”. These notes primarily explain the sources of metrics in MSCohort, the newly added metrics, as well as the novelty and application of the MSCohort software.

We also added a new section in the **Results** titled “Optimization and establishment of SOP for urinary proteomics with MSCohort QC system”, which illustrated the optimization and establishment process of our SOP, along with its advantages. Additionally, we provided and revised the detailed SOP in **Supplementary Information**: “Supplementary Note 3: The Standard Operating Procedure (SOP) for Urinary Proteomics”. We also added two additional notes related to the optimization and establishment of SOP in the **Supplementary Information**: “Supplementary Note 4: Optimization of LC-MS Method for Establishment of SOP for Urinary Proteomics” and “Supplementary Note 5: The Novelty of Our Developed Standard Operating Procedure (SOP) for Urinary Proteomics”. These notes primarily explain the process of optimizing SOP parameters using the MSCohort QC system and the novelty of our SOP for urinary proteomics.

Urinary proteomics differs from proteomics research of other cells and tissues in terms of technical emphasis, due to the urine samples exhibiting lower protein concentration, higher complexity, and a wider protein dynamic range. To our knowledge, prior to this study, there have been no published comparative evaluations across multiple LC-MS platforms in urinary proteomics. Currently, there is no standardized SOP for urinary proteomics research, with each laboratory employing its own experiment and data acquisition procedures, leading to significant variation across instruments, platforms, and laboratories. Our study provided a comprehensive QC system and reference SOP for large-scale urine proteomic analysis spanning different LC-MS platforms, which would benefit the applications of urine proteomics to clinical disease research.

Below, we give the responses one by one according to the reviewer's suggestions.

In this case, the reviewer would expect to see more technical details of MSCohort than that provided here. From the manuscript as it is, it is difficult to evaluate the novelty of this QC tool. Several key issues should be clarified:

1.1 explain more about the 70 metrics. Are they all existing ones used by the literature? Any new metrics developed here? What's the difference between MSCohort compared to others in the literature?

Reply:

Thanks for the reviewer's professional advice. In the revised manuscript, we specifically added two notes in the **Supplementary Information** to answer these two issues: "Supplementary Note 1: Comparison of the Metrics in MSCohort and the Existing Metrics in other Quality Control Tools" and "Supplementary Note 2: Comparison of MSCohort and other Quality Control Tools". We also made modifications in the **Results** "Design of MSCohort for comprehensive data quality control" section in the main text (**Page 5-7, Line 141-196**).

(1) explain more about the 70 metrics. Are they all existing ones used by the literature? Any new metrics developed here?

In the revised manuscript, we comprehensively analyzed and integrated the metrics proposed by existing quality control software. Building upon the initial 70 metrics in the previous manuscript, we included an additional 11 intra-experiment metrics from existing software, generating a total of 81 metrics. We strive to provide the most comprehensive set of QC metrics for proteomics. In **Supplementary Note 1**, we provided a detailed comparison between the metrics extracted in MSCohort and those of existing tools. The following is **Supplementary Note 1**:

"The NIST MSQC¹ is the most classic quality control (QC) metric for proteomics mass spectrometry experiments, upon which a series of subsequent QC software tools have been developed

with additional features and improvements, such as QuaMeter^{2, 3}, RawBeans⁴, DO-MS^{5, 6}, PTXQC⁷, QCloud^{8, 9}, QC-ART¹⁰, MSstatsQC 2.0¹¹, and QuiC¹², etc. Additionally, we have developed MSRefine¹³, a QC software specifically tailored for Data-Dependent Acquisition (DDA) experiments.

We strive to provide the most comprehensive set of proteomics QC metrics. Our MSCohort software summarizes and integrates metrics extracted by existing QC software and introduces 26 new ones, totaling 81 QC metrics. In **Table S.Note1**, we have compiled a comparison between the 81 QC metrics extracted by MSCohort and existing metrics. In this paper, based on whether the QC metrics are extracted from a single experiment or multiple experiments, all these metrics are classified into two categories: 58 intra-experiment metrics and 23 inter-experiment metrics. Detailed explanations and extraction processes for these 81 metrics are provided in **Supplementary Table 1A and Table 1B**.

1) 58 intra-experiment metrics. Existing 48 metrics reported by other tools mainly focus on DDA experiments, few metrics were developed specially for data-independent acquisition (DIA) experiments. In this study, we **proposed 10 new intra-experiment metrics for DIA individual experiments**.

We added 3 new metrics related to the DIA acquisition process, including W3. Redundant identified precursors/Identified scan rate, W4. Redundant identified precursors/Identified precursors rate, W5. Identified precursors/Identified scan rate. These metrics are derived from the principles of DIA acquisition and the DIA scoring formula proposed in this study, representing spectra complexity, precursors duplicate identification rate, and the utilization rate of the MS2 scans, respectively. These three metrics directly reflect the quality of the collected DIA data from a data interpretation perspective, playing a crucial role in the quality assessment and optimization of DIA experiments. A detailed explanation of these three metrics can be found in the Methods section.

We added 7 new metrics related to the DIA identification results, including S2. Median peptide length, W2. Median of window size (Da), W6. MS1 data points per peak, W7. MS2 data points per peak, M20. Maximum identification rate over m/z range, ID6. Average peptides per protein group, ID7. Average precursors per protein group. These metrics are extracted from the DIA search engine and reflect the quality of the identification results.

In addition to the 10 new metrics mentioned above, the remaining 48 metrics are integrated from existing quality control tools. The quality control software tool extracts relevant metrics based on specific data types (such as DDA or DIA).

According to steps in the LC-MS workflow, the 58 intra-experiment metrics can be assigned to seven categories: sample, chromatography, ion source, DIA windows, dynamic sampling, MS1 and MS2 signal, and identification result. Among them, the metrics in the DIA windows category are applicable only to data-independent acquisition (DIA) experiments, and the metrics in the dynamic sampling category are applicable only to data-dependent acquisition (DDA) experiments, the others are applicable to both DDA and DIA experiments.

2) 23 inter-experiment metrics. Previous quality control metrics did not specifically consider the contaminants for specific sample types, and did not systematically analyze the variation in precursors, peptides, and proteins levels. In this study, we **added 16 new inter-experiment metrics** for comprehensive inter-experiment analysis.

We added customizable contaminants metrics to assess the quality of samples. MSCohort offers configurable lists of custom protein contaminants. In this study, we added 3 contaminants metrics, including erythrocytes¹⁴, cellular debris¹⁵, and serum proteins¹⁶, for monitoring the quality and level of contaminants in urine samples. These 3 contaminants were previously reported as urine-specific quality marker panels to assess the degree of contamination of the urine samples¹⁵. A detailed explanation of these three sample contaminants metrics can be found in the **Methods** section. Users can customize and edit the contaminants list based on the actual sample situation. Detailed instructions for making these modifications are provided in the user manual of MSCohort (<https://github.com/BUAA-LiuLab/MSCOhort>).

We added 13 new metrics related to the quantification results in precursors, peptides and proteins level, including the ratio of missing values of precursors/proteins, median of precursors/proteins intensity, IQR (Inter Quartile Range) of precursors/proteins intensity, Pearson correlation of precursors/proteins intensity, robust dev (robust standard deviation)¹⁷ of precursors/ peptides /proteins intensity, and normalization factor of precursors/ peptides /proteins intensity, to comprehensively analyze the consistency and variation of quantification results.

In addition to the 16 new metrics mentioned above, the remaining 7 metrics are integrated from existing quality control tools. These 23 inter-experiment metrics are utilized to assess the stability of chromatography and mass spectrometry across experiments and to detect outlier data. **These metrics are universal and can be employed for inter-experiment quality evaluation of both DIA and DDA data, as well as for inter- experiment quality evaluation of Parallel Reaction Monitoring (PRM) data.**

Altogether, by extracting the compressive metrics, MSCohort software supports the analysis of data from different types of instrument platforms, including Thermo Scientific Orbitrap, Bruker timsTOF, and SCIEX ZenoTOF. The 81 metrics reported by MSCohort include identification-free (ID-free) metrics directly extracted from .raw/.d/.wiff raw files, as well as ID-based metrics extracted from identification/quantification results. MSCohort can apply not only to process evaluation and optimization of individual experiments for DDA and DIA data, but also to process quantitative assessment of system performance across multiple experiments for DIA, DDA, and PRM data.”

(2) What's the difference between MSCohort compared to others in the literature?

In the revised manuscript, we added a new note in **Supplementary Information**: “Supplementary Note

2: Comparison of MSCohort and other Quality Control Tools”. And a comparison between MSCohort software and existing quality control software has also been summarized in **Table S.Note2**. The following is **Supplementary Note 2**:

“Referring to the review by Wout et al.¹⁸, we divide the currently published quality control tools into two categories: tools evaluating individual experiments and tools comparing multiple experiments.

The first category of tools includes NIST MSQC¹, QuaMeter^{2, 3}, RawBeans⁴, DO-MS^{5, 6}, which focus on the evaluation and optimization of individual experiment. These tools only extract a limited number of metrics and display them in the form of charts, without quantifying or scoring the metrics. Users need to analyze and understand them on their own. None of these tools propose systematic and comprehensive metrics for deeper proteome profiling and illustrate the relationship between the metrics and identification results. These tools are unable to directly assist users in identifying issues and indicating optimization directions. Users must rely on expert experience to gradually optimize parameters and compare results, a time-consuming and labor-intensive process heavily influenced by human factors, thereby making it challenging to ensure experimental outcomes.

In particular, the above-mentioned software tools primarily focus on the quality evaluation of DDA data, with fewer software options available for DIA data quality control, offering limited metrics and simplistic functionalities.

The second category of tools includes PTXQC⁷, QCloud^{8, 9}, QC-ART¹⁰, MSstatsQC 2.0¹¹, QuiC¹², which focus on assessing the system performance and variation among multiple experiments. PTXQC and MSstatsQC 2.0 only support the extraction of limited inter-experimental metrics from the identification and quantitative results derived from search engines, and visualize them through charts. QC-ART calculates and reports a distance score using an rPCA model based solely on user-provided identification and quantification result tables to detect outlier data, lacking detailed reporting and comprehensive inter-experiment variation and quality analysis for each metric. QCloud and QuiC only support specific QC samples or require the inclusion of iRT peptides, performing consistency and bias analysis across experiments solely on peptides specified by the software, rendering them unsuitable for all sample types. These tools have limited scalability and are incapable of conducting comprehensive and systematic analysis for complex large-scale proteomic datasets.

These motivate us to develop MSCohort for comprehensive quality control for proteomics. Table S.Note2 presents a comparison of MSCohort with existing quality control software tools.

First, we strive to provide the most comprehensive set of proteomics QC metrics. MSCohort software summarizes and integrates metrics extracted by existing QC software and introduces new ones, totaling 81 QC metrics (see Supplementary Note 1 for details).

Second, MSCohort integrated MSRefine¹³, a quality control analysis system previously developed by our group for individual DDA experiments, and established a quality control analysis system for

individual DIA experiments. Consequently, regardless of whether the data is DDA or DIA, MSCohort extracts comprehensive metrics mapping to the whole LC-MS workflow, illustrates the relationship between extracted metrics and identification results, scores the metrics, and reports visual results, assisting users in evaluating the workflow, and locating problems.

Third, for the cohort proteomics data, MSCohort extracted 23 comprehensive inter-experiment metrics to evaluate multiply experiments. MSCohort also incorporates unsupervised machine learning algorithm (isolation forest) to detect potential outlier experiments. Furthermore, to guarantee the reliability of the subsequent statistical analyses, it incorporates various normalization methods to remove systematic bias in peptide/protein abundances that could mask true biological discoveries or give rise to false conclusions¹⁹.

Fourth, MSCohort is a powerful tool that supports not only whole LC-MS workflow evaluation and optimization for individual experiments but also robustness assessment and outlier data detecting among multiple experiments. MSCohort also supports comprehensive quality control for different sample types (e.g. cell, tissue, plasma, urine, etc.), different instrument types (Thermo Scientific Orbitrap, Bruker timsTOF, and SCIEX ZenoTOF), and different acquisition modes (DDA, DIA, PRM).”

(Supplementary Note 2)

Table S.Note2. An overview of the computational quality control tools for LC-MS based proteomics

Tool	Number of metrics	Experiment type	Instrument	ID-free metrics	ID-based metrics	Main Functions	Detailed Functions
MSQC ¹	46	DDA	Thermo	✓	✓	Tools evaluating individual experiments	 • Extract metrics
QuaMeter ^{2, 3}	45	DDA	Thermo, Bruker, SCIEX	✓	✓	Tools evaluating individual experiments	 • Extract metrics
RawBeans ⁴	13	DDA, DIA	Thermo, Bruker, SCIEX	✓	✓	Tools evaluating individual experiments	 • Extract metrics • Generate visual report
DO-MS ^{5, 6}	35	DDA, DIA	Thermo, Bruker	✓	✓	Tools evaluating individual experiments	 • Extract metrics • Generate visual report
MSRefine ¹³	47	DDA	Thermo, Bruker, SCIEX	✓	✓	Tools evaluating individual experiments	 • Extract metrics • Provide a scoring system • Illustrate the relationship between metrics and identification results • Generate visual report
PTXQC ⁷	24	DDA	Thermo, Bruker, SCIEX		✓	Tools comparing multiple experiments	 • Extract metrics • Compute the quality score • Generate visual report
QCloud ^{8, 9}	23	DDA, PRM, SRM	Thermo, Bruker, SCIEX		✓	Tools comparing multiple experiments	 • Extract metrics • Generate visual report
QC-ART ¹⁰	User-input	DDA	Not defined		✓	Tools comparing multiple experiments	 • Detect outlier experiments • Generate visual report
MSstatsQC 2.0 ¹¹	User-input	DDA, DIA, SRM	Not defined		✓	Tools comparing multiple experiments	 • Generate visual report
QuiC ¹²	22	DDA, DIA, MRM, PRM	Thermo, Bruker, SCIEX	✓	✓	Tools comparing multiple experiments	 • Extract metrics • Generate visual report
MSCohort (This study)	81	DDA, DIA, PRM	Thermo, Bruker, SCIEX	✓	✓	Tools evaluating individual experiments and comparing multiple experiments	 • Extract metrics • Provide a scoring system • Illustrate the relationship between metrics and identification results • Compute the quality score • Generate visual report • Detect outlier experiments • Data normalization

References

1. Rudnick, P.A., et al. Performance metrics for liquid chromatography-tandem mass spectrometry systems in proteomics analyses. *Mol. Cell. Proteomics* **9**, 225-241 (2010).
2. Ma Z., et al. QuaMeter: multivendor performance metrics for LC-MS/MS proteomics instrumentation. *Anal. Chem.* **84**, 5845-5850 (2012).
3. Wang X., Chambers M.C., Vega-Montoto L.J., Bunk D.M., Stein S.E., Tabb D.L. QC metrics from CPTAC raw LC-MS/MS data interpreted through multivariate statistics. *Anal. Chem.* **86**, 2497-2509 (2014).
4. Morgenstern D., Barzilay R., Levin Y. RawBeans: A Simple, Vendor-Independent, Raw-Data Quality-Control Tool. *J. Proteome Res.* **20**, 2098-2104 (2021).
5. Huffman R.G., Chen A., Specht H., Slavov N. DO-MS: Data-Driven Optimization of Mass Spectrometry Methods. *J. Proteome Res.* **18**, 2493-2500 (2019).
6. Wallmann G., Leduc A., Slavov N. Data-Driven Optimization of DIA Mass Spectrometry by DO-MS. *J. Proteome Res.* **22**, 3149-3158 (2023).
7. Bielow C., Mastrobuoni G., Kempa S. Proteomics Quality Control: Quality Control Software for MaxQuant Results. *J. Proteome Res.* **15**, 777-787 (2016).
8. Chiva C., et al. QCloud: A cloud-based quality control system for mass spectrometry-based proteomics laboratories. *PLoS One* **13**, e0189209 (2018).
9. Olivella R., et al. QCloud2: An Improved Cloud-based Quality-Control System for Mass-Spectrometry-based Proteomics Laboratories. *J. Proteome Res.* **20**, 2010-2013 (2021).
10. Stanfill B.A., et al. Quality Control Analysis in Real-time (QC-ART): A Tool for Real-time Quality Control Assessment of Mass Spectrometry-based Proteomics Data. *Mol. Cell. Proteomics* **17**, 1824-1836 (2018).
11. Dogu E., et al. MSstatsQC 2.0: R/Bioconductor Package for Statistical Quality Control of Mass Spectrometry-Based Proteomics Experiments. *J. Proteome Res.* **18**, 678-686 (2019).
12. QuiC 5: Quality Control Monitoring in the Blink of an Eye. https://biognosys.com/content/uploads/2023/10/UserManual__QuiC5.pdf. (2023).
13. Tang M., et al. Comprehensive Evaluation and Optimization of the Data-Dependent LC-MS/MS Workflow for Deep Proteome Profiling. *Anal. Chem.* **95**, 7897-7905 (2023).
14. Geyer, P.E., et al. Plasma Proteome Profiling to detect and avoid sample-related biases in biomarker studies. *EMBO Mol. Med.* **11**, e10427 (2019).
15. Virreira Winter, S., et al. Urinary proteome profiling for stratifying patients with familial Parkinson's disease. *EMBO Mol. Med.* **13**, e13257 (2021).
16. Guo, Z., et al. Analysis of the differential urinary protein profile in IgA nephropathy patients of Uygur ethnicity. *BMC Nephrol.* **19**, 358 (2018).
17. Cox, J. & Mann, M. MaxQuant enables high peptide identification rates, individualized p.p.b.-range mass accuracies and proteome-wide protein quantification. *Nat. Biotechnol.* **26**, 1367-1372 (2008).
18. Bittremieux W., Valkenburg D., Martens L., Laukens K. Computational quality control tools for mass spectrometry proteomics. *Proteomics* **17**, 1600159 (2017).
19. Jiang, Y., et al. Comprehensive Overview of Bottom-Up Proteomics using Mass Spectrometry. *ArXiv* (2023).

1.2 Are these metrics for DDA or DIA? DDA and DIA are different methods. Throughout the manuscript, it is frequently unclear whether the authors performed DDA or DIA QC runs, and how they analyzed the data.

Reply:

Thanks for the reviewer's detailed suggestion.

First, we need to specifically explain that our group previously developed MSRefine¹, which aims to evaluate and optimize the performance of the LC-MS workflow for individual data-dependent acquisition (DDA) experiments. MSRefine extracts 47 kinds of comprehensive metrics for DDA proteomics and provides a scoring formula to illustrate the relationship between the metrics and identification results. In addition, MSRefine scores the metrics, and reports visual results, assisting users in evaluating the workflow, and locating problems.

[1] Min Tang[#], Peiwu Huang[#], Lize Wu, Piyu Zhou, Pengyun Gong, Xiang Liu, Qiushi Wei, Xinhang Hou, Hongke Hu, Ao Zhang, Chengpin Shen, Weina Gao, Ruijun Tian^{*}, and Chao Liu^{*}. Comprehensive Evaluation and Optimization of the Data-Dependent LC-MS/MS Workflow for Deep Proteome Profiling. *Analytical Chemistry*, 2023, May 23;95(20):7897-7905.

Second, MSCohort established a quality control analysis system for individual DIA experiments, which is one of the important novelties of MSCohort. We proposed 10 intra-experiment metrics for DIA individual experiments, and the remaining 48 intra-experiment metrics are integrated from existing quality control tools. According to steps in the LC-MS workflow, the 58 intra-experiment metrics can be assigned to seven categories: sample, chromatography, ion source, DIA windows, dynamic sampling, MS1 and MS2 signal, and identification result. Among them, the metrics in the DIA windows category are applicable only to data-independent acquisition (DIA) experiments, and the metrics in the dynamic sampling category are applicable only to data-dependent acquisition (DDA) experiments, the others are applicable to both DDA and DIA experiments. These 23 inter-experiment metrics are utilized to assess the stability of chromatography and mass spectrometry across experiments and to detect outlier data. These metrics are universal and can be employed for inter-experiment quality evaluation of both DIA and DDA data, as well as for inter-experiment quality evaluation of PRM data. We provide the detailed explanation for these metrics in "Supplementary Note 1: Comparison of the Metrics in MSCohort and the Existing Metrics in other Quality Control Tools".

Third, we sincerely apologize for the confusion caused to reviewers and readers regarding the data analysis process. All data pertaining to urinary proteomics in this main text were acquired using the Data-Independent Acquisition (DIA) strategy. Only in Supplementary Note 6 and Supplementary Figure 15, the DDA results from 20 LC-MS platforms were presented to illustrate that in urinary proteomics analysis, the DIA strategy offers higher identification depth and better consistency across different platforms than DDA.

Fourth, from the software engineering perspective, MSCohort integrates MSRefine, enabling users to perform quality control for DDA experiments using MSCohort. **For-review Figure 1** displays the screenshots of the MSCohort interface. MSCohort as a whole software tool for users, assisting them in performing quality control analysis for DDA or DIA experiments by selecting appropriate parameters. The detailed user manual

of MSCohort is provided in Github: (<https://github.com/BUAA-LiuLab/MS Cohort>), and we also provide the user manual of MSCohort as an attachment in **For-review Note 1**.

For-review Figure 1. The screenshots of the MSCohort interface. Users can perform comprehensive quality control analysis for DDA, DIA, and PRM experiments by selecting appropriate parameters.

1.3 It seems the 70 metrics are divided into 47 metrics over the entire LC-MS workflow for DIA + 23 inter-experiment metrics. If so, the authors need to explain this much clearer. Are they applicable to DDA or SRM/PRM?

Reply:

Thanks for the reviewer's detailed suggestion. The response to this point is already covered in response to **Reviewer#1 Reply 1.1 and Reply 1.2** above. Specially, we added a detailed description in **Supplementary Information**: "Supplementary Note 1: Comparison of the Metrics in MSCohort and the Existing Metrics in other Quality Control Tools" and "Supplementary Note 2: Comparison of MSCohort and other Quality

Control Tools”. We also made modifications in the **Results** “Design of MSCohort for comprehensive data quality control” section in the main text (**Page 5-7, Line 141-196**).

We include it again below. In the revised manuscript, we comprehensively analyzed and integrated the metrics proposed by existing quality control software. Building upon the initial 70 metrics in the previous manuscript, we included an additional 11 intra-experiment metrics from existing software, generating a total of 81 metrics (58 intra-experiment metrics and 23 inter-experiment metrics). We strive to provide the most comprehensive set of QC metrics for proteomics.

According to steps in the LC-MS workflow, the 58 intra-experiment metrics can be assigned to seven categories: sample, chromatography, ion source, DIA windows, dynamic sampling, MS1 and MS2 signal, and identification result. Among them, the metrics in the DIA windows category are applicable only to data-independent acquisition (DIA) experiments, and the metrics in the dynamic sampling category are applicable only to data-dependent acquisition (DDA) experiments, the others are applicable to both DDA and DIA experiments.

The 23 inter-experiment metrics are utilized to assess the stability of chromatography and mass spectrometry across experiments and to detect outlier data. These metrics are universal and can be employed for inter-experiment quality evaluation of both DIA and DDA data, as well as for inter-experiment quality evaluation of PRM data.

The quality control software extracts relevant metrics according to different experiment types.

Taken together, MSCohort can apply not only to process evaluation and optimization of individual experiments for DDA and DIA data, but also to process quantitative assessment of system performance across multiple experiments for DIA, DDA, and PRM data (**For-review Figure 1**).

1.4 Line 148: how did the authors justify this equation? What is the "corresponding scores" in line 153? Related to this: line 713-743: more details of this equation is explained, but it is confusing since this was used for DDA. It is unclear why the DDA equation is applicable to DIA data.

Reply:

Thanks for the reviewer’s professional advice, and we answered the questions point by point as follows:

(1) Line 148: how did the authors justify this equation? Related to this: line 713-743: more details of this equation is explained, but it is confusing since this was used for DDA. It is unclear why the DDA equation is applicable to DIA data.

1) As mentioned in our reply for **Reviewer#1 Reply 1.2** above, our group previously developed quality control software MSRefine (*Tang, et al. Anal Chem 2023, 95(20):7897-7905*), and provided a scoring formula for individual DDA experiments, as illustrated in **For-review Figure 2:**

[Editorial note: this figure was redacted due to third-party rights. The scoring formula can be found in Tang, et al. Anal Chem 2023, 95(20):7897-7905

For-review Figure 2. The scoring formula for DDA experiment in our previously published quality control software MSRefine (Tang, et al. Anal Chem 2023, 95(20):7897-7905).

2) Based on the different principles of the DDA and DIA experiment, we formulated equations for DIA experiment built upon the foundation of our previous DDA equations. In the revised manuscript, to provide a clearer explanation of the formulated equations, we added the DDA scoring formula in the **Methods** section: (Page 23-24, Line 768-791) “To represent the experimental conditions of DIA with a mathematical model, we design a quality scoring system for DIA data based on our previous DDA data quality scoring system.

The DDA scoring formula is expressed as:

$$N_{\text{identified_precursors}} = N_{\text{acquired_MS2}} \times Q_{\text{MS2}} \times P_{\text{MS2_per_precursor}} \quad (1)$$

where $N_{\text{identified_precursors}}$ is the number of identified peptide precursors, $N_{\text{acquired_MS2}}$ is the number of acquired MS2 scans, Q_{MS2} is the identification rate of the MS2 scans (the number of identified MS2 scans/ the number of acquired MS2 scans), and $P_{\text{MS2_per_precursor}}$ is the utilization rate of the MS2 scans (the number of unique peptide precursors/ the number of identified MS2 scans)

The DIA scoring formula is expressed as:

$$N_{\text{identified_precursors}} = N_{\text{acquired_MS2}} \times Q_{\text{MS2}} \times (N_{\text{precursor_per_MS2}}/R_{\text{precursor}}) \quad (2)$$

where $N_{\text{identified_precursors}}$ is the number of identified peptide precursors, $N_{\text{acquired_MS2}}$ is the number of acquired MS2 scans, Q_{MS2} is the identification rate of the MS2 scans (the number of identified MS2 scans/ the number of acquired MS2 scans), $N_{\text{precursor_per_MS2}}$ is the spectra complexity of MS2 scans (the number of redundant identified precursors/ the number of identified MS2 scans), $R_{\text{precursor}}$ is the precursors duplicate identification rate (the number of redundant identified precursors / the number of identified precursors), and $N_{\text{precursor_per_MS2}}/R_{\text{precursor}}$ is the utilization rate of the MS2 scans (the number of unique peptide precursors/ the number of identified MS2 scans).

This DIA scoring formula was designed based on the DDA scoring formula, the utilization rate of the MS2 scans was divided into the spectra complexity of MS2 scans and the precursors duplicate identification rate. Since the DIA method was to fragment all the parent ions in the isolation window to obtain a mixture MS2 spectrum, theoretically an MS2 scan can be identified to multiple precursors. Therefore, we have established a spectrum complexity index to represent the number of precursors that can be identified by an average MS2 spectrum/scan.”

(2) What is the "corresponding scores" in line 153?

The "corresponding scores" in Line 153 represent the scoring results for each metric. It is not convenient for users to comprehensively evaluate the data if only displays the values of each metric. MSCohort computes a quality score for each of the QC metrics using a score function referring to our previously published MSRefine (Tang, *et al. Anal Chem* 2023, **95**(20):7897-7905).

We revised this sentence in the revised manuscript: (Page 6, Line 174-177) “MSCohort scores the relative metrics, reports the metric–score diagram, and flags the metrics with low scores to assist experimenters in assessing the quality of data directly, enabling systematic evaluation and optimization of individual DIA experiments (See Methods and Supplementary Fig.1 for details).”

In the revised **Methods** section of the revised manuscript, we have included detailed scoring procedures. This information can be found in (Page 25-26, Line 818-851) “Step 2.2 Calculation of the first-level and second-level scores”.

1.5 line 156: how to define "large cohort proteome data"? When should one use MSCohort?

Reply:

Thanks for the reviewer’s professional advice, and we answered the questions point by point as follows:

(1) how to define "large cohort proteome data"?

To obtain a comprehensive collection of publications involved large cohort proteome studies, we performed an unrestricted PubMed search specifying cooccurrence of the terms “large cohort”, “proteome”, and “mass spectrometry”. This yielded an initial list of 455 publications of which 50 were reviews. We further subtracted studies that did not deal with LC-MS based proteomics research or did not involve large proteome cohort, leaving **164** original publications. We listed these publications in **For-review Table 2**. In these large

cohort proteomic studies, the sample size is mainly between 50 and 500, with a median of 119 (**For-review Figure 3**). To clarify when should one use MSCohort, we have revised this sentence, changing "large cohort" as "cohort" (**Line 178**).

For-review Figure 3. An overview of publications dealing with large cohort proteome studies.

The analysis time of MSCohort increased linearly with the increase in the number of samples, as shown in our test results in (computer configuration: Processor (CPU): Intel Core i7-11800H 2.3GHz, Memory (RAM): 32GB). The analysis time was 2 min with MSCohort inter-experiment analysis to complete the quality control analysis results of 6 DIA experiments (for urinary proteomic data of 30-min acquisition time). The analysis time was about 40 min to complete the inter-experimental analysis of 160 DIA experiments (for urinary proteomic data of 30-min acquisition time). And The analysis time was about 420 min to complete the inter-experimental analysis of 1000 DIA experiments (for urinary proteomic data of 30-min acquisition time).

(2) When should one use MSCohort?

MSCohort consists of two main modules: intra-experiment analysis and inter-experiment analysis. MSCohort intra-experiment analysis can help users evaluate and optimize individual (single-run) experiments, and MSCohort inter-experiment analysis can help users evaluate the system stability across multiple (more than two runs) experiments and detect outlier experiments, ultimately enabling a comprehensive quality assessment of LC-MS experiments.

We advocate for the utilization of MSCohort in the following four application scenarios:

(i) Determining the appropriate experimental parameters for individual experiments to achieve deep coverage. Without MSCohort, this iterative process can be both time-consuming and laborious as it requires manual inspection and comparison of identification results across all parameter combinations. The intra-

experiment analysis in MSCohort helps users optimize experimental parameters by extracting comprehensive and systematic quality control (QC) metrics, establishing the relationship between metrics and identification results. By employing scoring formulas and generating comprehensive metric analysis reports, MSCohort provides direct explanations for the underlying causes of data outcomes. Therefore, users can determine appropriate parameter conditions through a limited number of experiments and determine the necessity for additional experiments based on MSCohort's reports, thereby facilitating rapid optimization of experimental parameters.

(ii) Evaluating the instantly completed experiments during the cohort experiments collection process. Without MSCohort, promptly locating the problems in experiments with suboptimal identification results can be challenging, necessitating a lengthy manual troubleshooting process. The intra-experiment analysis in MSCohort extracts comprehensive metrics that cover the entire LC-MS workflow, illustrates the relationship between extracted metrics and identification results, scores the metrics, and reports the metric–score diagram. This diagram effectively highlights underperforming metrics in relation to sample preparation, chromatography, and mass spectrometry, thereby facilitating prompt quality assessment and issue localization for these instantly completed experiments.

(iii) Assessing the stability and reproducibility of the LC-MS systems once the number of collected experiments exceeds 2 runs during the cohort experiments collection process. Without MSCohort, there is a lack of automated analytical tools for rapid assessment of reproducibility across multiple experiments and stability evaluation of LC-MS systems. The inter-experiment analysis in MSCohort helps users perform comprehensive and automated quality evaluation across multiple experiments, including evaluations of chromatographic retention time stability, qualitative and quantitative result stability, reproducibility, etc.

(iv) Conducting cohort data quality evaluation and detecting potential outlier experiments upon completion of the cohort experiment collection. Without MSCohort, the lack of comprehensive and objective assessment standards necessitates manual extraction and analysis of limited metrics, which is both time-consuming and highly susceptible to human bias. The inter-experiment analysis in MSCohort extracts comprehensive metrics that align with the entire LC-MS workflow and presents a heatmap overview, enabling a quick assessment of quality and facilitating the identification of low-performance experiments. Additionally, MSCohort also incorporates an unsupervised machine learning algorithm (isolation forest) to detect potential outlier experiments. Furthermore, to guarantee the reliability of the subsequent statistical analyses, it incorporates various normalization methods to eliminate systematic bias in peptide/protein abundances that could mask true biological discoveries or give rise to false conclusions.

In the revised manuscript, we demonstrate in Supplementary Note 3 how to use MSCohort for optimization and deeper proteome profiling for individual experiments in application scenarios (i); In the analysis of a dataset of 20 LC-MS urinary proteomics data, we showed in Supplementary Figure 3 how to conduct comprehensive quality assessment of individual experiments using MSCohort and identify the causes of experimental issues in application scenarios (ii); In the analysis of the benchmarking hybrid dataset,

we illustrate in Supplementary Figure 7 how to perform comprehensive quality assessment between multiple experiments, evaluating instrument system stability and reproducibility using MSCohort in application scenarios (iii); In the analysis of the CRC cohort dataset, we demonstrate in Supplementary Figure 9 how to conduct comprehensive quality assessment across cohort experiments, evaluating instrument system stability and detecting outlier experiments using MSCohort in application scenarios (iv).

1.6 line 168: please provide more details about "unsupervised machine learning algorithm". How did you justify this machine learning model? How was it trained, validated and tested?

Reply:

Thanks for the reviewer's professional advice, and we answered the questions point by point as follows:

(1) How did you justify this machine learning model?

Previous studies have shown that the LC-MS experimental process is complex, with numerous factors influencing LC-MS data, and these factors are not independent but may affect each other. Therefore, for high-dimensional and complex LC-MS data, supervised classifiers heavily rely on training data. Data from different instruments, laboratories, and sample types require re-labeling and retraining, leading to poor generalization. Consequently, unsupervised machine learning algorithms are commonly used for outlier data analysis (*Bittremieux, et al. Proteomics, 2017, 17(3-4); Stanfill, et al. Mol Cell Proteomics, 2018, 17(9):1824-1836*).

Isolation forest (iForest) is an excellent unsupervised and online outlier detection algorithm. iForest achieves outstanding success in most applications by taking advantage of the anomalies nature of "few and different" (*Zhou Z.H., et al. Isolation forest. in Proc. 8th IEEE Int. Conf. Data Mining. 2008:413-422*). iForest builds an ensemble of isolation trees (iTrees) for a given data set, then anomalies are those instances which have short average path lengths on the iTrees. This algorithm does not require a labeled dataset or pre-training of offline models, it can dynamically construct isolation trees online for any batch of data. It has unique advantages in dealing with large datasets due to its low-computational complexity (*Grekov, A.N. et al. Sensors (Basel), 2023, 23, 2687*).

As mentioned in the original iForest paper (*Zhou Z.H., et al. Isolation forest. in Proc. 8th IEEE Int. Conf. Data Mining. 2008:413-422*): "iForest is distinguished from existing model-based, distance-based, and density-based methods in the following ways: (i) iForest utilizes no distance or density measures to detect anomalies. This eliminates major computational cost of distance calculation in all distance-based methods and density-based methods; (ii) iForest has a linear time complexity with a low constant and a low memory requirement; (iii) iForest has the capacity to scale up to handle extremely large data size and high-dimensional problems with a large number of irrelevant attributes."

(2) How was it trained, validated and tested?

As mentioned in the original iForest paper (*Zhou Z.H., et al. Isolation forest. in Proc. 8th IEEE Int. Conf.*

Data Mining. 2008:413–422), the unsupervised and online outlier detection algorithm is a two-stage process. The first (training) stage builds isolation trees using sub-samples of the training set. The second (testing) stage passes the test instances through isolation trees to obtain an anomaly score for each instance. This algorithm does not require a labeled dataset or pre-training of offline models, it can dynamically construct isolation trees online for any batch of data.

First, the 23 inter-experiment metrics values for each experiment in the cohort were integrated into a two-dimensional matrix. Then, the outlier experiments detection was performed using the iForest algorithm with a two-stage process:

(1) Training stage: iForest randomly selects subsamples from the cohort, then a feature (metric) is randomly selected, and a separation value is randomly generated within the selected feature value range to “isolate” the sample point. Then iForest recursively selects different features and values from the child subset to split the child into smaller subsamples. iTrees are constructed by recursively partitioning the given training set until instances (samples) are isolated or a specific tree height is reached of which results a partial model. Many iTrees will make up the iForest. Thus, we can get the average path length of all iTrees in the iForest.

(2) Evaluating stage: iForest passes the samples through isolation trees to obtain an anomaly score for each sample. Outliers are those samples which have short average path lengths on the iTrees and low anomaly scores. The iForest was implemented using scikit-learn python library (version 0.23) module `sklearn.ensemble.IsolationForest` with default parameters.

To validate the performance of this model, we added the experiments known to be problematic to the cohort experiments as negative examples. The results showed that iForest model can detect negative examples as outlier experiments accurately, and these experiments have significantly lower scores. As shown in Supplementary Figure 9, the outlier experiments showed an obvious isolation than normal experiments.

We have added the details of the "unsupervised machine learning algorithm" in the **Methods** section (**Page 28-29, Line 937-971**)

1.7 As a user, can you provide detailed protocols as how to download, and use this software? So the reviewer suggests the authors to largely expand this MSCohort section, if this is the core of this manuscript. In particular, stress the novelty of the software compared to the literature.

Reply:

Thanks for your detailed suggestion. We answered the questions point by point as follows:

(1) As a user, can you provide detailed protocols as how to download, and use this software?

In the previous manuscript, we provided the download link and detailed protocols in **Code availability** as follows:

(Page 30) “MSCohort is developed in Python and is freely available. The latest software version can be downloaded from GitHub (<https://github.com/BUAA-LiuLab/MSCohort>).”

In the revised manuscript, we also added the information in the **Introduction (Page 5, Line 138-139)** and **Results (Page 7, Line 195-196)** section as follows:

“The MSCohort software tool and the user manual are available for download from Github: (<https://github.com/BUAA-LiuLab/MS Cohort>).”

For-review Figure 4 showed the screenshots of the MSCohort software download link in *GitHub* (<https://github.com/BUAA-LiuLab/MS Cohort>). For your convenience, we also provide the user manual of MSCohort as an attachment in **For-review Note 1**.

For-review Figure 4. The screenshots of the MSCohort software download link in *GitHub* (<https://github.com/BUAA-LiuLab/MS Cohort>). The user manual of MSCohort and demo data for users are provided in this link.

(2) So the reviewer suggests the authors to largely expand this MSCohort section, if this is the core of this manuscript. In particular, stress the novelty of the software compared to the literature.

As mentioned in our reply for **Reviewer#1 Reply 1.1** above, we have provided a response regarding the novelty of the software compared to others. We also made modifications in the **Results** “Design of MSCohort for comprehensive data quality control” section (**Page 5-7, Line 141-196**). Additionally, we added two notes in **Supplementary Information**: “Supplementary Note 1: Comparison of the Metrics in MSCohort and the Existing Metrics in other Quality Control Tools” and “Supplementary Note 2: Comparison of MSCohort and other Quality Control Tools”, stressing the novelty of the software compared to the literature.

More major issues in the rest sections to be clarified:

1.8 In the section starting from line 174 "comprehensive and comparative....": it is unclear what is the benefit of MSCohort. LC-SOP samples exhibited more reproducible results than that without LC-SOP, because the

latter have different LC systems. Can MSCohort rescue reproducibility for samples without LC-SOP, and further enhance reproducibility for samples with LC-SOP? How can one use MSCohort? This is a demo, so the author should explicitly demonstrate how to use MSCohort, and what is the added value.

Reply:

Thanks for the reviewer's professional suggestion, and we answered the questions point by point as follows:

(1) In the section starting from line 174 "comprehensive and comparative....": it is unclear what is the benefit of MSCohort. LC-SOP samples exhibited more reproducible results than that without LC-SOP, because the latter have different LC systems.

The results showed clear differences and lower reproducibility among the 10 LC-MS platforms without LC-SOP is in line with intuitive expectations, but the reason why it is low needs further detailed analysis. MSCohort helps us investigate the possible causes in detail (**Page 8-9, Line 244-268**). For instance, the main reasons for the low number of identifications in M03-E480^F and M03-E480^F are notably low identification rates of MS2 scans. The main reason for the low number of identifications in M06-Eclipse is the lower utilization rate of MS2 scans. And the main reason for the low number of identifications in M09-E240 is long chromatographic invalid acquiring time (LC delay time), leading to over 40% of spectra being wasted without identifying precursors. Without MSCohort, it would be challenging for different platforms to pinpoint such specific issues within the protocols they used.

(2) Can MSCohort rescue reproducibility for samples without LC-SOP, and further enhance reproducibility for samples with LC-SOP?

As mentioned above, the low reproducibility among the 10 LC-MS platforms without LC-SOP is attributed to different deficiencies in the protocols they used. Using MSCohort, we investigated these deficiencies and, of course, discovered areas worth adopting. Our SOP was established under the comprehensive optimization of MSCohort and it integrates the optimal strategies at each step under current conditions. We applied this SOP to urinary proteomics studies across multiple LC-MS platforms, and the results demonstrate improved reproducibility among the 10 LC-MS platforms with LC-SOP.

In addition, in terms of quantitative reproducibility, MSCohort provides various normalization algorithms (e.g. maxLFQ, directLFQ, quantile) to correct intensities across data generated from different instrument platforms, thereby enhancing reproducibility at the quantitative level.

(3) How can one use MSCohort? This is a demo, so the author should explicitly demonstrate how to use MSCohort, and what is the added value.

In our response to **Reviewer#1 Reply 1.5**, we summarized four application scenarios for applying MSCohort. Specifically, in the analysis of experimental results and discrepancies between experiments with and without LC-SOP. First, we conducted MSCohort intra-experiment analysis on each data, assessing the quality of individual experiments, identifying lower-performance data, and pinpointing their root causes.

Additionally, we performed MSCohort inter-experiment analysis on triplicate data collected from each platform to evaluate intra-platform reproducibility and consistency within each platform. Furthermore, we conducted MSCohort inter-experiment analysis on results from 10 platforms to analyze inter-platform reproducibility and consistency.

In addition, we provided the download link and detailed user manual for MSCohort in **Introduction (Page 5, Line 138-139)**, **Results (Page 7, Line 195-196)** and **Code availability (Page 30)** section as follows:

“MSCohort is developed in Python and is freely available. The latest software version can be downloaded from GitHub (<https://github.com/BUAA-LiuLab/MSCOhort>).”

We also provide the user manual of MSCohort as an attachment in **For-review Note 1**.

1.9 line 251+: it is not a surprise that DDA is less reproducible than DIA. Again, the authors should explicitly show the added value of MSCohort.

Reply:

Thanks for the reviewer’s constructive suggestion.

DDA is less reproducible than DIA mainly reflected in the identification results. MSCohort includes the scoring formulas for both individual DDA and DIA experiments (see **Methods** section and **Reviewer#1 Reply 1.4** for details):

The DDA scoring formula is expressed as:

$$N_{identified_precursors} = N_{acquired_MS2} \times Q_{MS2} \times P_{MS2_per_precursor} \quad (1)$$

The DIA scoring formula is expressed as:

$$N_{identified_precursors} = N_{acquired_MS2} \times Q_{MS2} \times (N_{precursor_per_MS2}/R_{precursor}) \quad (2)$$

The value of MSCohort is its ability to provide a very intuitive explanation using the aforementioned formulas as to why the identification results of DDA are less reproducible than DIA. We provide the detailed metrics results of DDA and DIA experiments in **Supplementary Data 1 and Supplementary Table 4**. Taking the identification results from the 7 Orbitrap instrument platforms (U01-U07) with LC-SOP as an example, due to stochastic MS2 sampling of DDA, there is a significant difference in the parameter $N_{acquired_MS2}$ and Q_{MS2} in DDA data in the above formulas, where the number of acquired MS2 spectra in DDA data ranges from 23098 to 32258 (relative standard deviation (RSD) was 13.3%), the identification rate of MS2 scans in DDA data ranges from 33.64% to 56.41% (RSD was 17.1%). While in DIA data, the number of acquired MS2 spectra ranges from 20240 to 23400 (RSD was 6.4%) and the identification rate of MS2 scans ranges from 66.1% to 74.2% (RSD was 4%). Although other metrics also exhibit variances, they are not as pronounced.

Without comprehensive metrics, scoring formulas, and fully automated analysis tools like MSCohort, analyzing why the identification results of DDA are less reproducible than DIA would be challenging.

1.10 in the section starting from line 262: again, what really matters here is the added value of MSCohort, unless the authors aimed to only show here that MSCohort works in three types of MS instruments.

Reply:

Thanks for the reviewer's good suggestion. In the "Performance evaluation of proteome quantification and detection of differentially expressed proteins from multi-platform study" section of the **Results**, the purpose of the experiments was to evaluate the quantitative accuracy, precision, and sensitivity of urinary proteome from different platforms under the unified SOP.

To achieve this objective, it is essential to ensure the quality of the collected data. This includes the data achieved good identification depth in individual experiments, adequate numbers of acquired spectra, high identification rate and utilization rate of MS2 scans, etc. Additionally, it is crucial to achieve high reproducibility across multiple experiments, including stable chromatographic retention times and high reproducibility in mass spectrometry quantification results, etc. All of these aspects could be directly analyzed through **MSCohort**. Without MSCohort, one would be limited to a superficial and manual examination of metrics, which may not provide a comprehensive quality control assessment.

Furthermore, supporting the analysis of data from three different instrument platforms is also an important feature of MSCohort.

1.11 line 294+: It is unclear how the DEPs analysis contributes to the added value of MSCohort.

Reply:

Thanks for the reviewer's good suggestion. As mentioned in **Reply 1.10 to Reviewer #1**, our primary objective in utilizing a benchmarking dataset with a mixture of species was to validate the quantitative performance across different instrument platforms and to confirm the accuracy in identifying biomarkers under our established Standard Operating Procedure (SOP). Therefore, the analysis of differentially expressed proteins (DEPs) was primarily conducted to assess the sensitivity and specificity of different LC-MS platforms in DEP detection under the unified SOP and the supervision of the MSCohort QC system.

In response to **Reviewer #1 Reply 1.13**, we also provided a detailed demonstration of the value of MSCohort in identifying biomarkers in actual cohort experiments.

1.12 line 318: "large cohort" -- a total of 160 samples may not well justify a large cohort.

Reply:

Thanks for the reviewer's good suggestion. As noted in **Reply 1.5 for Reviewer #1**, there is currently no clear definition of the number of large cohort samples. In these large cohort proteomic studies, the sample size is mainly between 50 and 500, with a median of 119 (**For-review Figure 3**).

According to your suggestion, to avoid ambiguity, we have revised this sentence, replacing "large cohort" with "cohort" (**Page 11, Line 356**).

1.13 in the last section of the CRC urine, the authors are commended for the comprehensive analysis, but it is unclear to the readers how MSCohort was applied, and what are the added value of MSCohort.

Reply:

Thanks for the reviewer's valuable suggestion. As mentioned in our reply for **Reviewer#1 Reply 1.5** above, we detailed four application scenarios in which MSCohort is required. In this section, the application scenarios (ii), (iii), and (iv) were performed. Our detailed response is as follows:

In the previous version manuscript, we described the value and quality control analysis results of MSCohort in the CRC urine proteome cohort, including detecting low-quality experiments and indicating the reasons for identifying these experiments as anomalies. According to the reviewer's suggestion, we added the detailed description about how MSCohort was applied and what are the added value of MSCohort in the CRC urine proteome cohort in the revised manuscript. The sentences are shown as follows:

(Page 12, Line 366-395) "In the process of cohort experiments collection, an intra-experiment analysis based on the MSCohort QC system was performed for each newly collected experiment to evaluate the individual data quality. After the number of experiments collected exceeded 2 runs, an inter-experiment analysis based on the MSCohort QC system was performed to evaluate the stability and reproducibility of the instrument system. Finally, after all the samples had been collected on one instrument platform, inter-experiment analysis based on the MSCohort QC system was performed to evaluate cohort data quality and detect low-quality experiments.

This cohort experiment is a time course study with at least five consecutive days of acquisition on each instrument platform. We analyzed the overall cohort data quality based on the MSCohort QC system. First, the QC samples demonstrated good technical repeatability, with median Pearson correlation > 0.94 for each of the 3 LC-MS platforms (Supplementary Fig.9a), indicating good LC-MS system stability. The results showed that the overall chromatographic retention time was stable (the average retention time deviation < 0.25 min) for 7 consecutive days (Figure S.Note 4.2 e). In addition, MSCohort detected and reported low-performance experiments based on the isolation forest algorithm, as shown in Supplementary Fig.9b, and 8 low-quality samples were reported in TIMS. Among them, 7 and 6 samples were also reported in Lumos and E480, respectively (Supplementary Data 3). The corresponding heatmap in the MSCohort report indicated that there were significant differences between these 8 samples and other samples in multiple inter-experiment metrics (at least 7 of 23 metrics showed a variation of more than two standard deviations (SD) from its median). Among them, 5 samples (D943, D1116, D1412, H771, D994) showed higher ratios of contaminants (erythrocytes, cellular debris, or serum high abundance proteins) than the other samples, which resulted in the sample-to-sample variability compared to regular urinary proteins (Methods). Another 3 samples (H349, D1036, D1069) showed lower Pearson correlation, and higher robust standard deviation at precursor, peptide, and protein groups intensity with the other samples, indicating these samples were heterogeneous compared with other samples (Supplementary Fig.9c-e). The detailed MSCohort report for each platform were

shown in Supplementary Data 3. Thus, these 8 experiments were excluded from further analysis (Supplementary Fig.9b).”

Some other big picture issues:

1.14 is MSCohort specific to analyzing urine samples? Has it been optimized for urine samples? It is understood that most of the samples used in this study are from urine. But are there any specific QC metrics or algorithm customized for urine proteome?

Reply:

Thanks for the reviewer’s professional advice, and we answered the questions point by point as follows:

(1) is MSCohort specific to analyzing urine samples?

As mentioned in the response to **Reviewer #1 Reply 1.1**, MSCohort is a universal software tool. MSCohort supports comprehensive quality control for different sample types (e.g. cell, tissue, plasma, urine, etc.), different instrument types (Thermo Scientific Orbitrap, Bruker timsTOF, and SCIEX ZenoTOF), and different acquisition modes (DDA, DIA, PRM). MSCohort showed a powerful generalization ability. It can apply not only to process evaluation and optimization of individual experiments, but also to process quantitative assessment of system performance across multiple experiments.

In this study, we primarily apply MSCohort to optimize and establish the Standard Operating Procedure (SOP) for urinary proteomics and to perform quality evaluation on the consistency and reproducibility of urinary proteomics across multiple LC-MS platforms.

In our other work, we also apply MSCohort to perform quality control for other types of samples (cell, tissue, plasma, etc.).

(2) Has it been optimized for urine samples?

MSCohort is a universal software tool, which supports comprehensive quality control for different sample types (e.g. cell, tissue, plasma, urine, etc.). In this study, we optimized and established an SOP for urinary proteome based on MSCohort (see “Supplementary Note 4: Optimization of LC-MS Method for Establishment of SOP for Urinary Proteomics” for details).

(3) are there any specific QC metrics or algorithm customized for urine proteome?

As mentioned in the response to **Reviewer #1 Reply 1.1**, in MSCohort, we integrate the metrics reported in the existing quality control tools and add 26 new quality control metrics, generating a total of 81 metrics.

In this study, we added 3 contaminants metrics, including erythrocytes, cellular debris, and serum proteins, for monitoring the quality and level of contaminants in urine samples. These 3 contaminants were previously reported as urine-specific quality marker panels to assess the degree of contamination of the urine samples (*EMBO Mol Med*, 2021, 13(3):e13257; *BMC Nephrol*, 2018, 19(1):358). A detailed explanation of these three sample contaminants metrics can be found in the **Methods** section (**Page 26-27, Line 879-897**).

According to the literature (*EMBO Mol Med*, 2019, 11(11): e10427; *Cell Rep Med*, 2022, 3(6):100661), these three contaminants quality marker panels can also be used for quality control of body fluid samples such as plasma and cerebrospinal fluid.

Users can customize and edit the contaminants list based on the actual sample situation. Detailed instructions for making these modifications are provided in the user manual of MSCohort (<https://github.com/BUAA-LiuLab/MSCOhort>).

All the other 78 metrics are universal and can be applied to different types of samples, such as cells, tissues, and so on. The algorithms in MSCohort are all generic and do not include specific algorithms tailored for urinary proteomics.

Minor issues:

1.15 In supplementary Figure 1, "median total ion current (TIC)" is mentioned, whereas supplementary table 1 lists "median total ion chromatography (TIC)". Correct one should be used.

Reply:

Thanks for the reviewer's detailed inspection. In the revised manuscript, we revised "median total ion chromatography (TIC)" to "median total ion current (TIC)" in **Supplementary Table 1A**.

1.16 Descriptions of the 23 inter-experiment features in the appendix are confusing.

Reply:

Thanks for the reviewer's helpful suggestion. In the revised manuscript, we have revised the descriptions of the 23 inter-experiment metrics in revised **Supplementary Table 1B**, including their specific meanings and extraction processes.

1.17 Tables in the appendix are inconsistently formatted with colons and periods, requiring standardization.

Reply:

Thanks for the reviewer's detailed inspection. In the revised manuscript, we have standardized the formatting of all tables in the appendix by using periods consistently.

Reviewer #2 (Remarks to the Author):

As the authors describe, there is a significant need to implement the standardization of large-scale urinary proteomic analyses, particularly those with clinical relevance. To systematically assess data quality from large-scale urinary proteomics datasets, Liu et al. developed a quality control (QC) system named MSCohort, which evaluates 70 metrics. A total of 20 LC-MS platforms were evaluated using pooled urine samples; standard mixtures of *E. coli*, yeast, and HEK293 lysate; and urine from healthy controls and patients with colorectal cancer. The evaluated MS systems include Orbitrap, TimsTOF, ZenoTOF, Orbitrap Fusion Lumos, and Exploris 480. The evaluated LC systems included EASY-nLC 1000, Vanquish Neo UHPLC, and Ultimate 3000 UHPLC.

The current study falls short of its stated goals in three main areas: 1) the lack of clarity regarding the establishment of the acceptability criteria for the 70 QC metrics included in MSCohort, 2) a less-than optimal experimental design wherein the same LC system was not evaluated in combination with each MS system, and 3) the lack of an explanation of the scores assigned to the QC metrics included in MSCohort, which are presented on a scale from 1–5 with “5 points being excellent and 1 point indicating plenty of room for improvement”. Consequently, this reviewer is unable to recommend the manuscript for publication in its current state. Several issues should be addressed:

Reply to general concerns:

We thank the reviewer for the valuable suggestions, which help us improve our manuscript, and we answered the questions point by point as follows:

(1) the lack of clarity regarding the establishment of the acceptability criteria for the 70 QC metrics included in MSCohort.

1) In the revised manuscript, we comprehensively analyzed and integrated the metrics proposed by existing quality control software. Building upon the initial 70 metrics in the previous manuscript, we included an additional 11 intra-experiment metrics from existing software, generating a total of 81 metrics. We attempt to provide the most comprehensive set of QC metrics for proteomics. In **Supplementary Note 1**, we provided a detailed comparison between the metrics extracted in MSCohort and those of existing tools. We also provide a detailed introduction to all 81 metrics in MSCohort in **Supplementary Table 1A and Table 1B**, including their specific meanings and extraction processes.

2) We acknowledge that the establishment of acceptability criteria is very challenging. We have designed a Metric-Score diagram (**Supplementary Figure 1a**) to illustrate the performance of various metrics in the workflow. In this diagram, each grid represents a metric which is distinguished by score, and we use different colors to indicate different scores (For example, 5 points is shown in green and 1 point is shown in red). We use this graph to help users evaluate experimental performance and identify which metrics influenced the identification results.

3) In our response to the question raised in point (3) regarding “**the lack of an explanation of the scores assigned to the QC metrics included in MSCohort, which are presented on a scale from 1–5 with “5 points being excellent and 1 point indicating plenty of room for improvement”**”, we provided an explanation of the scores assigned to the QC metrics.

(2) a less-than optimal experimental design wherein the same LC system was not evaluated in combination with each MS system.

In our developed SOP, each laboratory should use the same LC system, regardless of the MS system it is combined with. The same LC system includes parameters such as unified and easily obtainable chromatographic columns (commercially available), as well as unified parameters that can be set on mainstream chromatographs, such as mobile phase gradient conditions, flow rate, etc. Regardless of the model of the chromatograph used, as long as the parameters are set the same and a unified chromatographic column is used, we consider it to be the same LC system.

In practice, in our **Results** section titled “Comprehensive and comparative analysis of urinary proteome data from multi-platform study”, we indeed employed **different MS system combinations with the same LC system** strategy in with LC-SOP group (see **Supplementary Table 2** for details). The results demonstrate the qualitative and quantitative results showed higher consistency and reproducibility among the 10 LC-MS platforms with LC-SOP (see Figure 4 for details). Namely, **the same LC system WAS evaluated in combination with each MS system.**

Previous published studies have analyzed the reproducibility of DIA data generated across multiple laboratories using the same LC system and the same model of MS instrument platforms (*Nat Commun*, 2017, 8(1): 291; *Nat Commun*, 2020, 11(1): 5248), demonstrating high inter-laboratory consistency and reproducibility under the same LC and MS systems.

Our research has demonstrated that under a unified SOP, high reproducible results can also be achieved with the same LC system in combination with different types of MS. This extends the findings of previously published studies (*Nat Commun*, 2017, 8(1): 291; *Nat Commun*, 2020, 11(1): 5248), which employed the same LC and MS systems. We have included the relevant discussion in the manuscript (**Page 16, Line 511-528**).

(3) the lack of an explanation of the scores assigned to the QC metrics included in MSCohort, which are presented on a scale from 1–5 with “5 points being excellent and 1 point indicating plenty of room for improvement”.

1) The scoring system for the quality control (QC) metrics included in MSCohort, which ranges from 1 to 5 points, with 5 points representing excellent performance and 1 point indicating significant room for improvement, is based on our previously published MSRefine (*Tang, et al. Anal Chem* 2023, 95(20):7897-7905). We acknowledge that establishing standardized scores for QC metrics is a highly challenging task.

Currently, other quality control software tools for evaluating and optimizing individual experiments primarily extract a limited number of metrics and present them in graphical form, without quantifying or scoring these metrics. This approach can be inconvenient for users seeking a comprehensive evaluation of data quality. Therefore, we aim to provide a quantitative quality assessment that standardize the types of metrics with different value intervals.

Currently, user-defined metric thresholds are necessary to fully leverage the software's capabilities. We have established scoring standards based on the actual performance of the instrument and observable values from practical experiments. For example, consider the metric “M12”, which is the median of MS1 raw mass accuracy (the median of the delta mass between the monoisotopic theoretical and measured m/z of precursors). For the Thermo Orbitrap instrument, a median MS1 raw mass accuracy of ≤ 1 ppm earns 5 points (excellent), reflecting outstanding performance that is difficult to further improve upon. Conversely, a median MS1 raw mass accuracy of ≥ 5 ppm receives 1 point (significant room for improvement). For values falling between >1 ppm and <5 ppm, a linear scoring algorithm is applied. Notably, setting a threshold of ≤ 0.1 ppm for 5 points is not practical because current instrument performance rarely achieves this standard. Similarly, designating ≥ 20 ppm as 1 point is also impractical, as the performance of existing instruments (such as Orbitrap, but not ion trap) typically does not degrade to such levels.

2) In addition, users can also adjust scoring standards based on actual conditions. We anticipate that advancements in instrument performance will necessitate adjustments to the scoring standards for the Median of MS1 raw mass accuracy. In MSCohort software, users can adjust scoring standards in MSCohort_exe\ini_DIA file as shown in **For-review Figure 5**. We also provided the detailed instructions for users to adjust the scoring standards in the user manual of MSCohort (<https://github.com/BUAA-LiuLab/MSCohort>).

In accordance with your suggestions, we have incorporated a detailed explanation of the scoring system employed for the quality control metrics integrated in MSCohort in the **Methods** section (**Page 25, Line 818-844**).

1. Modifying the scoring standards for intra-experiment analysis

For-review Figure 5. The screenshots of explanation for users to adjust the scoring standards in MSCohort software.

2.1 One of the paper’s main premises is that the use of SOPs improves the data reproducibility and overall quality. This is as to be expected and is not a new observation originating from the current study that warrants any significant explanation.

Reply:

Thanks for the reviewer's comments. We acknowledge that the use of SOPs improves the data reproducibility and overall quality. It is precisely for this reason that we aimed to optimize and establish an SOP for urinary proteomics, which was previously nonexistent. Furthermore, in the revised manuscript, we added a new section in the **Results** titled “Optimization and establishment of SOP for urinary proteomics with MSCohort QC system”, which illustrated the optimization and establishment process of our SOP, along with its advantages. Additionally, we provided and revised the detailed SOP in **Supplementary Information**: “Supplementary Note 3: The Standard Operating Procedure (SOP) for Urinary Proteomics”. We also added two additional notes related to the optimization and establishment of SOP in the **Supplementary Information**: “Supplementary Note 4: Optimization of LC-MS Method for Establishment of SOP for Urinary Proteomics” and “Supplementary Note 5: The Novelty of Our Developed Standard Operating Procedure (SOP) for Urinary Proteomics”. These notes primarily explain the process of optimizing SOP parameters using the MSCohort QC system and the novelty of our SOP for urinary proteomics.

2.2 Lines 82-84: “Increasing numbers of urinary proteomic studies have been used to discover potential disease biomarkers, such as urological cancers, colorectal cancer, virus infection, neurodegenerative

disorders, and many others.” This sentence should be re-worded to avoid the current, incorrect implication that “urological cancers, colorectal cancer”, etc. are biomarkers.

Reply:

Thanks for the reviewer’s detailed inspection. In the revised manuscript, we have revised this sentence as “Increasing numbers of urinary proteomic studies have been used to discover potential biomarkers across various diseases, such as urological cancers, colorectal cancer, virus infection, neurodegenerative disorders, and many others.” (Page 4, Line 86-89)

2.3 Lines 182 – 187 and Fig. 1: It does not appear that the same 10 LC-MS platforms were used with and without an LC SOP; therefore it is not accurate to make conclusions regarding the role of the LC SOP in contributing to the data quality.

Reply:

Thanks for the reviewer’s professional advice. We agree with the reviewer’s opinion that the same 10 LC-MS platforms were used with and without an LC-SOP would be the optimal experimental design to make conclusions. In the same platforms, the results are better when using an SOP compared to not using an SOP, which is easily understandable and has been validated (*J Proteome Res*, 2010, 9(2):761-776; *Anal Chem*, 2014, 86(5), 2497-2509). In our study, we divided 20 LC-MS platforms into two groups: one group of 10 platforms adopted the unified SOP, and the other group of 10 platforms adopted their individual optimized experimental parameters without SOP. Our experiment aimed to demonstrate that significant differences arise when each LC-MS platform applies its own protocol, highlighting the necessity of establishing a standardized SOP. Our results showed higher qualitative and quantitative results consistency and reproducibility among the 10 LC-MS platforms with LC-SOP, demonstrating the role of SOP in promoting data consistency and reproducibility across different LC-MS platforms.

2.4 Lines 196 – 197: This is the only place where FAIMS is mentioned throughout the entire manuscript. It is unclear whether the instrument performance comparisons were conducted separately for the platforms with and without FAIMS. One would expect the use of this ion separation method to significantly influence several of the MS-specific metrics that are included within MSCohort.

Reply:

Thanks for the reviewer’s valuable suggestion. According to the reviewer's suggestion, we conducted the instrument performance comparisons using data collected from the same LC-MS platform (Orbitrap Exploris 480) with and without FAIMS. Under the with FAIMS condition, we set individual CVs at -45V, -65V, and combined different CVs: -45V / -65V. Additionally, we included a comparison without FAIMS, resulting in a total of four groups for analysis.

The instrument performance comparisons results were showed in **For-review Figure 6**. The MSCohort software comprehensively extracted quality control metrics mapping to the whole LC-MS workflow and

established a scoring formula, facilitating a quick and comprehensive analysis of instrument performance comparisons under both with and without FAIMS conditions. According to the metrics extracted by MSCohort, the primary differences between with and without FAIMS conditions were observed in Total Ion Current (TIC) intensity (**For-review Figure 6a**), the number of identified precursors and peptides (**For-review Figure 6b**), the number of identified protein groups (**For-review Figure 6c**), and MS1 and MS2 peak intensities (**For-review Figure 6d**). FAIMS as an ion selection device and an electrospray filter that prevents neutrals from entering the orifice of the mass spectrometer while reducing chemical background noise (*Bekker-Jensen, et al. Mol Cell Proteomics, 2020, 19(4):716-729*). While under the same sampling condition, it also results in lower TIC intensity, lower MS1 and MS2 peak intensities, and lower MS1 and MS2 peaks number in with FAIMS condition. Our metrics results and analysis reports indicate that due to the lower peak intensities and peak numbers obtained with FAIMS, the number of acquired MS2 scans, the identification rate and the utilization rate of MS2 scans were significantly lower compared to without FAIMS. Consequently, the identified number of precursors and protein groups was lower with FAIMS.

Previous study also reported that FAIMS enabled experiments tend to be more sensitive to sample amounts and lower TIC (*Hebert, et al. Anal Chem, 2018, 90(15):9529-9537*). “The success of the FAIMS setting is highly dependent on the general performance of the instrument, as any loss in sensitivity will immediately result in a dramatic decrease in protein identification. It only works well when the total ion flux in MS/MS mode is high as measured by total ion current chromatogram (TIC) distributions plot” (*Bekker-Jensen, et al. Mol Cell Proteomics, 2020, 19(4):716-729*).

For-review Figure 6. The instrument performance comparisons for the platforms with and without FAIMS. Four group comparisons were conducted: withoutFAIMS, withFAIMS combined CVs: -45V/-65V, withFAIMS single CV=-45, withFAIMS single CV=-65. **a.** The Total Ion Current (TIC) intensity. **b.** The number of acquired MS2 scans, identified MS2 scans, identified precursors, and identified peptides. **c.** The number of identified protein groups. **d.** The mainly differential metrics in the MSCohort results.

2.5 Lines 228 – 237 and other locations throughout the manuscript where protein identification data is mentioned: The authors do not provide any critical insight or interpretation of the properties of the proteins that were identified across all LC-MS platforms. For example, it would be informative to know the average sequence coverage of these proteins, in addition to their relative abundance. What was the dynamic range of these proteins?

Reply:

Thanks for the reviewer's valuable suggestion. According to your suggestion, we added the analysis of the protein sequence coverage, the relative abundance, and the dynamic range in the revised **Supplementary Figure 5** to provide more information.

Among the 10 LC-MS platforms without LC-SOP (M01-M10), the median protein sequence coverage was 7.58% (ranging from 6.37% - 9.10%), and the dynamic range of the quantified proteins was about 8 orders. Among the 10 LC-MS platforms with LC-SOP (U01-U10), the median protein sequence coverage was 8.5% (ranging from 8.07% - 9.67%), and the dynamic range of the quantified proteins was about 8 orders. The results demonstrated that the average sequence coverage and the relative abundance of the proteins that were identified among the 10 LC-MS platforms with LC-SOP showed a higher consistency than that in 10 LC-MS platforms without LC-SOP. We also added the above analysis results in the revised manuscript (**Page 9, Line 286-289**).

Supplementary Figure 5. The analysis of the protein sequence coverage, the relative abundance, and the dynamic range across 20 LC-MS platforms. **a** and **b** The protein sequence coverage for the 10 platforms without LC-SOP (**a**, orange) and with LC-SOP (**b**, blue). The vertical black line represents the median of protein sequence coverage, and the median value was listed. **c** and **d** The relative abundance of Log 10 protein intensity for the 10 platforms without LC-SOP (**c**, orange) and with LC-SOP (**d**, blue). **e** and **f** The dynamic range of quantified protein intensity for the 10 platforms without LC-SOP and with LC-SOP.

2.6 For the instances when any of the QC metrics evaluated by MSCohort are not within the stated acceptability limits/scores, the authors do not provide any guidance regarding whether these scores are indicative of random or systematic errors/variation.

Reply:

Thanks for the reviewer's valuable suggestion. In Supplementary Table 1A, we have added a detailed explanation and purpose/use of the 81 QC metrics in MSCohort, along with the optimization directions when

a QC metric falls outside the acceptable range.

In MSCohort, each score is directly associated with the values of metrics. For example, consider the metric “M12”, which is the median of MS1 raw mass accuracy (the median of the delta mass between the monoisotopic theoretical and measured m/z of precursors). For the Thermo Orbitrap instrument, a median value ≤ 1 ppm earns 5 points, whereas ≥ 5 ppm results in 1 point. For values between > 1 ppm and < 5 ppm, a linear scoring algorithm is applied. Specifically, minor deviations from 1 ppm to 2 ppm result in a score change from 5 points to 4 points, which is generally considered acceptable. However, a deviation reaching 5 ppm indicates a significant issue, earning a score of 1 point and suggesting the need for instrument correction, which means these scores are not indicative of random or systematic errors/variation.

During the experiment data collection process, if any QC metric exceeds the acceptable range, the initial step is to replicate the experiment to verify the reproducibility of the results. If the outlier values persist, they may be indicative of systematic issues. In such cases, targeted adjustments to the instrument system should be made, based on the specific metrics with lower scores.

2.7 Line 437: “...we should pay attention to these metrics when conducting urine proteomics experiments”. Such language is overly vague and does not provide any constructive guidance for readers.

Reply:

Thanks for the reviewer’s suggestions. We have revised the sentence in the **Discussion** section and added more information as follows:

(Page 15, Line 482-492) “The unified urinary proteome SOP was developed based on the MSCohort QC system and applied in multiple laboratories. Analysis results from 20 LC-MS platforms demonstrated the necessity of establishing the SOP. Meanwhile, results from 10 platforms without SOP indicated that these metrics showed lower scores, including the identification rate of MS2 scans, the utilization rate of MS2 scans, MS2 peak numbers, MS2 peak intensities, and chromatographic invalid acquiring time, etc. in experiments with fewer identification results (Supplementary Fig.3), which also indicated that we should pay attention to above these metrics when conducting urinary proteomics experiments. In particular, the identification rate of MS2 scans and the utilization rate of MS2 scans in the DIA scoring formula play an important role in evaluation and optimization for individual experiments.”

2.8 Line 162, 336, 810 & 814: The description of “contaminants” as a components of the “whole LC-MS workflow evaluation” in MSCohort should be clarified. Erythrocyte contaminants are mentioned in Line 814 in the context of erythrocyte lysis that occurs during the preparation of plasma; however, it is unclear how these type of contaminants would be relevant to the evaluation of other sample types.

Reply:

Thanks for the reviewer’s careful review. In the revised manuscript, we made modifications for the

description of contaminants:

(Page 27, Line 880-900) “Pre-analytical variation caused by contaminations during sample collection or inconsistent sample processing, can have an impact on the results and may cause the reporting of incorrect biomarkers. MSCohort offers configurable lists of custom protein contaminants to help users assess each sample for potential quality issues.

For urine sample quality control, we used three urine-specific quality marker panel to assess the degree of contamination of the samples. Firstly, we used two previously reported quality marker panels to determine the degree of contamination with erythrocytes and cellular debris. Contamination of erythrocytes occurs during urine collection due to hematuria or hemolysis caused by kidney function issues or systemic disorders, leading to a high sample-to-sample variability compared to regularly secreted urinary proteins. Insufficient removal of cells and cellular debris from urine will lead to increased detection of intracellular proteins with a high sample-to-sample variability compared to regularly secreted urinary proteins. In addition, proteinuria occurs due to abnormalities in kidney function or systemic disorders, resulting in the leakage of serum proteins into the urine. This can lead to increased detection of serum proteins with high sample-to-sample variability. We generate the third urine-specific quality marker panel to asses of contamination with serum high abundant protein.

According to the literature (EMBO Mol Med, 2019, 11(11): e10427; Cell Rep Med, 2022, 3(6):100661), these three contaminants quality marker panels can also be used for quality control of body fluid samples such as blood and cerebrospinal fluid.”

2.9 Considering the potential broader/”real world” applicability of MSCohort, what is the minimum number of LC-MS data files that can be analyzed by MSCohort given that intra- and inter-experiment parameters are evaluated? For the determination of intra-experiment metrics, it would seem that technical replicates are required.

Reply:

Thanks for the reviewer’s valuable suggestion. We advocate for the utilization of MSCohort in the following four application scenarios (also mentioned in our reply for Reviewer#1 Reply 1.5):

(i) Determining the appropriate experimental parameters for individual experiments to achieve deep coverage. (ii) Evaluating the instantly completed experiments during the cohort experiments collection process. (iii) Assessing the stability and reproducibility of the LC-MS systems once the number of collected experiments exceeds 2 runs during the cohort experiments collection process. (iv) Conducting cohort data quality evaluation and detecting potential outlier experiments upon completion of the cohort experiment collection.

In the revised manuscript, we presented in Supplementary Note 3 and Note 4 how to use MSCohort for optimization and deeper proteome profiling for individual experiments in application scenarios (i). In “Supplementary Note 4: Optimization of LC-MS Method for Establishment of SOP for Urinary Proteomics”,

we tested 12 sets of LC-MS parameters, with each set repeated three times. That is to say, we agree that technical replicates are required for the determination of intra-experiment metrics.

To evaluate inter-experiment parameters across more than two LC-MS runs, users can benefit from the MSCohort inter-experiment analysis, which evaluates system stability across all experiments, regardless of their number. In the analysis of a dataset of 20 LC-MS urinary proteomics data, we showed in Supplementary Figure 3 how to conduct comprehensive quality assessment of individual experiments using MSCohort and identify the causes of experimental issues in application scenarios (ii); In the analysis of the benchmarking hybrid dataset, we illustrate in Supplementary Figure 7 how to perform comprehensive quality assessment between multiple experiments, evaluating instrument system stability and reproducibility using MSCohort in application scenarios (iii); In the analysis of the CRC cohort dataset, we demonstrate in Supplementary Figure 9 how to conduct comprehensive quality assessment across cohort experiments, evaluating instrument system stability and detecting outlier experiments using MSCohort in application scenarios (iv).

2.10 Figure 2: The “Hours” label on the far left side of the figure does not provide any useful information.

Reply:

Thanks for the reviewer’s detailed inspection. We have removed the “Hours” label and changed it to “LC-MS Workflow” in revised **Figure 2**.

2.11 Figure 3: E240 is listed as one of the LC-MS platforms. Presumably, this refers to the Exploris 240; however, this instrument is only mentioned in one other location in the manuscript – line 208: “M09-E240” – again, presuming that E240 refers to the Exploris 240.

Reply:

Thanks for the reviewer’s detailed inspection. In the previous manuscript, we listed the full names corresponding to the abbreviations of all instruments in Supplementary Table 2. According to your suggestion, for better clarity, we have added the explanations for the abbreviations in the manuscript (**Page 9, Line 259**). Additionally, in the revised **Supplementary Table 2A**, we have added an overview of the LC-MS conditions for 20 LC-MS platforms.

2.12 Figure 4 title: “sample-A and sample-B”. These names are too vague and should be explained in more detail.

Reply:

Thanks for the reviewer’s careful review. In the revised legend of **Figure 5**, we have provided an explanation for the sample naming and mixing proportions of sample-A and sample-B, which were referenced from previous studies (*Nat Biotechnol*, 2016, 34(11):1130-1136; *Nat Commun*, 2020, 11(1):5248; *Nat Biotechnol*, 2023, 41(1):50-59).

“Figure 5. Overall quantitative performance of benchmarking sample-A and sample-B from 3 LC-MS

platforms. Benchmarking samples A and B were prepared containing known ratios of peptide digestions from human, yeast and *E. coli* organisms, resulting in expected peptide and protein ratios of 1:1 (A/B) for human, 1:2 for yeast and 4:1 for *E. coli* proteins. The samples A and B were analyzed in three technical replicates on three LC-MS platforms Orbitrap Fusion Lumos, Orbitrap Exploris 480, and timsTOF Pro 2 (Lumos, E480, and TIMS for short), respectively.”

2.13 Figure 4 legend: Instead of describing the lengths of the whiskers on the box plots as “1.5 box lengths away from the box ends”, it would be more appropriate to describe the length of the whiskers as 1.5-times the interquartile range.

Reply:

Thanks for the reviewer’s detailed inspection. We have changed the description in Figure 4 legend as “1.5-times the interquartile range” in revised Figure 5.

Reviewer #3 (Remarks to the Author):

Reply:

Thank you and we are very glad to get comments and feedbacks from Early Career Researchers.

Reviewer #4 (Remarks to the Author):

The manuscript introduces a comprehensive quality control (QC) metric system for evaluating the performance and validity of large-scale MS-based proteomics analysis of human urine samples. The QC metrics were applied to human urine proteomics data from different platforms to illustrate the impact of implementing the QC metric system. The experimental design and data provided for three MS platforms do not align with the objective of the manuscript to provide a QC metric and reference standard operating procedure for a "large-scale cohort" for urine proteomics. The manuscript also proposes "a comprehensive QC system for systematic and measurable quality assessment in large cohort LC-MS analysis over time and across instruments" (pg 6 lines 171-173); however, the experimental design or data do not include information about a time course study (short- or long-term). The manuscript suggests that the QC metrics provide "optimizing strategies" (pg 4, lines 107-108, pg 5, lines 134-135, 141-142); however, the author does not demonstrate how the optimization strategies for experimental workflows are provided in the QC metric output without relying on "expert expertise to optimize parameters and locate experimental problems" (pg 4 lines 102-104). The discrepancy between the study objective and the presented data/results affects the significance of the manuscript. I do not recommend this manuscript for publication in its current state unless major revisions are made to the study objective, data presentation, format, and manuscript content. The manuscript and supplemental materials contain a substantial amount of data to demonstrate the performance of the QC metrics. Below are additional concerns with the manuscript:

Reply to general concerns:

We thank the reviewer for the useful comments and pointing out the problems and putting forward suggestions, which help us improve our manuscript. And we answered the questions point by point as follows:

(1) The experimental design and data provided for three MS platforms do not align with the objective of the manuscript to provide a QC metric and reference standard operating procedure for a "large-scale cohort" for urine proteomics.

To demonstrate the substantial benefits of the introduced MSCohort and SOP in advancing urinary proteomics for clinical disease research, we applied these to hybrid benchmarking samples and conducted a comprehensive urinary proteome analysis of clinical colorectal cancer (CRC) across three different platforms. In the revised manuscript, we have enriched the **Results** "Analyses of Cohort Clinical Proteomics from Multi-Platform Study" section with detailed descriptions, showing how the MSCohort QC system was integrated into the CRC urinary proteome analysis. Notably, our findings demonstrate that approximately 7000 protein groups could be reliably identified, and a high degree of quantitative stability was achieved in the urinary proteome analysis, adhering to our established SOP.

(2) the experimental design or data do not include information about a time course study (short- or

long-term).

We think the reviewer's mention of a “time course study” aligns with our inter-experiment analysis across multiple experiments. In the Results “**Analyses of cohort clinical proteomics from multi-platform study**” section, we conducted a time course study involving 160 samples. Data collection was performed on three instruments (Lumos, E480, and TIMS) over a short-term period spanning at least 5 consecutive days on each instrument. In the process of cohort experiments collection, we conducted the data quality control using MSCohort.

In the process of cohort experiments collection, an intra-experiment analysis based on MSCohort QC system was performed for each newly collected experiment to evaluate the individual data quality. After the number of experiments collected exceeded 2 runs, an inter-experiment analysis based on MSCohort QC system was performed to evaluate the stability and reproducibility of the instrument system. Finally, after all the samples have been collected on one instrument platform, inter-experiment analysis based on MSCohort QC system was performed to evaluate cohort data quality and detect low-quality experiments.

In MSCohort software, **FLAG_SHOW_ORDER** set as **time series (For-review Figure 7)** to display the result of the time-line report.

For-review Figure 7. The screenshots of explanation for users to set the time-line report in MSCohort software.

(3) The manuscript suggests that the QC metrics provide "optimizing strategies" (pg 4, lines 107-108, pg 5, lines 134-135, 141-142); however, the author does not demonstrate how the optimization strategies for experimental workflows are provided in the QC metric output without relying on "expert expertise to optimize parameters and locate experimental problems" (pg 4 lines 102-104).

It is important to note that expert experience is indispensable in the process of optimizing experimental parameters. But in previous proteomics LC-MS method optimization, relying solely on expert experience can only manually extract a limited number of metrics, and require a large number of parameter arrangement and combination experiments to determine the optimal parameters.

With MSCohort providing a formula that establishes a mathematical relationship between quality control metrics and identification results, it provides direct explanations for the underlying causes of data results. Furthermore, we have generated a metric-score diagram to flag (with red color) the metrics with low scores to assist experimenters in assessing the quality of data directly (**Supplementary Figure 1a, Supplementary Data 1**). Users can determine appropriate parameter conditions through a limited number of experiments and figure out whether additional experiments are needed based on MSCohort's reports, facilitating quick optimization of experimental parameters and the establishment of an SOP.

In the revised manuscript, we added a new note in **Supplementary Information** to explain this question: "Supplementary Note 4: Optimization of LC-MS Method for Establishment of SOP for Urinary Proteomics" showed how the optimization strategies for experimental workflows with the MSCohort report include QC metrics output to optimize parameters and locate experimental problems.

4.1 Pg. 4 lines 88-89 – sentence structure issue

Reply:

We have changed the sentence in the revised manuscript.

4.2 Reference not included for DIA data studies - Pg. 4 lines 89-92

Reply:

We have added the references in the revised manuscript.

4.3 Pg. 4 line 94 – change "body" to "bodily"

Reply:

We have corrected the sentence in the revised manuscript.

4.4 Pg. 4 line 99 – add "a" before SOP

Reply:

We have corrected the sentence in the revised manuscript.

4.5 Pg. 4 lines 102-104: The experimental design and rationale are comparable to the work published by Rudick PA et al. Mol Cell Proteomics 9, 225-41 (2010), as referenced in the manuscript. The application of LC-MS/MS QC metrics for the optimization of a proteomics workflow proposed in the manuscript is similar to work published by Beasley-Green A, et al. Proteomics 12, 923-31 (2012); however, this publication is not referenced in the manuscript.

Reply:

Thanks for the reviewer's detailed suggestion.

(1) Quality control has become a key process in proteomics. Rudick PA, et al. Mol Cell Proteomics 9, 225-41(2010) described for the first time how computational QC metrics (called MSQC) can be used to objectively assess the quality of a mass spectrometry proteomics experiment. However, the metrics mentioned in Rudick PA, et al. study were designed based on the requirements of LC-MS technology 14 years ago. With the advancement of LC-MS techniques, such as DIA and large-scale studies, quality control-related metrics and software also need updates to meet new demands.

In the revised manuscript, we comprehensively analyzed and integrated the metrics proposed by existing quality control software. Building upon the initial 70 metrics in the previous manuscript, we included an additional 11 intra-experiment metrics from existing software, generating a total of 81 metrics (**Supplementary Note 1**). We attempt to provide the most comprehensive set of QC metrics for proteomics. We also made modifications in the **Results** "Design of MSCohort for comprehensive data quality control" section in the main text (**Page 5-6, Line 147-161**).

We also added two additional notes related to the MSCohort QC system in the **Supplementary Information**: "Supplementary Note 1: Comparison of the Metrics in MSCohort and the Existing Metrics in other Quality Control Tools" and "Supplementary Note 2: Comparison of MSCohort and other Quality Control Tools". These notes primarily explain the sources of metrics in MSCohort, the newly added metrics, as well as the novelty and application of the MSCohort software.

(2) We apologize for not citing the article by Beasley-Green A et al. Proteomics 12, 923-31 (2012) in the previous manuscript. In the revised manuscript, we have added this reference in Results section (**Page 7, Line 204**) as Reference 37. We think that optimizing LC-MS experiments requires not only quality control software to provide metrics but also software that reports on the relationship between metrics and identification results, explaining in detail why certain experimental conditions and parameters are optimal compared to other options. By conducting a limited number of experiments, it is possible to optimize the entire workflow of proteomics data acquisition, including sample preparation, chromatography, and mass spectrometry steps. In the revised manuscript, we have included a new note: "Supplementary Note 4: Optimization of LC-MS Method for Establishment of SOP for Urinary Proteomics" showed how the optimization strategies for experimental workflows with the MSCohort report include QC metrics output to

optimize parameters and locate experimental problems.

4.6 Pg. 7 lines 182-184: How was the “unified LC and MS” or reference SOP optimized for the study?

Reply:

Thanks for the reviewer’s professional advice. In the revised manuscript, we have provided detailed explanations on “unified LC and MS” and “LC-SOP”. Specifically, we have added a new section in the **Results** titled “Optimization and establishment of SOP for urinary proteomics with MSCohort QC system”, which illustrated the optimization and establishment process of our SOP, along with its advantages. Additionally, in the **Supplementary Information**, we have updated one note: “Supplementary Note 3: The Standard Operating Procedure (SOP) for Urinary Proteomics” and introduced two new notes related to the SOP: “Supplementary Note 4: Optimization of LC-MS Method for Establishment of SOP for Urinary Proteomics” and “Supplementary Note 5: The Novelty of Our Developed Standard Operating Procedure (SOP) for Urinary Proteomics”.

4.7 Figures 3a and 3e should be combined, and the uncertainty (error bars) of the protein identification results for each platform should be included.

Reply:

Thanks for the reviewer’s detailed suggestion. In the revised manuscript, according to your suggestion, we have added error bars to revised **Figures 4a and 4e**. To maintain consistency and clarity with the sequence described in the manuscript, we have not merged Figures 4a and 4e. But we added the reference lines in the figures to enhance the visual comparison.

4.8 The color scheme in Figure 3 should be maintained for each study condition (with LC-SOP-blue and without LC-SOP-orange) for clarity.

Reply:

Thanks for the reviewer’s detailed suggestion. Following the reviewer’s guidance, we have standardized the color scheme in **revised Figure 4** to align with "with LC-SOP" in blue and "without LC-SOP" in orange for clarity. The colors of Pearson correlation matrix have not been modified to ensure a more intuitive comparison.

4.9 The y- and x-axis labels for the figures should provide clear and concise information about the data presented. For example, the y-axis labels for Fig. 3b and 3f (“intersection size” for vertical bars and “set size” for horizontal bars) do not align with the figure text. The text states that the vertical bars represent the proteins identified across the platforms (or inter-/between-platform data) and the horizontal bars represent the proteins identified in each platform (or intra-/within-platform data).

Reply:

Thanks for the reviewer's detailed suggestion. Following the reviewer's guidance, we have updated the y-axis and x-axis labels in **revised Figure 4** to align with the figure text.

4.10 The discussion section states that "more than 500 proteome experiments" (pg 14 lines 447-449) were performed; however, this statement does not align with the experimental design in the methods section or the supplemental sections.

Reply:

Thanks for the reviewer's suggestion. The "more than 500 proteomics experiments" mentioned in the **Discussion** section refer to the data collected from a total of 160 samples, including 80 cases of colorectal cancer (CRC) and 80 healthy control (HC) samples, across three mass spectrometers. Each sample was run once on each instrument, resulting in a total of 480 proteomics experiments. During the data acquisition process on each instrument, urine quality control (QC) samples were collected daily to assess instrument stability and repeatability, leading to a total of 47 QC proteomics experiments. Therefore, in total, we obtained 527 proteomics experiments. We have provided a detailed explanation of these experiments in the **Results** section.

(Page 11-12, Line 357-365) "We further illustrated the generalization by applying the above urinary proteomics SOP to clinical biomarker discovery research in different platforms. Herein, we collected a clinical cohort comprising 80 urine samples from colorectal cancer (CRC) patients and 80 samples from matched healthy controls (HC) (The detailed clinical information is shown in Supplementary Table 6). Based on the above SOP, LC-MS data collection for these 160 samples was conducted on Lumos, E480, and TIMS, respectively (see Methods). QC samples were randomly analyzed during the collection process for systematic evaluation of reproducibility. In total, the three platforms generated 527 DIA experiments including 47 QC experiments."

4.11 In the Methods section (pg 16 lines 494-510), the author states that the peptide concentration is determined via colorimetric assay. How is the protein concentration determined for the accurate enzyme-to-protein ratio for the digestion protocol? What is the concentration of peptide material aliquoted prior to lyophilization (line 505)?

Reply:

Thanks for the reviewer's detailed suggestion. In the revised manuscript, we have added detailed experimental operational procedures for the "Preparation and distribution of the quality control urine samples" procedure in the **Methods** section (**Page 17-18, Line 549-574**).

Briefly, pooled urine mixture was reduced with 20 mM dithiothreitol (DTT) and alkylated with 50 mM iodoacetamide (IAM), then urine proteins were pelleted using 6-fold volume precooled acetone. **The concentration of pooled urine proteins was quantified using Pierce™ BCA protein assay kit (Thermo Fisher Scientific, USA) following the manufacturer's protocol.** For each well, **one hundred micrograms**

of proteins were transferred to the 96-well PVDF plate. The samples were then washed three times with 200 μL of 20 mM Tris buffer (pH 8.0) and centrifugated. Proteins were digested by adding 30 μL of 20 mM Tris buffer (pH 8.0) with trypsin at ratio 50:1 (w:w) on membrane (**2 μg trypsin for each well**). The eluted peptides were combined together, and purified with Sep-Pak C18 Vac Cartridge (Waters). The concentration of pooled peptides was determined by using PierceTM Quantitative Colorimetric Peptide Assay kit (Thermo Scientific) following the manufacturer's protocol. **The concentration of peptide material prior to lyophilization was 0.6 $\mu\text{g}/\mu\text{L}$.** And then, the peptides were aliquoted to 20 μg each tube and lyophilized.

Point-by-point Response to Reviewers' Comments

Standard operating procedure combined with comprehensive quality control system for multiple LC-MS platforms urinary proteomics

Reviewer #1 (Remarks to the Author):

Thanks for the revisions. A few more comments emerged from the revision.

1. In the clinical cohort application, the authors mentioned that the software detected several issues. However, in Supplementary Figures 9b and 9c, I noticed that some files (e.g., D1259, H71) show very low average scores (light yellow) in Figure 9c, but these files are not marked as "low-quality". How should this discrepancy be explained?

Reply:

Thanks for the reviewer's valuable suggestion. In this study, MSCohort detected and reported low-quality experiments based on the isolation forest (iForest) algorithm. First, the 23 inter-experiment metrics values for each experiment in the cohort were integrated into a two-dimensional matrix. Then, the outlier experiments detection was performed using the iForest algorithm with a two-stage process.

iForest algorithm comprehensively evaluates the performance of all metrics. For some experiments (e.g., D1259, H71), although some metrics showed a low score, others performed acceptable. Therefore, the algorithm did not determine them as outliers. In the context of clinical cohort applications, given the individual differences and various influencing factors, it is virtually impossible for every experiment to exhibit high similarity to all others across all metrics. Inevitably, some experiments will have metrics that are less than ideal. Our objective is to identify the experiments that are most anomalous or those that stand out as the most excluded based on this model.

2. The authors mention many metrics in the metrics section, but when using the software, they refer to vague terms such as "quantitative intensity" for analysis. What intensity? Need to specify. This issue arises multiple times throughout the manuscript. It is essential to clearly define which metrics were used in the analysis.

Reply:

Thanks for the reviewer's valuable suggestion. In the revised manuscript, we have supplemented specific quantitative intensity metrics in both the main text and the supplementary information.

3. In Supplementary Table 1A, there is a spelling error ("ionzation") and formatting issues, such as extra spaces. Similar issues should be checked throughout the manuscript.

Reply:

Thanks for the reviewer's detailed inspection. In the revised manuscript, we have corrected the spelling errors and conducted a thorough check and revision of the formatting issues throughout the manuscript.

4. Some figure captions and titles in the main text are unclear. For instance, "median peaks number of MS1" should be revised to "median peak count in MS1 spectra" for clarity.

Reply:

Thanks for the reviewer's valuable suggestion. In the revised manuscript, we revised "median peaks number of MS1" as "median peak count in MS1 spectra". We also conducted a thorough check and revision of the figure captions and titles in the main text.

Reviewer #2 (Remarks to the Author):

The effort the authors put into addressing the reviewers' comments is acknowledged and appreciated. The authors conducted additional experiments and revised their manuscript accordingly by adding new data and text to address the concerns raised by the reviewers. The manuscript has also been restructured to effectively highlight the main points the authors would like to convey regarding the novelty and applicability of their MSCohort QC system. The overall quality of the manuscript has been improved considerably. It would be helpful for the authors to add their response to my general comment #2 regarding the evaluation of different LC systems to their manuscript. Otherwise, I do not have any additional concerns regarding the soundness of the research described in the manuscript.

Reply:

Thanks for the reviewer's valuable suggestion. To better address the reviewer's comments, we have specifically augmented the dataset. We believe the reviewer's concerns is **“a less-than optimal experimental design wherein the same LC system was not evaluated in combination with each MS system” in general comment #2**. In response to this, we specifically conducted two sets of comparisons in two LC-MS platforms:

(1) The same LC system (UltiMate 3000) was connected to different mass spectrometers (Orbitrap Exploris 480 and timsTOF Pro 2), while maintaining **same** chromatographic conditions. The results indicated that the number of identified proteins was generally comparable (3631 vs. 3767), and the Pearson correlation of protein intensity between two LC-MS platforms was also relatively high (**For-review Figure 1a**).

(2) The same LC system (UltiMate 3000) was connected to different mass spectrometers (Orbitrap Exploris 480 and timsTOF Pro 2), while maintaining **different** chromatographic conditions. In the absence of a standardized operating procedure (SOP), different analytical columns and different analytical gradients were employed. The results indicated that the number of identified proteins revealed a notable disparity (2419 vs. 3502), and the Pearson correlation of protein intensity between two LC-MS platforms was also relatively low (**For-review Figure 1b**).

The chromatographic condition and the summary of metrics values reported from MSCohort were listed in **For-review Table 1**.

The above MS data file can be freely downloaded at iProX (Integrated Proteome resources, <http://www.iprox.org>; link: <https://www.iprox.cn/page/PDV0141.html>; url=<https://www.iprox.cn/page/DSV021.html?url=1735309056189XQrW>; password: bqm8).

For-review Figure 1. The Pearson correlation matrix based on log₂ protein intensity of the common proteins from 2 platforms with LC-SOP (a) and without LC-SOP (b). Three technical replicates were acquired on each platform.

		The same LC condition		Different LC condition	
		A: E480-withSOP	B: TIMS-withSOP	C: E480-withoutSOP	D: TIMS-withoutSOP
LC method	File name	E480-withSOP_DIA_1.raw	TIMS-withSOP-DIA-1.d	E480-withSOP_DIA_1.raw	TIMS-withoutSOP-DIA-1.d
	LC type	UltiMate 3000	UltiMate 3000	UltiMate 3000	UltiMate 3000
	MS type	Orbitrap Exploris 480	timsTOF Pro 2	Orbitrap Exploris 480	timsTOF Pro 2
	Loading amount	2 µg	500 ng	2 µg	500 ng
	Precolumn	no	no	no	no
	Analytical column	Monolithic Column, 50 cm × 50 µm	Monolithic Column, 50 cm × 50 µm	Monolithic Column, 50 cm × 50 µm	Monolithic Column, 15 cm × 75 µm
	Mobile phase A	100% H ₂ O/0.1% FA	100% H ₂ O/0.1% FA	100% H ₂ O/0.1% FA	100% H ₂ O/0.1% FA
	Mobile phase B	100% ACN/0.1% FA	100% ACN/0.1% FA	100% ACN/0.1% FA	100% ACN/0.1% FA
	Flow rate (nL/min)	500	500	500	500
	Gradient	30min 0min, 4%B; 22min, 16%B 25.3min, 24%B 25.4min, 72%B 26.4min, 72%B 26.5min, 1%B 30min, 1%B	30min 0min, 4%B; 22min, 16%B 25.3min, 24%B 25.4min, 72%B 26.4min, 72%B 26.5min, 1%B 30min, 1%B	30min 0min, 5%B; 15min, 20%B 20min, 30%B 21min, 90%B 25min, 90%B 25.1min, 1%B 30min, 1%B	30min 0min, 4%B; 22min, 16%B 25.3min, 24%B 25.4min, 72%B 26.4min, 72%B 26.5min, 1%B 30min, 1%B
Mobile phase					
Metrics-Values		A: E480-withSOP	B: TIMS-withSOP	C: E480-withoutSOP	D: TIMS-withoutSOP
Sample	S1. Missed cleavages(n=0) of peptides	86.66%	87.64%	0.8489	87.81%
	S2. Median peptide length	10	11	11	12
	S3. Median Total Ion Current (TIC)	3.11E+12	3.21E+10	3.703E+12	7.15E+10
Chromatography	C1. Chromatographic invalid acquiring time(min)	2	1	2	1
	C2. Median of full width at half maximum (FWHM)(min)	0.12	0.1	0.09	0.12
	C3. Median of peak width(min.)	0.2	0.14	0.15	0.18
	C4. Proportion of peptides with long eluting width	0.16%	0.03%	0.0021	0.04%
	C5. Median precursors identified over RT	955	748	949	830
	C6. Max. identification rate over RT	0.9494	0.7696	0.9372	0.7678
MS2 Windows	W1. Acquired MS2 scans in one cycle	80	50	60	53
	W2. Median of window size (Da)	6	16	11	16
	W3. Redundant identified precursors/ Identified Scan Rate	2.60	3.83	2.59	4.79
	W4. Redundant identified precursors/ Identified precursors R	1.53	6.08	1.62	5.91
	W5. Identified precursors/ identified scan rate	1.7	0.63	1.6	0.81
	W6. MS1 data point per peak	6	6	6	6
	W7. MS2 data point per peak	1	6	1	6
MS1 and MS2 Signal	M1. Proportion of MS1 time	0.13	0.07	0.17	0.05
	M2. Median MS1 ion injection time(ms)	1.41	-	1	-
	M3. Median MS2 ion injection time(ms)	50	-	43.98	-
	M4. MS1 cycle time(s)	2.09	1.5	1.67	2.02
	M5. MS2 cycle time(s)	6.4	1.5	5.02	2.02
	M6. Median peaks intensity of MS1	2.28E+09	1.02E+07	2773000000	3.61E+07
	M7. Median peaks number of MS1	1624	26476.5	1999	113837.5
	M8. Median peaks number of MS2	404	2754	671	3357
	M9. Median peaks intensity of MS2	7.75E+06	1.76E+05	22450000	4.05E+05
	M10. Identified / detected features	0.85	-	0.72	-
	M11. Signal to noise (S/N) for precursors	11.89	11.48	16.1	16.99
	M12. Median of MS1 raw mass accuracy (ppm)	1.1	1.26	2.83	4.45
M13. Median of MS2 raw mass accuracy (ppm)	1.9	0.94	1.95	4.73	
M14. Median intensities for precursors	5.69E+03	5.45E+01	99600	3.03E+02	
M15. Median intensities for peptides	5.70E+03	5.84E+01	87720	3.20E+02	
M16. Median intensities for protein groups	6.27E+03	7.17E+01	109000	3.46E+02	
M17. Acquired MS1 scans per minute	28.13	41.38	35.88	29.6	
M18. Acquired MS2 scans per minute	749.4	2070.07	717.08	1570.03	
M19. MS2 identification rate	0.74	0.63	0.75	0.6	
M20. Max. identified precursors rate over M/Z range	1.8	0.84	1.62	1.13	
Identification Result	ID1. Identified MS2 scans	16568	37776	13392	28456
	ID2. Number of precursors	28137	23772	21399	23049
	ID3. Number of peptides	22507	21224	15773	19176
	ID4. Number of protein groups	3631	3767	2419	3502
	ID5. Number of proteins	3719	3857	2545	3657
	ID6. AVG peptides per protein group	6.2	5.63	6.52	5.48
	ID7. AVG precursors per protein group	7.75	6.31	8.85	6.58

For-review Table 1. Chromatographic condition and summary of metrics values.

Reviewer #3 (Remarks to the Author):

Reply:

Thank you and we are very glad to get comments and feedbacks from Early Career Researchers.

Reviewer #3 (Remarks on code availability):

Reply:

Thank you and we are very glad to get comments and feedbacks from Early Career Researchers.

Reviewer #5 (Remarks to the Author):

In their revised manuscript, the authors have taken strides forward by addressing comments from the earlier revision. While some sections of text remain somewhat nebulous, the revision helped clarify the metric formulations and their applications.

Reply:

Thanks for the reviewer's positive comments. We have tried to address the following points to further improve our manuscript.

Supplementary Note 1 in the "sm3njz" Supplementary Information provides a table detailing the 58 intra-experiment metrics employed by MSCohort, and the authors indicate novel metrics by blue text. They observe that DIA has been less well-supported by quality metrics software, and they're correct in that assessment. An interesting preprint describing iDIA-QC appeared in June of 2024, but I believe it has not appeared "in press" yet. The choices they have made for DIA metric inclusion seem reasonable. I do not believe that I could reproduce all their metrics based on the five- or six-word descriptions, but their captions are reasonably clear.

Reply:

Thanks for your comments. The first author of this study, Xiang Liu, is one of the co-authors of the iDIA-QC paper (ranked 23rd). In the iDIA-QC article, the primary focus was on quality control of LC and MS system status based on dedicated QC samples. The 15 proposed metrics were specifically designed for these dedicated QC samples and, consequently, were unable to extract metrics intrinsic to the actual samples themselves. Furthermore, they couldn't perform quality assessment and optimization on individual samples within a cohort or across different samples. In this study, building upon the iDIA-QC research, we have expanded the number of metrics, striving to provide the most comprehensive set of proteomics QC metrics. Additionally, we have introduced more algorithms and diverse data presentation formats to achieve comprehensive quality assessment and optimization for individual samples and inter-experiment analysis in a cohort.

The description of applying the metrics is improved. The authors describe an interesting mapping process for grading each metric to one of five values; I believe the intent here is to contextualize a given metric in light of the values produced by all other experiments in a set. The authors describe the use of unsupervised learning to judge whether a RAW file represents an outlier (inconsistent performance); this has been handled in other cases by PCA, but unsupervised learning can certainly be applied here.

Reply:

Thanks for the reviewer's comments.

First, as the reviewer mentioned that we grade each metric to one of five values to contextualize a given metric in light of the values produced by all other experiments in a set. By normalizing the metrics across different measurements, we can ensure the comparability of the data. Since the measurements of various metrics differ, merely displaying their values would not facilitate a comprehensive evaluation for users. For example, the unit of intra-experiment metric "M12. median of MS1 raw mass accuracy" is "ppm", and the unit of intra-experiment metric "C3. Median of peak width" is "minute". Therefore, we assigned scores to each metric, which help users to quickly find low-performance metrics.

Second, previous studies have shown that the LC-MS data is complex and high-dimensional, and unsupervised machine learning algorithms are commonly used for outlier LC-MS data analysis ((Bittremieux, et al. *Proteomics*, 2017, 17(3-4); Stanfill, et al. *Mol Cell Proteomics*, 2018, 17(9):1824-1836)). In essence, PCA is a dimensionality reduction method for unsupervised clustering and can only be used for classification after improvements. For example, Matzke, et al. (*Bioinformatics*, 2011, 27(20):2866-2872) and Stanfill, et al. (*Mol Cell Proteomics*, 2018, 17(9):1824-1836) applied robust principal components analysis (rPCA) method, and the robust Mahalanobis distance was used to score the instrument runs transformed to the rPCA space. However, this calculation method requires choosing and setting a baseline, followed by fitting a model to determine the cutoff. A baseline data set consisting of high-quality instrument runs is critical and has a significant impact on both model training and the screening results. In this study, we applied an unsupervised and online outlier detection algorithm, Isolation forest (iForest), to distinguish outlier experiments. This algorithm does not require a labeled dataset or pretraining of offline models, it can dynamically construct isolation trees online for any batch of data. It has unique advantages in dealing with large datasets due to its low-computational complexity (Grekov, A.N. et al. *Sensors (Basel)*, 2023, 23, 2687).

On the other hand, I believe this statement represents an overclaim: "Through the DIA scoring formula (1) and comprehensive metrics analysis reports, MSCohort provides direct explanations for the underlying causes of data results." The DIA scoring formula relates identifications to precursors and MS/MS scans, so it surely contributes little to identify specific causes of DIA failure. If the electrospray needle "coughs" in the middle of an experiment, it may impact identification relatively little even though it disrupts quantification for a range of retention times; no aspect of the scoring

formula (1) would capture that event.

Reply:

Thanks for the reviewer's comments. Extracting more accurate and comprehensive metrics for quality control is critical. The DIA scoring formula (1) serves to categorize and describe the metrics, facilitating rapid identification of issues. Without this formula or the classification of metrics, users would be confronted with numerous metrics, lacking a clear comprehension of the original steps leading to problematic metrics or how these metrics influence identification outcomes. MSCohort not only provides the scoring formula but also offers a detailed report, including a metric-score diagram (Supplementary Figure 1), where all metrics are mapped onto a two-dimensional matrix. This intuitive display the metrics with low scores to assist experimenters in assessing the quality of data.

In addition, MSCohort generates a report with comprehensive tables and charts. As shown in the For-review Figure 2, if the electrospray needle "coughs" in the middle of an experiment, the graph of accumulated number of MS2 scans or precursors (For-review Figure 2a), scatter plot of the peak width for each of the identified precursors (For-review Figure 2b) and identification rate per unit time (For-review Figure 2c) in the MSCohort analysis report can reflect the anomaly of the experimental system and its impact on the identification rate of MS2 spectrum. This raw data is the M10-ZenoTOF-DIA1 file, and a detailed and complete MSCohort analysis report is provided in ProteomeXchange Consortium (<https://proteomecentral.proteomexchange.org/cgi/GetDataset?ID=PXD050291>).

a

b

For-review Figure 2. The MSCohort report for an experiment with electrospray needle "coughs" in the middle of an experiment. **a** Graph showing the accumulated number of MS2 scans or precursors for this experiment. The number of identified MS2 scans and identified precursors do not show a significant increase from 5 minutes to 8 minutes. **b** The scatter plot showing the peak width for each of the identified precursors. The identified precursors is rare from 5 minutes to 8 minutes. **c** The bar plot showing the identification rate of MS2 scans per minute. The identified rate is low from 5 minutes to 8 minutes.

The authors delay citations for "96 DRA-Urine" and "HRMS1" to relatively late in the manuscript even though both make their first appearances earlier. I was surprised to see that the SOP DIA methods (Supplementary Note 3) rarely include overlaps between successive windows. This can reduce sensitivity when peptide ions have m/z values very close to the window boundary. I believe that the inclusion of 80 windows in the Orbitrap DIA method will mean that fragment ion chromatograms are updated with new intensities not more than once every nine or ten seconds, destroying the quantitative value of the fragment ion chromatograms. The frequency of MS scans will be higher due to multiple MS events per cycle, so the precursor chromatograms should be reasonably good. The shortcomings in fragment ion chromatograms would likely be increased problems with "interference," when different precursors have similar m/z and retention time (we have battled this problem with in-house use of the "High-Resolution MS1" protocol for DIA).

Reply:

Thanks for the reviewer's detailed inspection.

First, we have added citations for "96 DRA-Urine" and "HRMS1" in the "Optimization and Establishment of SOP for Urinary Proteomics with MSCohort QC System" section of the revised manuscript (Page 7, Line 232-235).

Second, the setting of DIA methods in Supplementary Note 3 is successive between 80 variable DIA windows with 1 Da overlap. This method setting is based on the DIA article (*Nat Commun*, 2017, 8(1): 291; *Nat Commun*, 2020, 11(1): 5248), comparable to the method provided by current commonly used in DIA article.

Third, we agree with the reviewer's opinion that 80 windows in the Orbitrap DIA method will lead to fewer quantitative points in fragment ion spectra. Therefore, we utilized the method that involves multiple MS1 scans per duty cycle to ensure that a sufficient number of MS1 scan events could be acquired. Quantification was based on precursor ion signals measured through high-resolution full MS scans with 120k resolution setting, and the MS2 scan with 30k resolution setting was utilized for peptide identification only.

The authors praise their SOP for its technical achievement in individual samples, and yet the goal of an SOP is comparability among instruments rather than excellence in individual experiments. I would have expected to see more commentary about improved CV values for invariant peptides or proteins to establish its quantitative stability.

Reply:

Thanks for the reviewer's comments. We first optimized sample, chromatography, and mass spectrometry methods with urine QC samples on individual instrument (Supplementary Note 4: Optimization of LC-MS Method for Establishment of SOP for Urinary Proteomics). Then, In the "Comprehensive and comparative analysis of urinary proteome data from multi-platform study" section and Figure 4 of our manuscript, we demonstrated that the implementation of the SOP significantly enhances the comparability among results obtained from ten LC-MS platforms using the SOP, compared to those from ten LC-MS platforms without the SOP. In terms of proteomic qualitative results, the number of identified proteins overlapping across ten LC-MS platforms (namely invariant proteins) increased from 2045 to 3080 (Figure 4b, 4f) after employing an SOP. For proteomic quantitative results, due to the differences in experimental platforms, the quantitative intensities of proteins obtained from different instruments such as Orbitrap and timsTOF are not fully comparable. Therefore, we applied Pearson correlation to assess the quantitative reproducibility among instruments. The Pearson correlation coefficient for quantitative protein intensities among different instruments improved from 0.9 to 0.93 (Figure 4c, 4h) after employing an SOP.

The authors have used human SwissProt proteins from the reference proteome rather than the full reference proteome. If they wanted their sequence database to contain a sequence for every protein-coding gene in the human genome, they cannot restrict the TrEMBL components of the reference proteome from inclusion. The database they've chosen here omits many human genes. This is a common phenomenon for human proteomics papers, but I believe most authors are unaware that they are omitting real protein-coding genes by the practice.

Reply:

Thanks for the reviewer's comments. We utilized the human Swissprot database to ensure comparability with the urinary proteomics research results reported in the literature. From the perspective of method development in this paper, researchers can also use our Standard Operating Procedure (SOP) to search the TrEMBL database, this is merely a setting option within the software.

I would have hoped to see mzML rather than .ms1, .ms2 for metric computation, but I appreciate that simple text formats are easier to manage.

Reply:

Thanks for the reviewer's comments. As you have pointed out, we use the .ms1 and .ms2 formats for easier to manage and review. Currently, software such as msconvert, which is commonly used in the field, can also be employed to export the original mass spectrometry files into mzML format.

The manuscript frequently contains missing articles ("a," "an") or other language problems. I would point to "outstanding success in most sceneries" (meaning scenarios) or "anomalies nature" (meaning anomalous) since they appear in close proximity.

Reply:

Thanks for the reviewer's detailed inspection. In the revised manuscript, we carefully examined and corrected the language problems.